# High-entropy alloy Janus artificial enzymes for pH-gated sequential redox therapy of drug-resistant bacterial infection

Cong Han[1,6], Yongqi Wang[2,6], Shihuan Gao[2], Ting Wang[2], Huili Du[1,3], Jie Long[1], Weidong Tian[1], Mohsen Adeli [4], Liang Cheng[5], Zhi Liu[1] ✉, Tian Chen [1,3] ✉ & Chong Cheng [1,2] ✉

Drug-resistant bacterial infections in chronic wounds remain a critical challenge, particularly under persistent inflammation. Here, we report the de novo design of high-entropy alloy (HEA, PtFeCuCoNi)-based Janus artificial enzymes with pH-gated redox biocatalysis for sequential antibacterial and repair functions. The multi-metal synergy stabilizes the *d*-band center, allowing acidic oxidase/peroxidase-like activity and neutral antioxidase-like activity. In infection, the enzymes generate bactericidal reactive oxygen species (ROS) to eliminate methicillin-resistant *Staphylococcus aureus* (*MRSA*) and biofilms at ultralow concentrations (8 µg/mL). During healing, they scavenge ROS, alleviate oxidative injury and support cellular proliferation. In *MRSA*-infected wounds, this dual-action system clears bacteria and then accelerates regeneration through enhanced neovascularization and matrix remodeling. Mechanistic analyses reveal *PFKFB3*-mediated metabolic reprogramming, suppression of pro-inflammatory cytokines, and macrophage polarization toward the M2 phenotype. Integrating pH-gated antimicrobial and immunomodulatory repair within one nanoplatform, this strategy addresses the conflicting demands of infection control and tissue healing.

Chronic non-healing wounds infected with antibiotic-resistant bacteria present a growing global health challenge, compounded by the complex pathophysiology of these persistent lesions[1–4]. The wound microenvironment exhibits dynamic pH evolution: initial bacterial proliferation and biofilm formation create an acidic milieu through organic acid secretion, which simultaneously compromises host immunity and diminishes antibiotic effectiveness[5]. When sterilization is completed, the wound will undergo chronic inflammation with a near-neutral pH condition, sustained pro-inflammatory macrophage (M1) phenotype activity, and pro-inflammatory cytokine production, which will facilitate the production of massive reactive oxygen species (ROS) that impair vascular and stromal cell function, stalling tissue repair[6,7]. Current approaches, including the antibiotics, peptides, and charged polymers, fail to achieve effective microbial eradication and anti-inflammatory action concurrently[8]. The development of non-antibiotic and pH-responsive agents that can sequentially eliminate pathogens in acidic environments and then achieve anti-inflammatory and immunomodulatory modes to promote wound healing, therefore, represents a critical therapeutic frontier.

[1]State Key Laboratory of Oral Diseases, National Center for Stomatology, National Clinical Research Center for Oral Diseases, West China Hospital of Stomatology, Sichuan University, Chengdu, China. [2]College of Polymer Science and Engineering, State Key Laboratory of Advanced Polymer Materials, Sichuan University, Chengdu, China. [3]Department of Orthodontics, West China Hospital of Stomatology, Sichuan University, Chengdu, China. [4]Institute of Chemistry and Biochemistry, Free University of Berlin, Berlin, Germany. [5]Department of Materials Science and Engineering, The Macau University of Science and Technology, Taipa, Macau, China. [6]These authors contributed equally: Cong Han, Yongqi Wang. ✉e-mail: zliu830@scu.edu.cn; tchen0629@scu.edu.cn; chong.cheng@scu.edu.cn

Peroxisome-mediated ROS regulation plays a pivotal role in innate immunity, orchestrating precisely controlled oxidative bursts that eliminate pathogens while protecting host tissues[9]. These organelles employ peroxisomal oxidases (OXD) and peroxidases (POD) to generate targeted ROS attacks that disrupt bacterial membranes, oxidize vital biomolecules, and degrade protective biofilms[10,11]. Following pathogen clearance, antioxidases, such as catalase (CAT), decompose residual hydrogen peroxide ($H_2O_2$) into water and oxygen, restoring redox equilibrium and safeguarding tissue integrity[12–14]. However, chronic inflammatory conditions disrupt this delicate balance, leading to sustained ROS production that overwhelms endogenous defenses and perpetuates tissue damage through excessive oxidative stress and amplified inflammatory signaling. While peroxisomal enzymes exemplify nature's sophisticated ROS management system, their therapeutic potential is constrained by poor pharmacokinetic profiles and immunogenic risks[15]. This limitation has spurred the development of engineered ROS-modulating platforms as next-generation anti-infective strategies. Current nanotechnology approaches, spanning inorganic nanoparticles (NPs) to metal-organic frameworks[16,17], attempt to recapitulate enzymatic ROS generation and scavenging capabilities[18]. Yet most synthetic systems still lack the precision of their biological counterparts, often exhibiting suboptimal catalytic efficiency and spatiotemporal control that result in either insufficient antimicrobial activity or off-target oxidative damage.

High-entropy alloy (HEA) represents an emerging paradigm in enzyme-mimetic catalysis[19,20], leveraging its unique multi-element synergy to overcome limitations of conventional catalysts[21]. These alloys incorporate five or more transition metals in near-equiatomic ratios (5–35%), where the high configurational entropy, pronounced lattice distortion, and sluggish diffusion kinetics collectively stabilize single-phase solid solutions[22–24]. This distinctive "cocktail effect" generates an exceptionally diverse array of active sites while enabling fine-tuning of the $d$-band center and oxygen adsorption characteristics, which may enable superior catalytic performance and stability compared to monometallic systems[25,26]. While HEAs have made significant progress in the fields of energy conversion and environmental catalysis, their application in biomedical redox regulation remains largely unexplored, constrained by demanding activation conditions, aqueous instability, and unresolved biocompatibility issues[27,28]. Realizing their therapeutic potential will necessitate precise engineering of HEA with optimized size distributions[29], tailored surface chemistries[30,31], and intelligent pH-responsive behavior to achieve spatiotemporal control over ROS modulation in physiological environments[32].

Drawing inspiration from the dynamic redox regulation of peroxisomes, we report the de novo design of HEA, PtFeCuCoNi-based Janus artificial enzymes with pH-gated redox biocatalysis for sequential therapy of drug-resistant bacteria and inflammatory wounds. The precisely balanced incorporation of five transition metals with comparable atomic radii enables synergistic electronic effects while minimizing precious metal content. Density functional theory (DFT) reveals strong $d$-electron interactions within the lattice HEA NPs that redistribute electron density near the Fermi level, optimizing oxygen intermediate adsorption/desorption at catalytic sites (Fig. 1a). In acidic infected wounds, these NPs exhibit potent OXD/POD-mimetic activity, generating bactericidal ROS bursts that eliminate methicillin-resistant *Staphylococcus aureus* (*MRSA*) and biofilms at ultralow concentrations (8 μg/mL). As infection resolves and pH normalizes, the particles switch to superoxide dismutase (SOD)/CAT-like functions, quenching excess ROS and mitigating oxidative damage (Fig. 1b). Kinetic analyses demonstrate exceptional catalytic efficiency (POD-like activity: the maximum reaction rate $(V_{max})$ = 10.23 μM s$^{-1}$; CAT-like activity: $V_{max}$ = 74.69 μM s$^{-1}$), outperforming existing ROS-catalytic materials while maintaining minimal cytotoxicity at therapeutic doses. Crucially, the platform orchestrates dual-phase immunomodulation, which directs macrophage polarization in a stage-dependent manner, first enhancing M1-mediated bacterial clearance and then inducing anti-inflammatory macrophage (M2)-driven regenerative responses. In vivo studies confirm rapid pathogen clearance coupled with protection of endothelial cells, fibroblasts, and macrophages, resulting in accelerated wound closure. This HEA-based Janus artificial enzymes with pH-gated redox biocatalysis represents a programmable, antibiotic-free approach that concurrently addresses antimicrobial resistance and chronic inflammation while promoting regeneration (Fig. 1c).

## Results

### Synthesis and structural characterization

The HEA-based Janus artificial enzymes were synthesized via a one-step solvothermal approach using Pd(acac)$_2$, Fe(acac)$_3$, Cu(NO$_3$)$_2$·3H$_2$O, Co(NO$_3$)$_3$·6H$_2$O, Ni(NO$_3$)$_3$·6H$_2$O as metal precursors (Fig. 2a), N,N-dimethylformamide (DMF) and ethylene glycol (EG) as the solvents. A mauve, finely dispersed mixture of the raw materials was formed through ultrasonication. As the temperature increased, the mixture turned black, possibly due to the coordination and reduction of the five metals. Subsequently, the single-phase HEA formed after the diffusion and rearrangement of atoms. Transmission electron microscope (TEM) and high-resolution transmission electron microscopy (HRTEM) were used to characterize the obtained products. The HEA NPs revealed spherical NPs with uniform morphology (Fig. 2b and Supplementary Fig. 1). Meanwhile, the number-averaged size of PtFeCuCoNi HEA NPs is measured to be ≈122.4 nm in dynamic light scattering, indicative of a stable aqueous phase with low agglomeration tendency. (Supplementary Fig. 2).

X-ray diffraction (XRD) analysis confirmed a face-centered cubic structure (Fig. 2c), with characteristic peaks shifted to higher angles relative to pure Pt (PDF# 35–1358), consistent with alloy formation. We also investigated the effect of precursor ratio on the crystal structure of the HEA NPs (Supplementary Fig. 3). It was observed that the PtFeCuCoNi HEA NPs formed from an equimolar precursor ratio exhibited the most intense XRD diffraction peaks. This is attributed to the near-random distribution of the five constituent elements among the $fcc$ lattice sites, which allows the crystal to maintain superior periodicity and thereby generate strong XRD diffraction peaks. Conversely, when the proportion of any single element was doubled, the excessive metal atoms introduced more intense and localized lattice expansion and strain, thereby disrupting the periodic arrangement of the crystal lattice and ultimately leading to a significant reduction in XRD diffraction peak intensity[33].

Inductively coupled plasma-mass spectrometry (ICP-MS) quantification established a Pt/Fe/Cu/Co/Ni atomic ratio of 19.1:14.7:20.9:22.3:23.0 (Fig. 2d). HRTEM imaging revealed well-defined (111) lattice planes (Fig. 2e), with fast Fourier-transform (FFT) analysis confirming crystalline perfection. Inverse FFT reconstruction showed lattice spacings ranging from 0.212 to 0.215 nm (Fig. 2f and Supplementary Fig. 4), corresponding to ~3.3% intrinsic tensile strain arising from atomic-size mismatch and the ultrathin particle morphology. The low-magnification, high-angle annular dark-field scanning TEM (HAADF-STEM) energy-dispersive X-ray spectroscopy (EDS) images show a homogeneous distribution of Pt, Fe, Cu, Co, and Ni throughout the NPs (Fig. 2g).

Surface chemical states of the PtFeCuCoNi HEA NPs were analyzed by X-ray photoelectron spectroscopy (XPS) (Fig. 2h and Supplementary Fig. 5)[34–36], and distinct photoelectron peaks confirmed the presence of all five metallic components. The Pt 4$f$ spectrum exhibited characteristic doublets at 71.22 eV (4$f_{7/2}$) and 74.54 eV (4$f_{5/2}$), while Fe 2$p$ peaks appeared at 714.12 eV (2$p_{3/2}$) and 723.69 eV (2$p_{1/2}$). Similarly, Cu 2$p$ signals were observed at 931.85 eV (2$p_{3/2}$) and 951.62 eV (2$p_{1/2}$), with Co 2$p$ peaks at 778.14 eV (2$p_{3/2}$) and 792.83 eV (2$p_{1/2}$), and Ni 2$p$ emissions at 852.41 eV (2$p_{3/2}$) and 869.69 eV (2$p_{1/2}$). The systematic shifts in binding energies relative to their pure metallic states, coupled

a  Structure design of the HEA-based biocatalysts

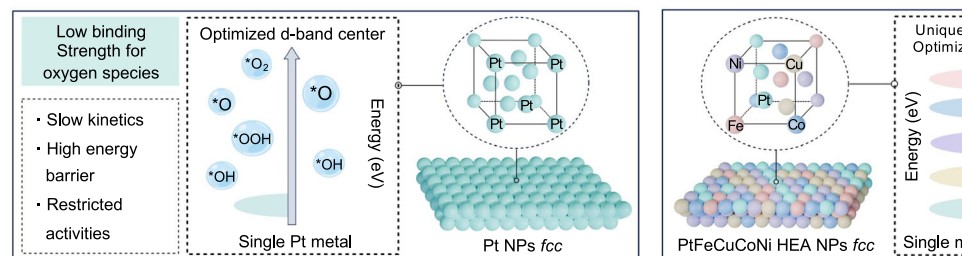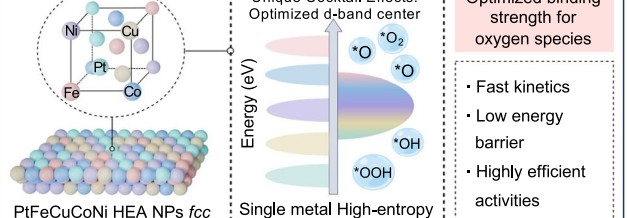

b  Janus artificial enzymes with pH-gated redox biocatalysis

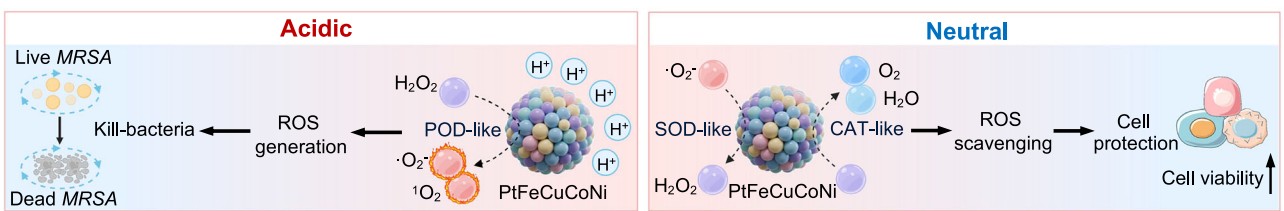

c  Sequential therapy of drug-resistant bacteria and inflammatory wounds

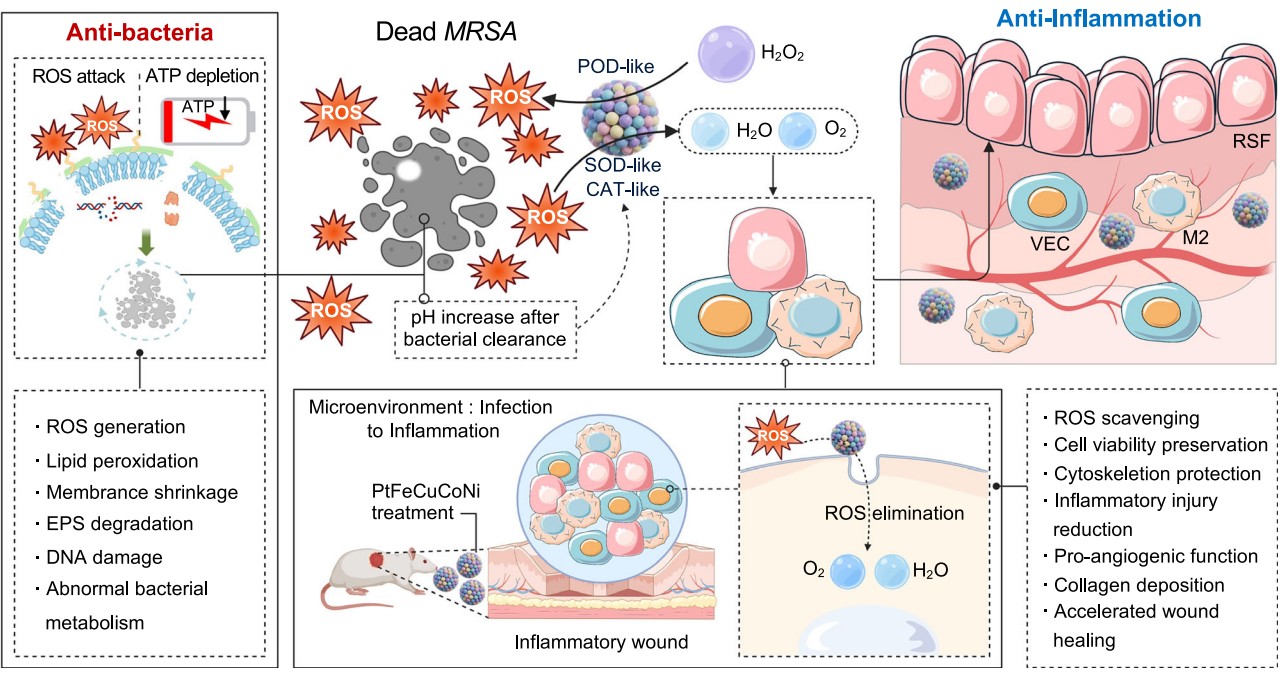

**Fig. 1 | Structure design and biocatalytic properties of HEA-based artificial enzymes. a** Structure advantages of PtFeCuCoNi HEA NPs when compared with bare Pt NPs. **b** HEA-based Janus artificial enzymes with pH-gated redox biocatalysis. **c** Schematic illustration of sequential therapy of drug-resistant bacteria and inflammatory wounds by PtFeCuCoNi HEA NPs (VEC: vascular endothelial cells; M2: anti-inflammatory macrophages; RSF: Rat skin fibroblasts). Created in BioRender. Tian Chen (2025) https://BioRender.com/hs7ivcs.

with relatively weak oxide signatures, provide strong evidence for alloy formation rather than phase-segregated or metal oxides.

X-ray absorption near-edge structure (XANES) analysis at the Pt $L_3$-edge reveals a characteristic white line intensity comparable to Pt foil and markedly reduced relative to $PtO_2$, confirming the predominantly metallic state of Pt atoms in the HEA NPs (Fig. 2i). Fourier-transform extended X-ray absorption fine structure (FT-EXAFS) analysis demonstrates a primary coordination peak at ≈2.3 Å, corresponding to Pt-Pt interactions with a distinct shoulder attributable to Pt-M (M = Fe, Cu, Ni, Co) bonding (Fig. 2j). The complete absence of Pt-O scattering paths provides definitive evidence against surface oxidation[37]. Complementary wavelet transform (WT) analysis of the Pt $L_3$-edge oscillations resolves the spatial distribution of

atomic interactions, with PtFeCuCoNi HEA NPs exhibiting a prominent Pt-M bonding signature at 2.36 Å in $R$-space that is absent in both Pt foil and $PtO_2$ references (Fig. 2k). These results verify the formation of a homogeneous multimetallic coordination environment within the HEA matrix.

## Experimental and theoretical studies on pH-gated ROS-catalytic activities

With the structural and electronic characterization established, we examined the pH-gated enzymatic activities of HEA NPs for treating bacterially infected chronic wounds. In acidic microenvironments of infected wounds, the HEA-based Janus artificial enzymes exhibit potent POD-like activity, catalyzing the conversion of $H_2O_2$ into

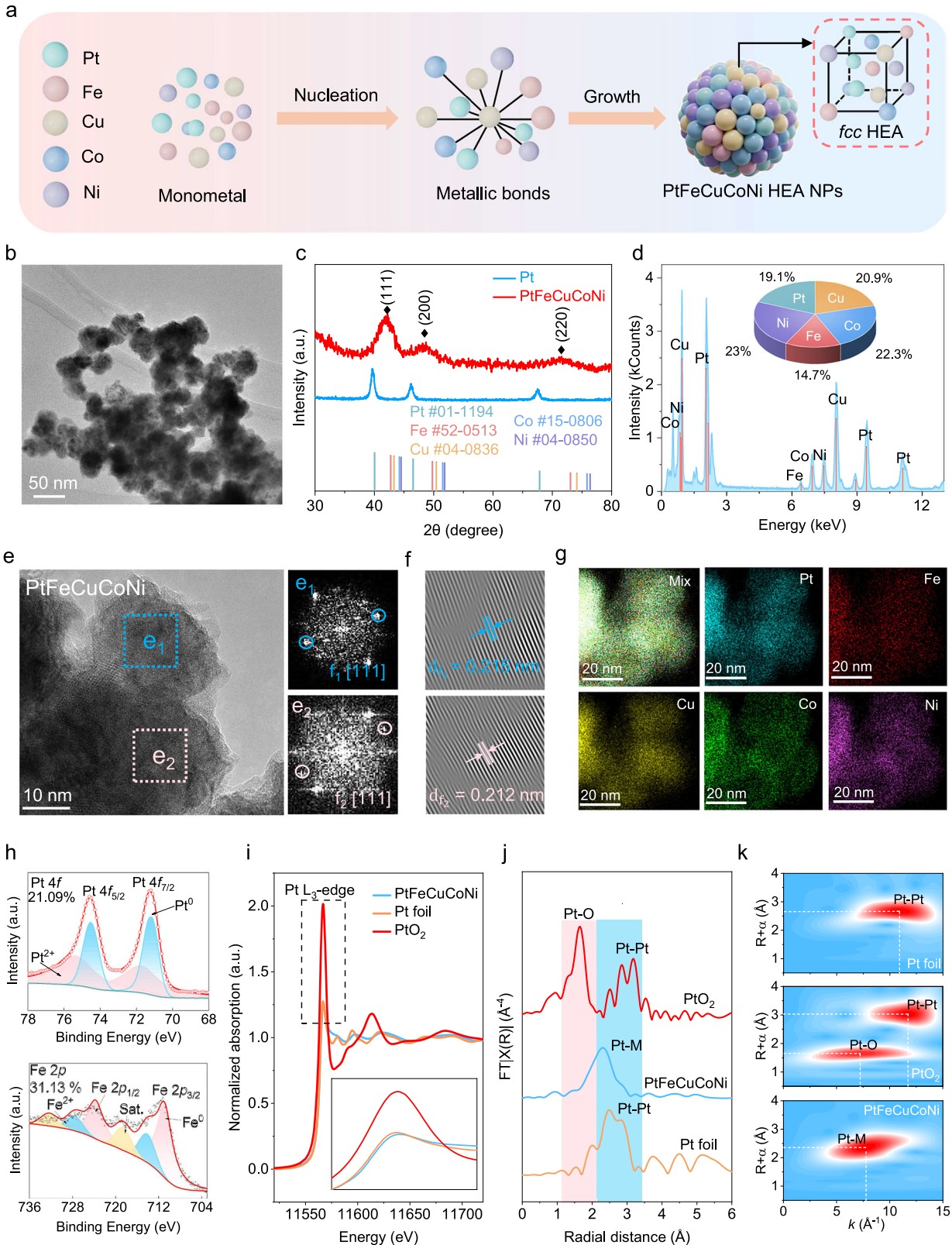

cytotoxic •$O_2^-$ and $^1O_2$ that effectively eliminate bacterial pathogens. As the wound pH normalizes during healing, the HEA-based Janus artificial enzymes transition to a protective mode, sequentially mimicking SOD to dismutase •$O_2^-$ to $H_2O_2$ and CAT to decompose excess $H_2O_2$ into harmless $O_2$ and $H_2O$ (Fig. 3a). This dual-phase catalytic switching not only eradicates infection but also creates a redox-balanced microenvironment conducive to tissue regeneration[38].

Experimental validation of the predicted ROS-catalytic properties began with a quantitative assessment of ROS generation under acidic conditions using a 3,3,5,5-tetramethylbenzidine (TMB) chromogenic assay. The PtFeCuCoNi HEA NPs demonstrated superior POD-mimetic activity, as evidenced by intense oxTMB absorption at 652 nm (Fig. 3b). To elucidate the compositional synergy, we systematically synthesized and evaluated quaternary nanocatalysts (FeCuCoNi, PtCuCoNi,

**Fig. 2 | Synthesis and characterization of the HEA-based Janus artificial enzymes. a** Schematic illustration of the synthesis of PtFeCuCoNi HEA NPs (Created in BioRender. Tian Chen (2025) https://BioRender.com/hs7ivcs). **b** TEM image of PtFeCuCoNi HEA NPs. **c** XRD patterns of PtFeCuCoNi HEA NPs that matched the standard PDF cards. **d** STEM-EDS spectra (inset is the corresponding ICP-OES result) of PtFeCuCoNi HEA NPs. **e** HRTEM image of the PtFeCuCoNi HEA NPs and $e_1$, $e_2$, the corresponding FFT patterns from the dashed areas in (**e**). **f** $f_1$, $f_2$ the corresponding inverse FTT patterns of $e_1$, $e_2$. **g** HAADF-STEM-EDS elemental mapping of

PtFeCuCoNi HEA NPs. Experiments were repeated independently (**b**, **e**, **g**) three times with similar results. **h** Pt $4f$ and Fe $2p$ XPS spectra of PtFeCuCoNi HEA NPs. **i** Pt $L_3$-edge XANES spectra of PtFeCuCoNi HEA NPs and references ($PtO_2$ and Pt foil). **j** Fourier-transformed $k^2$-weighted EXAFS spectra. **k** WT analysis at the Pt $L_3$-edge of different samples. Experiments were repeated independently (**b**, **e**, **f**, **g**) three times with similar results. In (**c**, **h**, **i**), a.u. indicates the arbitrary units. Source data are provided as a Source data file.

PtFeCoNi, PtFeCuNi, and PtFeCuCo) with controlled elemental stoichiometries (Supplementary Fig. 6). Kinetic analysis revealed a hierarchical dependence of catalytic activity on elemental composition: Fe removal caused the most pronounced activity reduction, followed by Pt, Cu, Co, and Ni. The consistent underperformance of all quaternary systems relative to the pentametallic HEA confirms the essential cooperative interactions among all five elements in optimizing ROS-generation kinetics.

Subsequently, we systematically analyzed the characteristic steady-state kinetic constants, including the catalytic constant ($K_m$/mM), $V_{max}/\mu M\,s^{-1}$, and the turnover number (TON/$s^{-1}$). As shown in Fig. 3c and Supplementary Fig. 7, PtFeCuCoNi HEA NPs ($V_{max} = 10.23$, TON = 0.208) exhibited superior POD-like catalytic kinetics compared to the low activity of Pt ($V_{max} = 7.67$, TON = 0.03), highlighting the beneficial impact of the high entropy effect. Furthermore, the enzymatic performance of PtFeCuCoNi HEA NPs surpasses that of most recently reported metal oxides and metal nanoparticle-based POD mimics (Fig. 3d and Supplementary Table 1). We conducted free radical quenching experiments to qualitatively identify the ROS species, which could be identified as $\bullet O_2^-$ and $^1O_2$ (Fig. 3e and Supplementary Figs. 8 and 9). Subsequently, the $^1O_2$ species can be detected by the 9,10-diphenanthraquinone (DPA) in a time-dependent manner (Fig. 3f). In addition, adopting 5,5-dimethyl-1-pyrroline N-oxide (DMPO) and 2,2,6,6-tetramethylpiperidine (TEMP) as the specific spin trap reagents, respectively, the electron paramagnetic resonance (EPR) detection also confirms that the major ROS in HEA NPs is $\bullet O_2^-$ and $^1O_2$ (Fig. 3g and Supplementary Fig. 10). In-situ Fourier-transform infrared spectroscopy (FTIR) is utilized to investigate the intermediates' evolution during the POD-like reaction process, indicating the formation of *OOH intermediate and $\bullet O_2^-$ product (Fig. 3h, i). Together, these results demonstrate the superior ROS-generating capacity of PtFeCuCoNi HEA NPs, attributable to their optimized electronic structure and multi-element synergy.

Recognizing that persistent ROS after bacterial eradication may exacerbate oxidative damage and impair tissue regeneration, we evaluated the antioxidant capacity of PtFeCuCoNi HEA NPs at physiological pH (neutral). Using a nitrotetrazolium blue chloride (NBT) assay, the HEA NPs demonstrated significant ROS scavenging efficiency, achieving 90.42% $\bullet O_2^-$ elimination within 5 min, which significantly surpasses the performance of Pt NPs (73.48%) (Fig. 3j). Besides $\bullet O_2^-$ radicals, $H_2O_2$ is another important ROS molecule that contributes to intracellular oxidative stress. Thus, we examined the CAT-like activity of HEA NPs in neutralizing $H_2O_2$, a key mediator of oxidative stress. The PtFeCuCoNi HEA NPs demonstrated rapid $H_2O_2$ decomposition, achieving 85% scavenging within 30 min, significantly surpassing the performance of Pt NPs (Fig. 3k). Oxygen evolution measurements confirmed complete conversion of $H_2O_2$ to $H_2O$ and $O_2$ (Fig. 3l), with kinetic analysis revealing a 1.6-fold enhancement in $V_{max}$ ($V_{max} = 82.14\,\mu M\,s^{-1}$) compared to pure Pt ($51.53\,\mu M\,s^{-1}$) (Fig. 3m and supplementary Figs. 11 and 12). The exceptional TON (TON = 7.58 $s^{-1}$) further positions these HEA NPs among the most efficient metal-based ROS-scavenging systems reported (Fig. 3n and Supplementary Table 2).

The long-term activity and stability of PtFeCuCoNi HEA NPs as a CAT-like antioxidant have also been evaluated, revealing consistent

maintenance of high activity with no observable decline after five cycles of testing (Supplementary Fig. 13). It was found that the HEA synthesized with an equimolar precursor ratio exhibited the optimal enzyme-mimicking performance (Supplementary Fig. 14). This is attributed to the optimal $d$-band center and lattice structure of the equimolar PtFeCuCoNi HEA NPs, which strikes an optimal balance between the adsorption and desorption of reactants[39]. Importantly, PtFeCuCoNi HEA NPs exhibited pH-dependent ROS production and ROS scavenging, with a predominance of ROS production in acidic environments and a predominance of ROS scavenging in neutral conditions, confirming their microenvironmental adaptability (Supplementary Fig. 15).

The PtFeCuCoNi HEA NPs demonstrate significant pH-gated redox biocatalysis, preferentially generating bactericidal ROS in acidic microenvironments while switching to antioxidant functions at physiological pH. This dynamic behavior stems from the HEA's unique electronic configuration, which enables microenvironment-adaptive ROS biocatalysis. By simultaneously addressing pathogen clearance and oxidative stress resolution via pH-gated mechanisms, the HEA-based Janus artificial enzymes enable sequential therapy for infection control and tissue regeneration, thereby overcoming a fundamental challenge in chronic wound therapy.

To explore the structural advantages of PtFeCuCoNi HEA NPs, DFT calculations have been employed to reveal the ROS-catalytic reaction process. The optimized full geometry structure of the PtFe-CuCoNi HEA NPs is modeled based on the $fcc$ lattice, then a surface model is built with Pt: Fe: Cu: Co: Ni composition ratio close to the experimental ratio 19.1: 14.7: 20.9: 22.3: 23.0 by ICP-MS characterization (Fig. 4a). We evaluated the overall catalytic activity and catalytic sites of the PtFeCuCoNi HEA NPs using electron localization function (ELF) map (Fig. 4b) and the electrostatic potential (ESP) simulation calculations (Fig. 4c) before determining the enzymatic activity reaction path[40]. Notably, the Pt center in PtFeCuCoNi HEA NPs, with a higher charge density, plays a role in providing electrons in catalytic processes. Accordingly, the Fe and Co metal centers with lower ESP are obvious nucleophilic reaction sites, which have strong interactions with oxygen intermediates. Therefore, we have chosen the part circled in black in Fig. 4b as the main reactive site. The partial projected density of state (pDOS) for the HEA NPs is presented in Fig. 4d to illustrate their electronic structures. Distinct $d$-orbital overlaps among the different metals were clearly observed, demonstrating that the elements in the alloy were strongly bonded to each other. Notably, Fe and Co exhibit more unoccupied orbitals above the Fermi level, suggesting that Fe and Co atoms in PtFeCuCoNi are more susceptible to interaction with oxygen intermediates during the catalytic process, which is consistent with the calculated results of ESP. Bader charge analysis further corroborates this mechanism, showing substantial electron transfer from Fe (the highest positive charge) to Pt (the electron reservoir) (Fig. 4e and Supplementary Fig. 16). These electronic structure features enable efficient oxygen intermediate binding and activation at the Fe/Co sites while maintaining electron supply through Pt centers.

Analysis of spin-polarized $d$-orbitals reveals critical electronic structure modifications in the HEA NPs (Fig. 4f, g and Supplementary Fig. 17). According to the $d$-band center theory and Sabatier principle, a

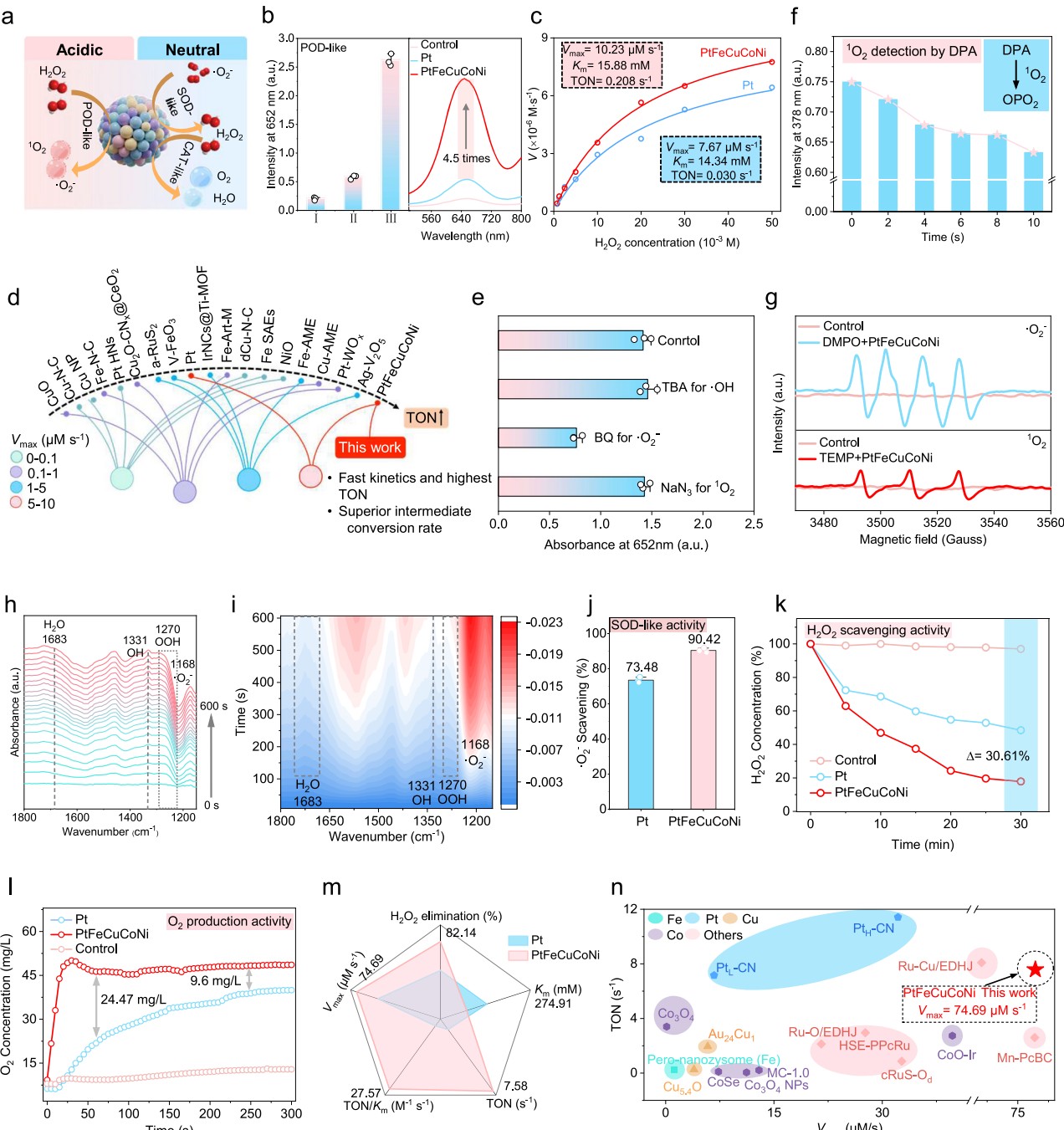

**Fig. 3 | Exploration of pH-gated ROS-catalytic activities of HEA-based Janus artificial enzymes. a** Schematic illustration of ROS-production activity for HEA NPs in an acid environment and ROS-scavenging activity for HEA NPs in a neutral environment (Created in BioRender. Tian Chen (2025) https://BioRender.com/hs7ivcs). **b** Quantitative analysis and UV-vis spectra of POD-like activity of Pt NPs and PtFeCuCoNi HEA NPs (I, II, III represent Control, Pt, and PtFeCuCoNi NPs groups, representatively). Data are presented as mean ± SD; $n = 3$ independent replicates. **c** Michaelis-Menten kinetic analysis for Pt NPs and PtFeCuCoNi HEA NPs in POD-like activity as $H_2O_2$ substrate. **d** Comparison and analysis of the TON and $V_{max}$ values with previously reported state-of-the-art POD-like biocatalysts. **e** •OH quenched by tert-butanol (TBA), •$O_2^-$ quenched by benzoquinone (BQ), $^1O_2$ quenched by sodium azide (NaN$_3$) in the catalytic oxidation process of TMB in HEA NPs

system. Data are presented as mean ± SD; $n = 3$ independent replicates. **f** The •$O_2^-$ production ability of HEA NPs was determined by a HE-specific probe. **g** EPR spectra for recording the •$O_2^-$ and $^1O_2$ signals. **h** In-situ FTIR spectrum and **i** the corresponding contour plot of PtFeCuCoNi HEA NPs for $H_2O_2$ decomposition. **j** SOD-like activity of Pt NPs and HEA NPs ($n = 3$ independent replicates, data are presented as mean values ± SD). **k** Time-dependent CAT-like performances via TiSO$_4$-based method with the presence of biocatalysts and $H_2O_2$. **l** The produced $O_2$ concentration was measured by an oxygen dissolving meter with the presence of biocatalysts and $H_2O_2$. **m** Statistics of $O_2$ production kinetic parameters of Pt NPs and HEA NPs. **n** Comparison and analysis of the TON and $V_{max}$ values with previously reported state-of-the-art CAT-like biocatalysts. In (**b**, **e**, **f**, **g**, **h**), a.u. indicates the arbitrary units. Source data are provided as a Source data file.

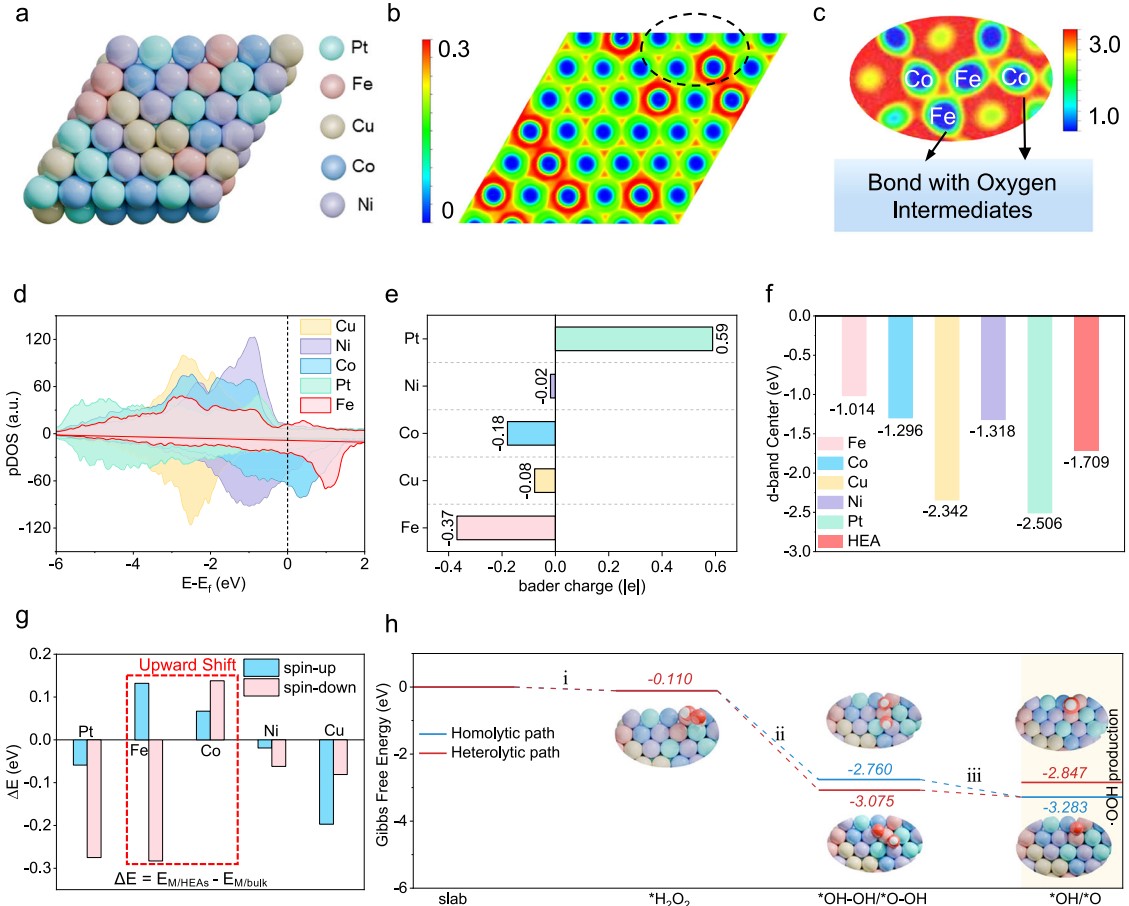

**Fig. 4 | DFT calculation of the electron distribution and structural configuration. a** 3D atomic model showing the crystal structure of the PtFeCuCoNi HEA NPs. **b** The calculated ELF diagrams of PtFeCuCoNi HEA NPs. **c** The calculated ESP of PtFeCuCoNi HEA NPs (ESP-mapped surface charge density with the isosurface of 0.01 e/bohr³) from the dashed black areas in (**b**). **d** pDOS of PtFeCuCoNi HEA NPs. **e** The amount of electron obtained from Bader charge analysis. **f** The d-band center for the individual elements in the PtFeCuCoNi HEA NPs. **g** The d-band center

comparisons for the individual elements and the bulk HEA NPs ($E_{M/HEAs}$: the d-band center of M in HEAs; $E_{M/bulk}$: the d-band center of M in pure bulk; M: Pt, Fe, Cu, Co, Ni). **h** The calculated $H_2O_2$ decomposition energy barrier on PtFeCuCoNi HEA NPs and the optimized structures of $*H_2O_2$ and intermediate adsorbed on PtFeCuCoNi HEA NPs. In (**d**), a.u. indicates the arbitrary units. Source data are provided as a Source data file.

rational up-shift of the d-band center position relative to the $E_f$ generally enhances the bond strengths between the metal atoms and oxygen intermediates[41], thus facilitating the POD-like activity. As shown in Fig. 4f, in PtFeCuCoNi HEA NPs, Fe and Co atoms exhibit the highest d-band centers among the constituent metals. Consequently, due to their favorable electronic structure for the adsorption of oxygen intermediates, Fe and Co atoms serve as the primary ROS-catalytic sites. In contrast, Pt possesses the lowest d-band center, which promotes desorption of oxygen intermediates, thereby rendering Pt atoms the primary desorption sites. An excessively elevated d-band center can hinder product desorption. Therefore, in PtFeCuCoNi HEA NPs, Cu and Ni atoms also play crucial roles, effectively modulating the d-band center and facilitating the production of ROS.

Subsequently, we compared the d-band centers of individual metals with those in the HEA NPs (Fig. 4g). We found that upon alloy formation, the d-band centers of Fe and Co exhibited a significant upward shift, which facilitated the adsorption of oxygen intermediates. Conversely, the d-band centers of Pt, Cu, and Ni shifted downward, indicating enhanced desorption of final products during the POD-like reaction, thereby preventing active-site poisoning and enhancing ROS-catalytic efficiency. In summary, the d-electron structure of the HEA NPs was regulated by different constituent metals, which guaranteed efficient and stable ROS-catalytic activity.

Take the POD-like ROS generation process as a representative example; Fig. 4h presents the key intermediate structures and corresponding Gibbs free energy profiles. The catalytic pathway initiates with $H_2O_2$ adsorption at Fe sites (step i), followed by either homolytic or heterolytic cleavage (step ii). DFT calculations reveal favorable adsorption energetics for both dissociation pathways, with binding energies of −3.075 eV for $*OH + *O$ and −2.760 eV for $2*OH$, indicating that $2*OH$ and $*OH + *O$ were adsorbed stably (Fig. 4f), which confirms the spontaneous stabilization of these species. The Fe sites facilitate this process, enabling strong interfacial interactions between the HEA surface and adsorbed $*OH/*O$ radicals, as evidenced by pronounced orbital hybridization in pDOS analysis and substantial electron transfer (0.89 |e| to $*OH$; 0.56 |e| to $*O$) in Bader charge calculations (Supplementary Figs. 18 and 19). Notably, the homolytic pathway demonstrates particular thermodynamic preference, requiring only a 0.174 eV energy barrier for •OOH formation from $*OH + *O$ intermediates (step iii). This low activation energy highlights the kinetic advantage of homolytic $H_2O_2$ dissociation on the HEA surface. The catalytic proficiency arises from the collective effects of multiple transition metals. While Fe dominates the initial activation, the surrounding metal atoms provide essential coordination environments and electron donation capacity that collectively enhance the intrinsic POD-like activity. Pt primarily functions as an electron reservoir, and Cu/Ni atoms play a

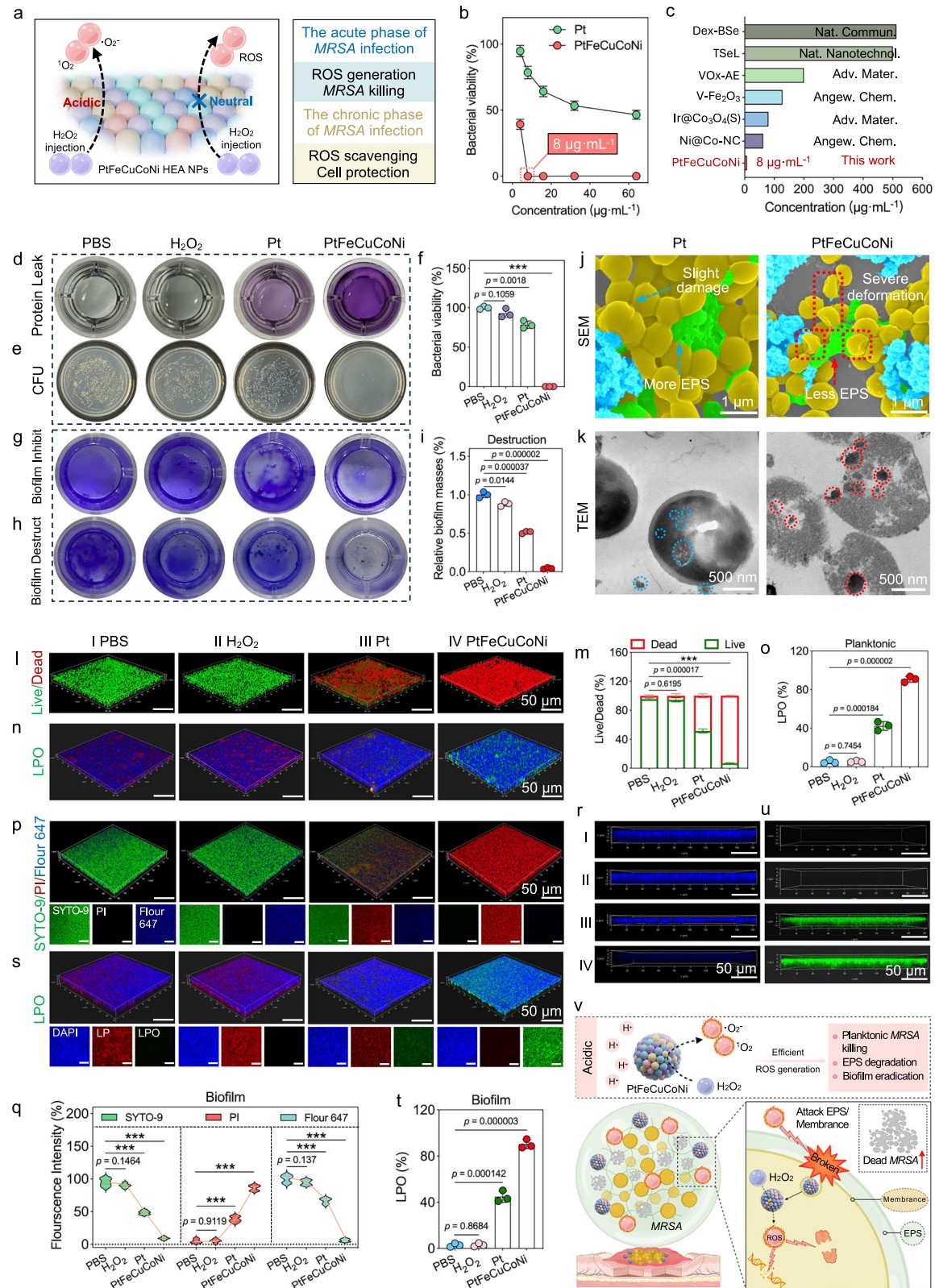

key role in modulating the *d*-band center. These computational insights comprehensively characterize the electronic structure-catalytic function relationship in HEA NPs, demonstrating how elemental synergy enables their exceptional biomimetic performance.

Furthermore, according to our previous studies, the structure of adsorbed O* is a key intermediate in compartmentalizing the POD-like and CAT-like pathways[42]. Therefore, to demonstrate the continuous

pH-switchable catalytic activity of PtFeCuCoNi HEA NPs, we complemented the DFT studies of the free energy of the structure of PtFeCuCoNi HEA NPs adsorbing O* (Structure 1) for the adsorption of $H^+$ or $H_2O_2$ under acidic and neutral conditions. As shown in Supplementary Fig. 20, the free energies of structure 1 for the adsorption of both $H^+$ or $H_2O_2$ are negative, indicating that both processes can occur spontaneously. However, compared with adsorbing $H_2O_2$ (−0.384 eV),

**Fig. 5 | Evaluation of in vitro bactericidal actions on *MRSA*. a** POD-like ROS production of PtFeCuCoNi HEA NPs in acidic conditions, while no ROS generation occurs in neutral environments (Created in BioRender. Tian Chen (2025) https://BioRender.com/hs7ivcs). **b** Viability of *MRSA* treated with different concentrations of Pt and HEA NPs ($n = 3$ independent replicates). **c** Comparison of antibacterial concentrations of recently reported ROS-generating materials. **d** Protein leakage photographs. **e** Representative *MRSA* colonies and **f** antibacterial statistics, ***$p_{(PtFeCuCoNi)}$ = 0.000000078. **g**, **h** Crystal violet-stained biofilm inhibition and destruction ($n = 3$ independent replicates). **i** Quantified biofilm biomass ($n = 3$ independent replicates). Representative SEM (**j**) and TEM (**k**) images of bacteria cultured with Pt and HEA NPs. Pt group: minor surface damage, abundant EPS (blue arrows), and limited NP entry (blue circles). HEA NPs group: marked structural disruption (red arrows) and visible internalization of HEA NPs (pink circles). 3D CLSM images (**l**) and ratio (**m**) of Live/Dead ratios of planktonic *MRSA*, Live: ***$p_{(PtFeCuCoNi)}$ = 0.000000061 ($n = 3$ independent replicates). **n** C11-BODIPY$^{581/591}$ stained confocal microscopy images of different materials co-interacting with planktonic *MRSA*. **o** LPO statistics, $p_{(PtFeCuCoNi)}$ = 0.000001807, $p_{(Pt)}$ = 0.000183607 ($n = 3$ independent replicates). In (**p**, **r**), Alexa Flour 647 stained confocal images of *MRSA* biofilms. **q** Statistics of SYTO-9/PI/Alexa Flour 647 intensity ($n = 5$ independent replicates), PI: propidium iodide, SYTO-9: ***$p_{(PtFeCuCoNi)}$ = 0.000000002, ***$p_{(Pt)}$ = 0.000000303; PI: ***$p_{(PtFeCuCoNi)}$ = 0.0000000005, ***$p_{(Pt)}$ = 0.000000794; Alexa Flour 647: ***$p_{(PtFeCuCoNi)}$ = 0.0000000007, ***$p_{(Pt)}$ = 0.000007075. In (**s**, **u**), C11-BODIPY$^{581/591}$ stained confocal images of *MRSA* biofilms. **t** LPO statistics ($n = 3$ independent replicates). **v** Schematic illustration of pH-dependent ROS-mediated antibacterial action of HEA NPs (Created in BioRender. Chen, T. (2025) https://BioRender.com/hs7ivcs). Data are presented as mean ± SD, ***$p < 0.001$, statistical significance was calculated using one-way ANOVA followed by Tukey's post-hoc test, all tests were two-sided. Experiments were repeated independently (**d**, **e**, **g**, **h**, **j**, **k**, **l**, **n**, **p**, **r**, **s**, **u**) at least three times. Source data are provided as a Source data file.

the free energy of adsorbing $H^+$ by structure 1 is lower (−0.436 eV), indicating that structure 1 will preferentially combine with $H^+$ to form *OH to complete the POD-like activity pathway in an acidic environment with the presence of $H^+$. Furthermore, under neutral or slightly alkaline conditions, structure 1 is more prone to adsorb $H_2O_2$ to carry out the CAT-like activity pathway to realize the continuous pH-switchable catalytic activity.

## Acid-triggered ROS catalysis to eradicate *MRSA* and disrupt biofilms

Having established the robust pH-responsive ROS generation of PtFeCuCoNi HEA NPs, we next evaluated their antimicrobial performance against *MRSA*, a clinically relevant, extensively drug-resistant pathogen implicated in skin and soft-tissue infections[43–45]. In acidic media containing $H_2O_2$, the NPs predominantly produced $•O_2^-$ and $^1O_2$, whereas at near-neutral pH they exhibited no detectable POD-like activity (Fig. 5a). To directly verify ROS generation in bacteria, intracellular ROS levels were assessed using 2′,7′-dichlorodihydrofluorescein diacetate (DCFH-DA) staining. Bacteria treated with PtFeCuCoNi HEA NPs exhibited strong green fluorescence, indicative of pronounced intracellular ROS accumulation and presenting direct evidence for the ROS-mediated antibacterial mechanism of the artificial enzyme (Supplementary Fig. 21). Notably, the particles achieved a minimum inhibitory concentration (MIC) and minimum bactericidal concentration (MBC) of 8 µg/mL (Fig. 5b and Supplementary Fig. 22), a dosage substantially lower than those typically reported for comparable ROS-catalytic antimicrobial systems (generally >100 µg/mL) (Fig. 5c). Thus, HEA NPs are capable of maintaining potent bactericidal activity against drug-resistant pathogens even under physiologically challenging microenvironmental conditions[46–49].

To delineate the cellular basis of the antibacterial activity, we first quantified ROS-augmented membrane disruption in planktonic *MRSA*. A protein leakage assay, which measures cytoplasmic release from damaged cells[50], revealed markedly increased permeability and leakage in HEA-treated bacteria (Fig. 5d and Supplementary Fig. 23), consistent with envelope disruption. In line with this, colony-forming unit (CFU) analysis showed almost complete eradication of *MRSA* when HEA NPs were combined with $H_2O_2$[51–54], whereas $H_2O_2$ alone had a negligible effect. Under identical oxidative conditions, monometallic Pt NPs reduced viable counts by only 21.49% (Fig. 5e, f and Supplementary Fig. 24), underscoring the pronounced synergistic antibacterial effect of the HEA formulation. Quantitative crystal-violet assays further demonstrated an approximate 90% reduction in biofilm biomass following HEA NPs treatment (Fig. 5g–i and Supplementary Fig. 25)[55,56].

Scanning electron microscopy (SEM) also revealed minor surface damage with abundant extracellular polymeric substances (EPS) in Pt-treated bacteria[57,58], whereas HEA-exposed group displayed marked membrane roughening, localized collapse, and visibly reduced EPS coverage (Fig. 5j and Supplementary Fig. 26). Consistently, Dextran Alexa Fluor 647 staining showed markedly decreased fluorescence intensity of polysaccharides within the EPS after HEA treatment, confirming substantial matrix degradation (Supplementary Figs. 34 and 35). Furthermore, sodium dodecyl sulfate polyacrylamide gel electrophoresis (SDS-PAGE) and agarose gel electrophoresis of EPS proteins and total DNA (including extracellular DNA) demonstrated extensive degradation following PtFeCuCoNi treatment, whereas Pt treatment induced only partial degradation, and controls remained largely intact (Supplementary Figs. 27 and 28), corroborating the SEM observations. TEM images revealed highly fragmented membranes and clear internalization of HEA NPs following incubation with $H_2O_2$ (Fig. 5k), indicative of a ROS-potentiated, membrane-disruptive bactericidal process.

Afterward, we have rigorously evaluated the solution stability of HEA NPs. The PtFeCuCoNi HEA NPs exhibit exceptional dispersibility, with no detectable precipitation or phase separation following static incubation at room temperature for 12 days (Supplementary Fig. 29). Moreover, assessments of particle size and zeta potential of PtFeCuCoNi HEA NPs across various intervals (days 3, 6, 9, and 12) indicate no significant dimensional alterations or aggregation (Supplementary Fig. 30). Therefore, PtFeCuCoNi HEA NPs exhibit robust stability and hold promise for enhanced therapeutic efficacy in forthcoming in vivo and in vitro studies.

To visualize bacterial viability at high spatial resolution, we acquired three-dimensional reconstructions by fluorescence microscopy and confocal laser scanning microscopy (CLSM)[59–62]. In the PtFeCuCoNi HEA NPs-treated group, bacterial clusters showed dominant red fluorescence, indicating extensive membrane compromise and effective killing of *MRSA* (Fig. 5l, m and Supplementary Fig. 31). In contrast, monometallic Pt NPs control retained strong green fluorescence signals, reflecting a high proportion of viable bacteria under comparable conditions. To determine whether ROS-induced lipid damage contributed to this effect, we assessed membrane lipid peroxidation (LPO) using the ratiometric probe C11-BODIPY$^{581/591}$. A pronounced shift toward green fluorescence was observed by CLSM in the HEA NPs-treated group, corresponding to elevated lipid peroxide accumulation and confirming extensive oxidative degradation of bacterial membrane lipids (Fig. 5n, o and Supplementary Figs. 32 and 33).

Accordingly, we then evaluated the biofilm eradication efficacy[62,63]. Live/Dead fluorescence imaging revealed a substantial loss of viability and pronounced architectural degradation in HEA NPs-treated biofilms, accompanied by a significant decrease in matrix thickness (Fig. 5p, q and Supplementary Figs. 34 and 35). Conversely, Pt NPs-treated biofilms largely preserved a compact, densely packed morphology with minimal thickness change (Fig. 5r). Consistently,

CLSM-based LPO analysis showed a strong green fluorescence shift in HEA NPs-exposed biofilms[64,65], particularly within deeper layers, suggesting extensive phospholipid oxidation and ROS-induced damage throughout the biofilm matrix (Fig. 5s–u and Supplementary Figs. 36 and 37)[66–68]. In line with this, malondialdehyde (MDA) quantification further confirmed enhanced lipid peroxidation in *MRSA* treated with PtFeCuCoNi HEA NPs compared with Pt NPs and controls (Supplementary Fig. 38). To place these findings in context, we propose a conceptual model in which PtFeCuCoNi HEA NPs facilitate oxidative degradation of biofilm EPS and bacterial membranes[69,70], followed by membrane penetration and intracellular accumulation, thereby exerting multifaceted bactericidal effects (Fig. 5v).

To elucidate how HEA-catalyzed ROS reshape *MRSA* biology under acidic conditions, we performed transcriptomic sequencing on *MRSA* exposed to $H_2O_2$ alone (as a control) or $H_2O_2$ with PtFeCuCoNi HEA NPs. A total of 401 differentially expressed genes (DEGs) were identified, including 226 upregulated and 175 downregulated genes (Fig. 6a). Hierarchical clustering of global expression profiles revealed a clear distinction between treated and control groups (Fig. 6b), indicating a treatment-induced transcriptional shift. Gene Ontology (GO) enrichment analysis comparing HEA + $H_2O_2$ with $H_2O_2$ alone highlighted terms related to envelope and nucleoprotein architecture in the Cellular Component (CC) category[11], including "cell wall", "external encapsulating structure", and "protein-DNA complex" (Fig. 6c). Together with the downregulation of Biological Process (BP) terms such as "response to oxygen-containing compound" and "response to starvation", these enrichments are consistent with ROS-induced structural damage accompanied by a starvation-like stress program.

Furthermore, Kyoto Encyclopedia of Genes and Genomes (KEGG) analysis revealed broad perturbation of metabolic networks, with repression of sulfur metabolism[71], cysteine[72] and methionine metabolism[73], histidine metabolism[74], and glycerophospholipid metabolism[73], indicating disruption of thiol-based antioxidant defenses and membrane-lipid homeostasis (Fig. 6d). Notably, gene set enrichment analysis (GSEA) further showed positive enrichment of "oxidative phosphorylation" and "ATP-dependent activity"[75], suggesting an early compensatory surge in energy metabolism, likely reflecting transient activation of bacterial respiratory and ATP-generating pathways in response to acute ROS stress[42,76]. In contrast, "ATP-binding cassette (ABC) transporters"[77] and "histidine metabolism"[78] were negatively enriched (Fig. 6e), indicating impaired substrate trafficking and amino-acid biosynthesis[12]. Collectively, these transcriptomic data support a model in which HEA-driven ROS inflict envelope and macromolecular damage, trigger maladaptive energy overactivation, and erode transport and biosynthetic capacity, culminating in a dual hit of oxidative injury and metabolic exhaustion that underlies efficient bacterial killing (Fig. 6f).

## Neutral-pH ROS scavenging preserves cellular function and reprograms macrophages

Given its demonstrated antioxidase-like activity[79–81], we evaluated PtFeCuCoNi HEA NPs for the mitigation of oxidative stress and restoration of cellular homeostasis at physiological pH (7.4)[41,82]. Cytocompatibility testing showed no appreciable toxicity of PtFeCuCoNi HEA NPs or Pt NPs to human umbilical vein endothelial cells (HUVEC), rat skin fibroblasts (RSF), or murine macrophages (Raw264.7) up to 4 $\mu g\,mL^{-1}$ (Supplementary Figs. 39–41); therefore, 2 $\mu g\,mL^{-1}$ was used in subsequent assays. Upon $H_2O_2$ challenge, intracellular ROS rose sharply by DCFH-DA readout, whereas HEA NP pretreatment reduced fluorescence to near-baseline across all three cell types (Fig. 7a and Supplementary Fig. 42), with quantification confirming significant suppression of ROS (Fig. 7b). In agreement, Live/Dead staining showed marked protection from $H_2O_2$-induced cytotoxicity in the HEA NPs group compared with untreated or Pt NPs-treated cells (Fig. 7c, d). To determine whether this protection extends

to cell structure and function, we examined F-actin organization[83]. Phalloidin staining revealed that $H_2O_2$ exposure led to pronounced cytoskeletal collapse and cell rounding, whereas HEA NPs preserved cortical actin and cell spreading comparable to unstressed controls (Fig. 7e–g and Supplementary Fig. 43). Functionally, we found that HEA NPs significantly accelerated scratch-wound closure by both endothelial cells and fibroblasts under oxidative challenge (Supplementary Figs. 44 and 45). Likewise, in tube-formation assays, HEA NPs-treated HUVEC formed denser, more extensive capillary-like networks than $H_2O_2$-exposed controls, indicating that cytoskeletal preservation underpins enhanced migratory and angiogenic capacity (Fig. 7h, i).

Beyond cytoprotection, effective resolution of the pro-inflammatory milieu is essential for recovery from bacterial infections, a process critically governed by macrophage plasticity[84,85]. To assess this, we examined macrophage polarization under lipopolysaccharide (LPS) stimulation. Of note, HEA NPs pretreatment significantly reduced C-C chemokine receptor type 7 (CCR7) expression, a signature marker of M1 macrophages, relative to both LPS alone and Pt treatment (Fig. 7j, k and Supplementary Fig. 46). In parallel, CD163 expression was strongly upregulated (Fig. 7l, m), suggesting a phenotypic shift from a pro-inflammatory M1 phenotype toward a reparative M2 state. Furthermore, RT-qPCR analysis revealed that, compared to Pt NPs, PtFeCuCoNi HEA NPs exerted a stronger inhibitory effect on the mRNA expression of pro-inflammatory cytokines, such as inducible nitric oxide synthase (iNOS), interleukin-6 (IL-6), and tumor necrosis factor-alpha (TNF-α), while concurrently enhancing genes associated with anti-inflammatory activity, including arginase-1 (Arg-1), interleukin-10 (IL-10), and transforming growth factor-beta (TGF-β) (Supplementary Figs. 47 and 48). Together, these results demonstrate that, at neutral pH, PtFeCuCoNi HEA NPs extinguish intracellular ROS, preserve cytoskeletal architecture, sustain angiogenic function, and reprogram macrophages toward tissue repair, thus maintaining redox homeostasis and protecting cells from oxidative injury (Fig. 7n).

To delineate how PtFeCuCoNi HEA NPs restore cellular redox homeostasis at neutral pH, we performed RNA sequencing (RNA-seq) on cells treated with PBS, $H_2O_2$, or $H_2O_2$ + HEA NPs. Principal component analysis (PCA) separated the three groups, with the co-treatment clustering closer to control than to $H_2O_2$ (Fig. 8a), consistent with a partial reversal of oxidative stress-induced transcriptional alterations. Differential expression analysis identified significant upregulation of protective genes, including *PFKFB3*[86] (glycolysis) and *THBS1*[87] (extracellular matrix remodeling), together with downregulation of stress-associated genes such as *SLC7A11*[65], *PSAT1*[88], *CHAC1*, *MTHFD2*, and *SESN2* (Fig. 8b), suggesting a transcriptional switch from injury signaling to programs supporting metabolic recovery and structural repair. To validate these transcriptomic trends, we further performed RT-qPCR analysis of representative genes involved in glycolytic activation and oxidative stress response. Consistent with the RNA-seq results, PtFeCuCoNi HEA NPs treatment markedly upregulated *PFKFB3*, *THBS1*, and *HSPA8*, while significantly downregulating *CHAC1*, *ASNA*, and *PSAT1* compared with both $H_2O_2$ and Pt NPs groups (Supplementary Fig. 49). These results confirm that HEA NPs transcriptionally activate glycolysis- and repair-related programs while suppressing stress-associated pathways, thereby supporting the restoration of redox homeostasis and cellular recovery. Hierarchical clustering corroborated a shift toward a homeostatic expression profile after HEA NPs treatment (Fig. 8c).

Furthermore, GO enrichment analysis highlighted the upregulation of terms linked to cell proliferation and angiogenesis, as well as dioxygenase activity (Fig. 8d), suggesting engagement of pro-repair programs. GSEA showed positive enrichment trends for tube development and cytoskeleton organization, alongside significant negative enrichment of apoptosis signaling and a trend toward reduced cellular response to oxygen-containing compounds (Fig. 8e and

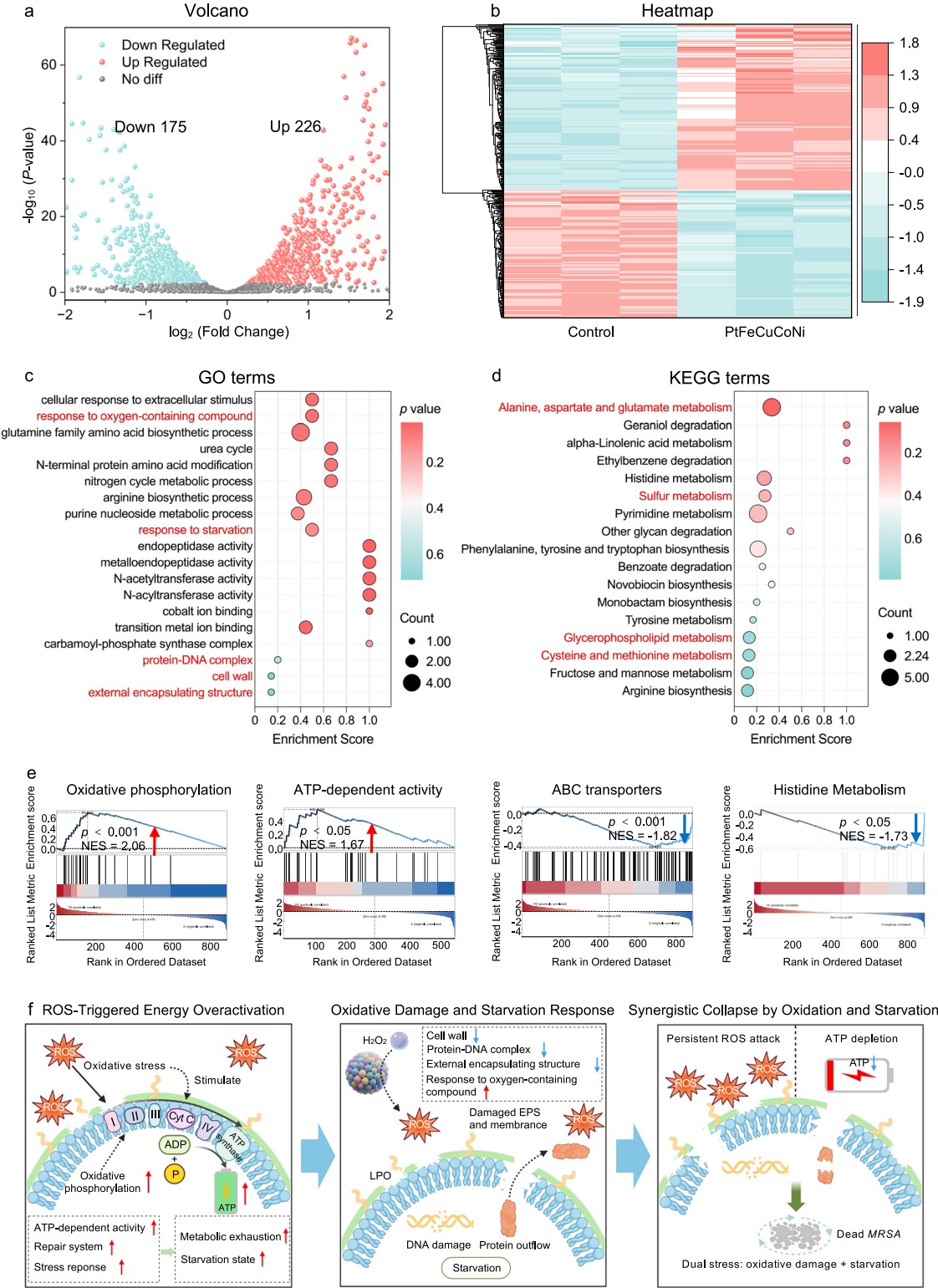

**Fig. 6 | Transcriptomic analysis of the *MRSA* killing mechanism. a** Volcano plot of DEGs (gray: not significantly different genes; red: upregulated genes; blue: downregulated genes) in the PtFeCuCoNi HEA NPs and PBS (Control) treated groups. **b** Heat map of DEGs involved in bacterial metabolism pathway (red represents genes with relatively high expression levels, blue represents genes with relatively low expression levels). **c** GO annotation analysis of DEGs in *MRSA* treated with HEA NPs. **d** KEGG enrichment analysis scatter plot. **e** GSEA analysis of PBS (Control) versus HEA NPs. **f** Schematic diagram of the antimicrobial mechanism of

HEA NPs (Created in BioRender. Tian Chen (2025) https://BioRender.com/hs7ivcs). In (**a**), *p*-value was obtained from two-sided DESeq2 test without multiple comparison. In (**c**, **d**), *p*-value was obtained from a one-sided Hypergeometric test without multiple comparisons. In (**e**), *p*-value was obtained from a one-sided Permutation test without multiple comparisons. In (**e**), NES indicates the normalized enrichment score. Experiments were repeated independently three times with similar results. Source data are provided as a Source data file.

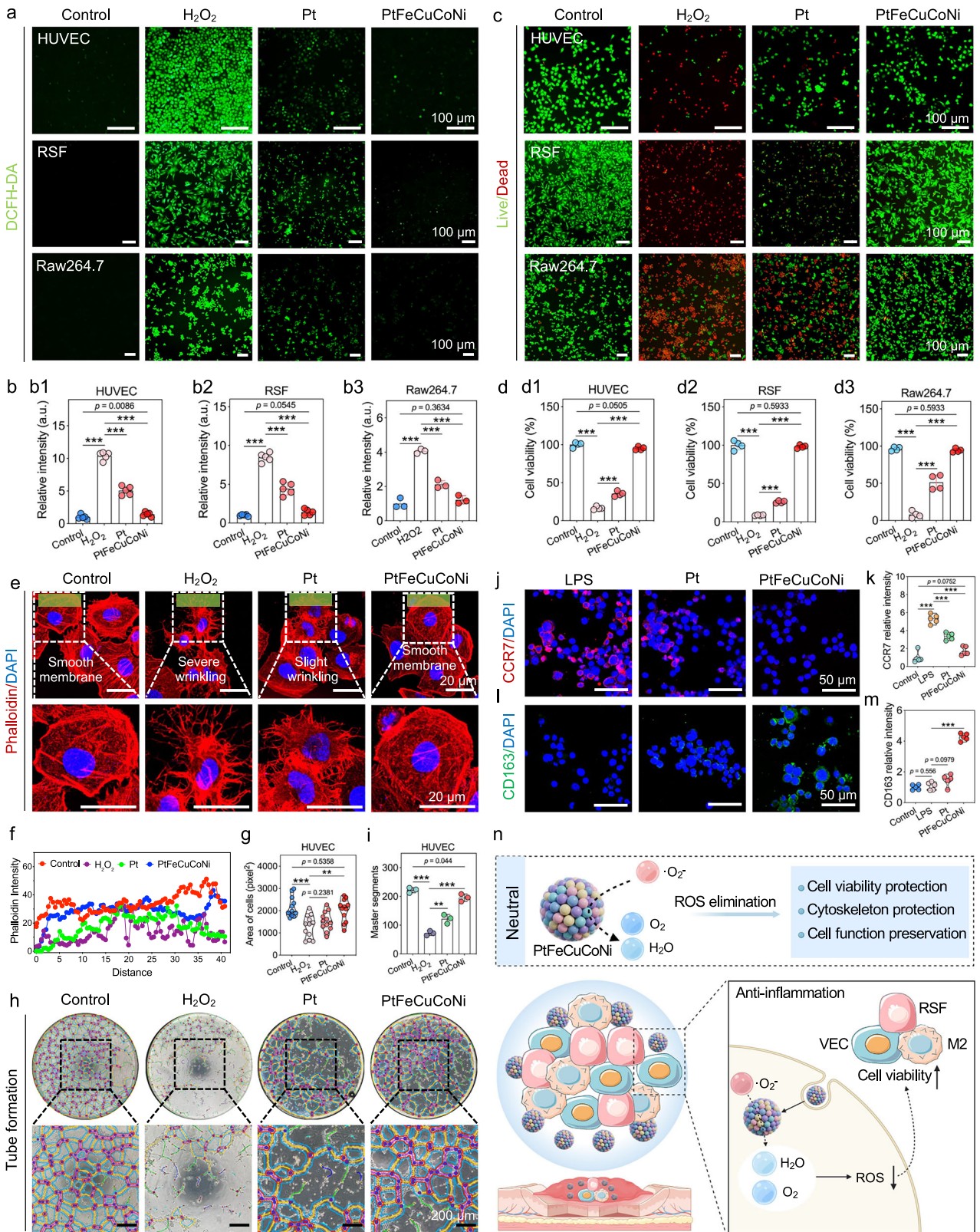

Supplementary Fig. 50). KEGG pathway analysis further revealed broad attenuation of inflammatory and death-related cascades, including TNF[87], IL-17[89], NF-kB[86], p53[88], and MAPK[90,91] signaling as well as the ferroptosis and necroptosis pathways, which were all diminished in the HEA NP group relative to $H_2O_2$ alone (Fig. 8f). Together, these data support a model in which ROS scavenging by PtFeCuCoNi shifts the transcriptome from injury and inflammatory signaling toward

pathways associated with cytoskeletal maintenance, cell proliferation, and vascular development, thereby facilitating the transition from damage to repair (Fig. 8g).

To evaluate pH-gated redox biocatalysis of PtFeCuCoNi HEA NPs in vivo[92–94], a murine full-thickness wound model was inoculated with *MRSA*, with intradermal vancomycin serving as a positive control (Fig. 9a)[95]. Treatments were administered in two phases to reflect the

**Fig. 7 | In vitro ROS clearance and cell protection by HEA NPs-based Janus artificial enzymes. a** Representative images showing ROS elimination by different biocatalysts. **b** (b1–b3) Quantification of DCFH-DA fluorescence in **a**, (b1): ***$p_{(PtFeCuCoNi)}$ = 0.000000002, ***$p_{(Pt)}$ = 0.000001097, ***$p_{(H2O2)}$ = 0.000000002; (b2): ***$p_{(PtFeCuCoNi)}$ = 0.000000016, ***$p_{(Pt)}$ = 0.000016077, ***$p_{(H2O2)}$ = 0.000000003 (b1, b2: $n$ = 5 independent replicates); (b3): ***$p_{(PtFeCuCoNi)}$ = 0.0000056301, ***$p_{(Pt)}$ = 0.000209558, ***$p_{(H2O2)}$ = 0.000077599 ($n$ = 3 independent replicates). **c** Representative Live/Dead fluorescence images of different cell types under ROS stimulation and various treatments. **d** (d1–d3) Quantification of cell viability in **c** ($n$ = 4 independent replicates), (d1): ***$p_{(PtFeCuCoNi)}$ = 0.000000003, ***$p_{(Pt)}$ = 0.000067323, ***$p_{(H2O2)}$ = 0.000000011; (d2): ***$p_{(PtFeCuCoNi)}$ = 0.0000000002, ***$p_{(Pt)}$ = 0.000000668, ***$p_{(H2O2)}$ = 0.000000071; (d3): ***$p_{(PtFeCuCoNi)}$ = 0.000000043, ***$p_{(Pt)}$ = 0.000211491, ***$p_{(H2O2)}$ = 0.00000004. **e** Fluorescence images of paxillin (Red: paxillin, Blue: DAPI). **f** Fluorescence intensity distribution of cytoskeleton staining. **g** Quantification of the area of cells ($n$ = 12 independent replicates).

**$p_{(PtFeCuCoNi)}$ = 0.002, ***$p_{(H2O2)}$ = 0.000242567. **h** Vessel-forming experiments with HUVEC after different interventions. **i** Quantification of master segments ($n$ = 3 independent replicates). ***$p_{(PtFeCuCoNi)}$ = 0.0002, **$p_{(Pt)}$ = 0.008049255, ***$p_{(H2O2)}$ = 0.000028428. **j** CCR7 (red) and DAPI (blue) immunofluorescence images of Raw264.7 cells after 24 h of LPS stimulation. **l** CD163 (red) and DAPI (blue) immunofluorescence images of Raw264.7 cells after 24 h of LPS stimulation. Relative semiquantitative analysis of the mean fluorescence intensity of CCR7 (**k**) and CD163 (**m**), **k**: ***$p_{(PtFeCuCoNi)}$ = 0.000003387, ***$p_{(Pt)}$ = 0.000132712, ***$p_{(LPS)}$ = 0.000002242; **m**: ***$p_{(PtFeCuCoNi)}$ = 0.000000032, **k, m**: $n$ = 5 independent replicates. **n** Schematic of cell responses to high-concentration $H_2O_2$ and HEA NPs' protection (Created in BioRender. Tian Chen (2025) https://BioRender.com/hs7ivcs). Data are presented as mean ± SD, **$p$ < 0.01, ***$p$ < 0.001, statistical significance was calculated using one-way ANOVA with Tukey's post-hoc test, two-sided. Experiments (**a, c, e, j, l, h**) were repeated independently at least three times with similar results. In (**b**), a.u. indicates the arbitrary units. Source data are provided as a Source data file.

evolving niche: an initial acid-activated bactericidal window (days 1–3) and a subsequent neutral-pH regenerative phase (days 4–12)[96]. Digital imaging and planimetric analysis revealed that PBS and $H_2O_2$ controls remained inflamed and largely unclosed through day 6, whereas both vancomycin and HEA NPs achieved minimal swelling and near-complete closure by day 12 (Fig. 9b–e and Supplementary Fig. 51)[97,98]. Consistently, CFU assays on day 3 showed extensive bacterial growth in PBS and $H_2O_2$ groups. In contrast, vancomycin and HEA NP treatment yielded few or no colonies, confirming potent antimicrobial activity during the acidic phase (Fig. 9f, g)[99,100].

Reflecting the transition from infection control to tissue repair, hematoxylin and eosin (H&E) staining on day 12 revealed persistent neutrophil infiltration and necrosis in controls[101], whereas vancomycin- and HEA-treated wounds displayed well-organized epidermal regeneration with minimal inflammation (Fig. 9h). Masson's trichrome staining showed dense, uniformly aligned collagen only in treated wounds[102], in contrast to the sparse, disordered matrices in controls (Fig. 9i–k). Angiogenesis, assessed by α-smooth muscle actin (α-SMA)[18], CD31[103,104], and vascular endothelial growth factor (VEGF)[105] immunostaining, was most pronounced in the HEA group, which exhibited the highest densities of α-SMA⁺ vessels and CD31⁺ capillaries together with the strongest VEGF signal (Fig. 9l–o and Supplementary Figs. 52 and 53). Interestingly, the HEA NPs treatment group displayed even better pro-angiogenic activity than that seen in vancomycin-treated wounds, which only offer antibacterial effect, suggesting that PtFeCuCoNi HEA NPs not only eradicate pathogens but also actively stimulate vascular formation, potentially associated with reduced levels of inflammatory factors such as interleukin-1beta (IL-1$\beta$) (Supplementary Fig. 54).

Moreover, peripheral blood analysis on day 12 revealed that neutrophil and white blood cell counts in the HEA NPs-treated rats were comparable to those of healthy controls, indicating attenuation of systemic inflammation. In addition, lymphocyte levels were higher and close to normal, suggesting recovery of immune homeostasis (Supplementary Fig. 55). Furthermore, macrophage dynamics at the wound edge were examined by immunofluorescence on days 3 and 7. Early after infection, HEA-treated wounds recruited significantly more F4/80⁺ iNOS⁺ M1-like macrophages than controls, supporting rapid bacterial clearance (Fig. 9p–s and Supplementary Figs. 56 and 57). By day 7, M1 markers decline while F4/80⁺ CD206⁺ M2-like macrophages increased, indicating a timely transition to a reparative phenotype that was not observed in untreated lesions[106–109]. Major-organ histology, hemolysis test, and serum biochemistry showed no adverse effects following HEA NPs administration (Supplementary Figs. 58–60). Taken together, these findings demonstrate that HEA NPs execute a programmed, microenvironment-adaptive therapy: they enable rapid *MRSA* eradication in the acidic phase and subsequently promote inflammation resolution, angiogenesis, and matrix remodeling as pH

normalizes, thereby accelerating wound repair in an antibiotic-independent manner.

## Discussion

Infections caused by antibiotic-resistant pathogens represent an escalating clinical challenge due to antibiotic treatment failure. They can lead to hard-to-heal wound infections[110], urinary tract infection[111], pulmonary infections, intra-abdominal infections[112], and infective endocarditis[113]. Compounded by complex inflammation and dynamic microenvironmental alterations, these factors greatly complicate clinical management. Various high-entropy alloys have been explored for antibacterial applications. For example, MnFeCoNiCu HEA showed antibacterial activity against *S. enteritidis* and *E. coli* at 40 μg/mL[114]; the $Al_{0.6}CoCrCu_{0.1}FeNiSi_{0.2}$ alloy, designed as a low-copper HEA, achieved complete E. coli elimination by day 3[115]; Zhou et al. reported a Cu-Ag HEA with strong antibacterial effects, though the high cost of Ag may limit its broader application[116]. In summary, most previously reported HEAs exhibit single-mode antibacterial activity and lack the multifunctional or biocompatible properties required for effective biomedical use. Inspired by the precise redox modulation capabilities of natural peroxisomes, we herein engineered PtFeCuCoNi HEA-based Janus artificial enzymes with pH-gated redox biocatalysis for sequential therapy of drug-resistant bacteria and inflammatory wounds. These Janus artificial enzymes dynamically transition between OXD-/POD-like functions in acidic infectious environments to CAT-/SOD-like activities under neutral inflammatory conditions, effectively balancing rapid microbial clearance and timely inflammation resolution. Comprehensive in vitro and in vivo evaluations demonstrated rapid eradication of *MRSA* and efficient disruption of biofilms, coupled with robust protection and regeneration of host tissues, highlighting a dual-phase therapeutic paradigm for infected wound management.

Compared to existing antimicrobial and anti-inflammatory platforms, the designed PtFeCuCoNi HEA-based Janus artificial enzymes offer several distinct and significant advantages. First, their multimetallic configuration leverages the high configurational entropy and pronounced lattice distortions inherent to HEAs, yielding abundant, versatile catalytically active sites with finely tuned electronic structures near the Fermi level. This optimized electronic structure significantly improves electron transfer kinetics and oxygen intermediate binding dynamics, achieving high catalytic efficiencies unattainable with traditional monometallic or simple alloy-based ROS catalysts. Second, owing to their superior catalytic efficiency, these Janus artificial enzymes exert potent bactericidal effects at exceptionally low therapeutic doses (as low as 8 μg mL⁻¹), thus substantially minimizing potential cytotoxicity and off-target oxidative damage, and markedly enhancing their safety profile. Extensive systemic biosafety evaluations further validate their biocompatibility, underscoring their potential for safe clinical application.

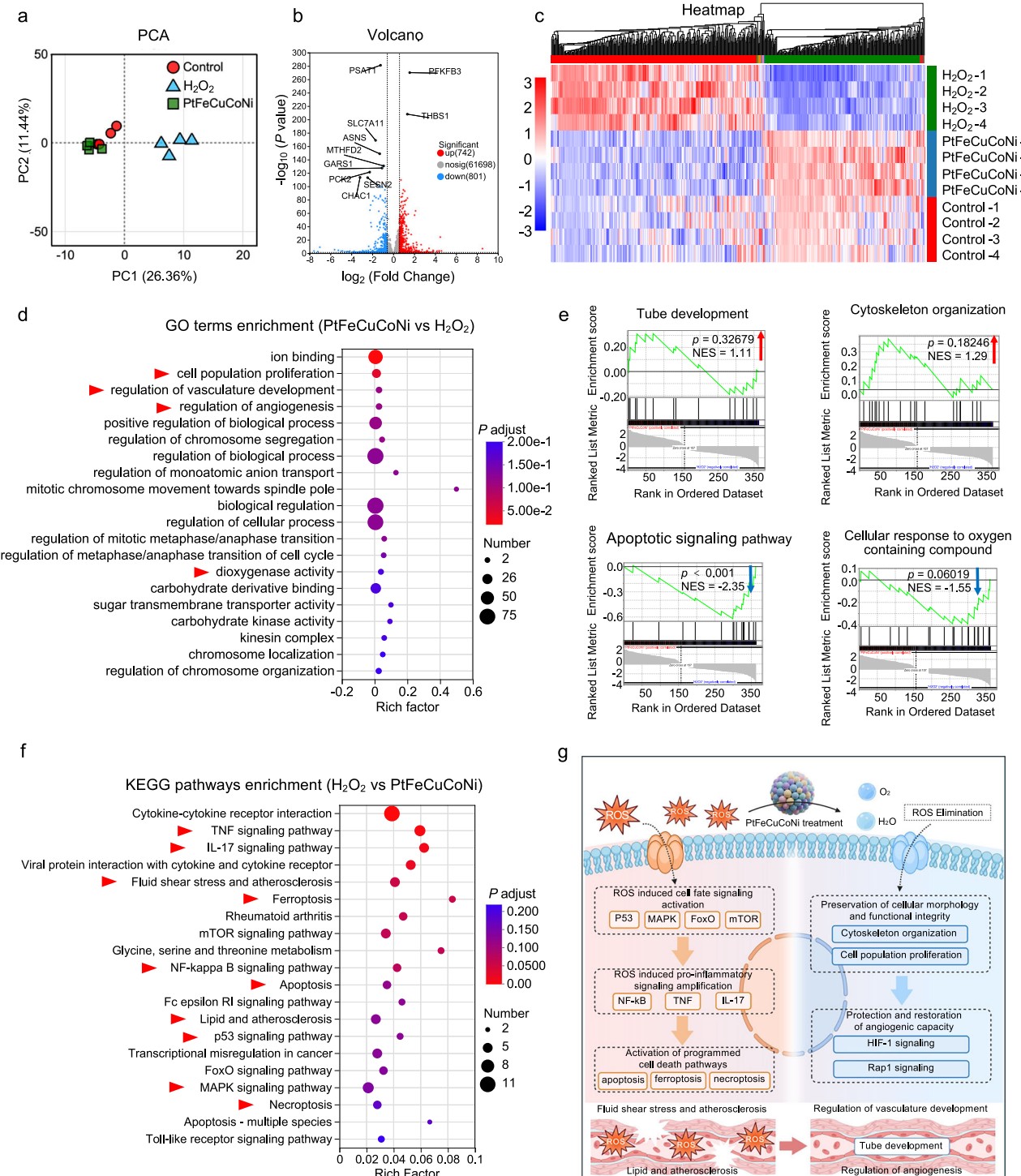

**Fig. 8 | Transcriptomic analyses reveal inflammation and repair-related pathways regulated by PtFeCuCoNi HEA NPs. a** PCA of transcriptomic profiles from control, $H_2O_2$-treated, and PtFeCuCoNi HEA NPs + $H_2O_2$-treated groups. **b** Volcano plots of DEGs (gray: not significantly different genes; red: upregulated genes; blue: downregulated genes). **c** Hierarchical clustering of DEGs from the HUVEC after different treatments. **d** Enriched GO terms of PtFeCuCoNi HEA NPs + $H_2O_2$ versus $H_2O_2$. **e** GSEA of $H_2O_2$ versus PtFeCuCoNi HEA NPs + $H_2O_2$. **f** KEGG pathways of $H_2O_2$ versus PtFeCuCoNi HEA NPs + $H_2O_2$. **g** Schematic diagram of the antimicrobial

mechanism of HEA NPs (Created in BioRender. Tian Chen (2025) https://BioRender.com/hs7ivcs). In (**b**), *p*-value was obtained from two-sided DESeq2 test without multiple comparison. In (**d**, **f**), *p*-value was obtained from a one-sided Hypergeometric test without multiple comparisons. In (**e**), *p*-value was obtained from a one-sided Permutation test without multiple comparisons. In (**e**), NES indicates the normalized enrichment score. Experiments were repeated independently three times with similar results. Source data are provided as a Source data file.

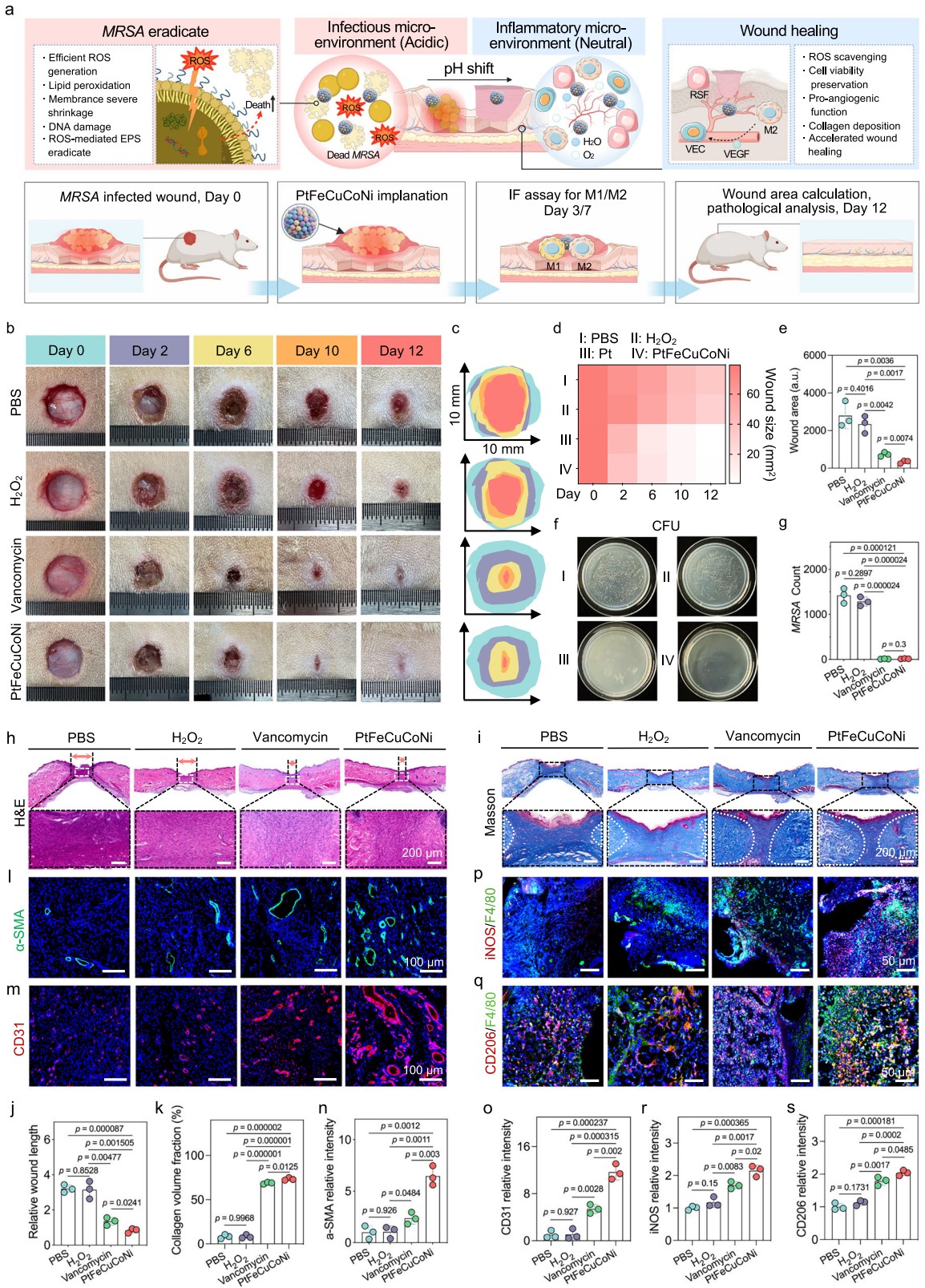

Third, beyond their catalytic efficacy and safety, these Janus artificial enzymes precisely orchestrate macrophage polarization, transitioning macrophages from an initial pro-inflammatory M1 phenotype critical for rapid pathogen clearance toward an anti-inflammatory and regenerative M2 phenotype vital for inflammation resolution and tissue repair. This programmed immunomodulation closely mimics natural wound-healing processes, providing a significant advantage over conventional single-function therapeutics. Transcriptomic analyses support this immunoregulatory capacity, demonstrating coordinated activation of protective antioxidant pathways (e.g., *PFKFB3*[86], *THBS1*[87]) coupled with the suppression of pro-inflammatory and cellular stress signals (e.g., *SLC7A11*[65], *PSAT1*[88]). Finally, although our current study focused primarily on infected chronic wounds, the fundamental advantages of HEA-based Janus artificial enzymes for

**Fig. 9 | Microenvironment-adaptive and pH-gated ROS-catalytic healing of MRSA-infected cutaneous lesions. a** Schematic of pH-gated redox biocatalysis for sequential therapy: PtFeCuCoNi HEA NPs kill bacteria (stage 1) and promote wound healing (stage 2) (Created in BioRender. Tian Chen (2025) https://BioRender.com/hs7ivcs). **b** Representative images of wound size change on different days. **c** Traces of wound closure over 12 days for groups treated with PBS, $H_2O_2$, Vancomycin, and HEA NPs, respectively. **d** Wounds sizes after being treated by different systems. **e** The sizes of the wounds treated by different systems on day 12. **f** MRSA colonies that were harvested from different groups on day 3. **g** The counted bacterial colony numbers in (**f**) ($n = 3$ independent replicates). **h, i** H&E, Masson staining images of the epidermal histological sections in different groups after 12 days. **j** Quantitative analysis of relative epidermal length in (**h**) ($n = 3$ independent replicates). **k** Quantitative analysis of collagen volume fraction ($n = 3$ independent replicates).

Representative immunofluorescence staining of **l** α-SMA and **m** CD31 in different groups. **n** Quantification of fluorescence intensity in (**l**) ($n = 3$ independent replicates). Quantification of fluorescence intensity in **o** ($n = 3$ independent replicates). **p** Representative immunofluorescence staining of F4/80 + iNOS (M1-like) on day 3. **q** Representative immunofluorescence staining of F4/80 + CD206 (M2-like) on day 10. **r** Quantification of fluorescence intensity in **p** ($n = 3$ independent replicates). **s** Quantification of fluorescence intensity in **q** ($n = 3$ independent replicates); Data are presented as mean ± SD, statistical significance was calculated using one-way ANOVA followed by Tukey's post-hoc test, all tests were two-sided. Experiments were repeated independently (**b, f, h, i, l, m, p, q**) at least three times with similar results. In (**e**), a.u. indicates the arbitrary units. Source data are provided as a Source data file.

redox modulation, such as high catalytic efficiency, minimal cytotoxicity, and versatile immunomodulation, suggest their broad applicability to a variety of drug-resistant infections beyond skin lesions. For instance, systemic infections involving biofilm-forming pathogens, infections associated with implanted medical devices, or resistant microbial colonization in mucosal sites could all benefit from the adaptive antimicrobial and anti-inflammatory potential of these Janus artificial enzymes.

Furthermore, the rational design strategy presented here provides a valuable prototype for the development of multifunctional therapeutic platforms aimed at other redox-related pathological contexts, such as diabetic ulcers, ischemic tissue injuries, chronic inflammatory diseases, and even redox-modulated tumor microenvironmental therapies. Nevertheless, future research could further elucidate the detailed pharmacokinetic behaviors, long-term catalytic stability, systemic immune responses, and specific tissue-targeting capacities of these HEA-based Janus artificial enzymes to realize their clinical translational potential fully. Continued optimization of atomic configurations, surface chemistry modifications, and delivery system integration represent promising directions to expand and enhance their therapeutic versatility and efficacy in broader clinical settings.

## Methods

### Materials and reagents

The iron acetylacetonate ($Fe(acac)_3$), copper nitrate hydrate ($Cu(NO_3)_2 \cdot 3H_2O$), cobaltous nitrate hexahydrate ($Co(NO_3)_2 \cdot 6H_2O$), nickel nitrate hexahydrate ($Ni(NO_3)_2 \cdot 6H_2O$), DMF, EG, $H_2O_2$, TMB, DMPO, and TEMP, TBA, BQ, sodium azide ($NaN_3$), potassium superoxide ($KO_2$), NBT and DPA were purchased from Aladdin reagents (Shanghai, China). The platinum bis(acetylacetonate) ($Pt(acac)_2$) was purchased from Energy Chemical (Anhui, China). The titanic sulfate ($Ti(SO)_4$) was purchased from Macklin Reagents (Shanghai, China).

### Synthesis of PtFeCuCoNi HEA NPs

$Pt(acac)_2$ (20 mg), $Fe(acac)_2$ (17.96 mg), $Cu(NO_3)_2 \cdot 3H_2O$ (12.1 mg), $Co(NO_3)_2 \cdot 6H_2O$ (17.65 mg), and $Ni(NO_3)_2 \cdot 6H_2O$ (17.63 mg) were dissolved in a 100 mL Teflon-lined autoclave with DMF (12 mL) and EG (8 mL) mixed solution and ultrasonic for 30 min to make all reactants completely dissolved. The resulting solution was heated at 200 °C for 8 h. The final product was washed and collected via centrifugation (10,000 rpm) with ethanol and acetone. Moreover, Pt metallene was prepared under typical conditions using only the $Pt(acac)_2$ as a metal precursor. The preparation conditions of FeCuCoNi, PtCuCoNi, PtFeCoNi, PtFeCuNi, and PtFeCuCo NPs are similar to those of PtFeCuCoNi HEA NPs except that one of the metal precursors is absent.

### In-situ FTIR tests

Operando infrared spectroscopy was recorded on an infrared spectrometer (Thermo Scientific, iS50 FTIR) equipped with an operando spectrum cell (SPECEL-III). To be specific, dissolve 10 mg of the catalyst in a 10 mL mixture solution (210 μL isopropyl, 750 μL $H_2O$, and 40 μL Nafion) to prepare the test sample. Then, adding 40 μL of the above catalyst solution into 5 mL acetic acid buffer (pH = 4.5) solution with 0.5 M $H_2O_2$ at room temperature for testing.

### Structural characterization

SEM images were obtained by using an Apreo S HiVoc (Thermo Fisher Scientific, FEI). Transmission electron microscopy (TEM) images and mapping were obtained via a Talos F200x TEM microscope (FEI Ltd., USA) operated at 200 kV. Aberration-corrected scan transmission electron microscopy (AC-STEM) characterization (FEI Titan Cubed Themis G2 300) was used for magnified high-angle annular dark-field scanning transmission electron microscopy (HAADF-STEM). X-ray diffraction (XRD) pattern presented the crystal phase state via a DX-2700BH X-ray diffractometer with Cu radiation at a voltage of 40 kV. XPS spectra were measured using a K-Alpha™ + X-ray Photoelectron Spectrometer System (Thermo Scientific) equipped with a Hemispherical 180° dual-focus analyzer and a 128-channel detector. EPR measurements were carried out using a Bruker EPR EMX Plus spectrometer (Bruker Beijing Science and Technology Ltd, USA) at a frequency of 9.8 GHz (microwave power: 1 mW). Inductively coupled plasma-Mass Spectrometry (ICP-MS) was carried out using an Agilent 7850 (Agilent Technologies, CA, USA). In-situ FTIR measurements were conducted using an infrared spectrometer (Thermo Scientific, iS50 FTIR) equipped with an in-situ spectrum cell (Shanghai Yuanfang Technology Co., Ltd., SPECEL-lll). The X-ray absorption spectra (XAS) were collected on the beamline BL07A1 at the NSRRC, and the radiation was monochromatized using a Si (111) double-crystal monochromator. XANES (X-ray absorption near-edge structure) and EXAFS (extended X-ray absorption fine structure) data reduction and analysis were handled via Athena software.

### POD-like activity

The PtFeCuCoNi NPs solution (4 mg mL$^{-1}$, 10 μL), $H_2O_2$ (0.1 M, 25 μL), and TMB (10 mg mL$^{-1}$, 24 μL) were added into a NaOAc-HOAc buffer (100 mM, pH 4.5). The final volume of the mixture was adjusted to 2 mL with NaOAc-HOAc buffer. Then, a portion of the mixture was used for UV-vis spectroscopy measurements at an absorbance of 652 nm.

### Steady-state enzyme dynamic studies of POD-like activity

The Michaelis-Menten constant was calculated based on the Michaelis-Menten saturation curve. For each $H_2O_2$ concentration, the initial reaction rates ($V_O$) were calculated from the absorbance variation using the Beer-Lambert Law (Eq. 1) ($\varepsilon = 39000$ M$^{-1}$ cm$^{-1}$ for oxTMB; c indicates the oxTMB concentration; l cm for the length of the solution in the light path). The reaction rates were then plotted against their corresponding $H_2O_2$ concentration and then fitted with the Michaelis-Menten curves (Eq. 2). Furthermore, a linear double-reciprocal plot (Lineweaver-Burk plot, Eq. 3) was used to determine the maximum reaction velocity ($V_{max}$) and Michaelis constant ($K_m$). Furthermore, the turnover number (TON) was calculated according to Eq. (4). The

amount of $H_2O_2$ was 15 μL, 25 μL, 50 μL, 100 μL, 200 μL (0.1 M), and 40 μL, 60 μL, 100 μL (1 M) respectively.

$$A = \varepsilon l c \tag{1}$$

$$v_0 = \frac{V_{max} \cdot [S]}{K_m + [S]} \tag{2}$$

$$\frac{1}{V_0} = \frac{K_m}{V_{max}} \cdot \frac{1}{[S]} + \frac{1}{V_{max}} \tag{3}$$

$$TON = V_{max}/[E_0] \tag{4}$$

[S] is the concentration of $H_2O_2$, and $[E_0]$ is the molar concentration of metal in enzyme mimics.

## Free radical quenching test

6 μL biocatalysts (4 mg mL$^{-1}$), 25 μL TMB (10 mg mL$^{-1}$), 25 μL $H_2O_2$ (0.1 M), and appropriate quenching agent (200 μL TBA), 100 μL BQ (10 mg mL$^{-1}$), and 100 μL NaN$_3$ (10 mg mL$^{-1}$) were added to NaOAc-HOAc buffer (100 mM, pH 4.5) to reach 2 mL. The whole color rendering process was evaluated by measuring the absorbance of oxTMB at 652 nm (UV-Vis spectroscopy, L6S, INESA, China). Among them, •OH quenched by TBA, •O$_2^-$ quenched by BQ, and $^1O_2$ quenched by NaN$_3$ in the catalytic oxidation process of TMB.

## Detection of $^1O_2$ by 9,10-diphenanthraquinone (DPA)

DPA was a specific probe that could react with $^1O_2$ to produce 9,10-diphenanthraquinone dioxide (DPO$_2$), which tested the absorbance around 378 nm. 25 μL PtFeCuCoNi NPs (4 mg mL$^{-1}$), 25 μL $H_2O_2$ (0.1 M), and 100 μL DPA-Dimethyl sulfoxide (DMSO) solution (1 mg mL$^{-1}$) were added to 1850 μL NaOAc-HOAc buffer (pH 4.5) to reach 2 mL. Then, the entire spectrum was evaluated by measuring the absorbance at 378 nm (UV-Vis spectroscopy, L6S, INESA, China).

## EPR measurements

5,5-dimethyl-1-pyrroline N-oxide (DMPO) was used to detect •O$_2^-$. 10 μL of PtFeCuCoNi NPs solutions (10 mg mL$^{-1}$), 10 μL of DMPO, and 10 μL of 10 M $H_2O_2$ were added into 500 μL of DMSO. EPR measurements were carried out via the Bruker EPR EMX Plus (Bruker Beijing Science and Technology Ltd, USA) at a frequency of 9.8 GHz (microwave power: 1 mW).

2,2,6,6-tetramethylpiperidine (TEMP) was used to detect $^1O_2$. 10 μL of PtFeCuCoNi NPs solutions (10 mg mL$^{-1}$), 10 μL of TEMP, and 10 μL of 10 M $H_2O_2$ were added into 500 μL of NaOAc-HOAc buffer (pH = 4.5). The EPR was performed using the Bruker EPR EMX Plus, as mentioned above.

## •O$_2^-$ scavenging test

1 mg Potassium superoxide (KO$_2$) was dissolved into 1 mL dimethyl sulfoxide solution (DMSO, containing 3 mg/mL 18-crown-6-ether) to generate and stabilize •O$_2^-$. Then, the biocatalysts were dispersed into the above KO$_2$/DMSO solution at a final concentration of 50 μg/mL. After reaction for 5 min, the remnant •O$_2^-$ will be trapped by nitroblue tetrazolium (NBT)-DMSO solution (10 μL, 10 mg/mL). The absorbance of the solution at 680 nm was measured and then compared with the original concentration of •O$_2^-$ to ensure the •O$_2^-$ scavenging ability.

## CAT-like test: $H_2O_2$ elimination

A total of 10 mM of $H_2O_2$ and 50 μg/mL of biocatalysts were mixed in PBS (pH = 7.4) to 2 mL. Then, 50 μL of the above solution was mixed with Ti(SO$_4$)$_2$ solution (100 μL, 13.9 mM), and the absorbance value at 405 nm was recorded every 5 min for 30 min.

## O$_2$ generation assay

A total of 100 mM of $H_2O_2$ and 10 μg/mL of biocatalysts were mixed in PBS (pH = 7.4) to 20 mL, followed by measuring the O$_2$ concentration using a dissolved oxygen meter (INESA, JPSJ-605F) every 5 s until 300 s.

## Steady-state dynamic parameters

To analyze the biocatalytic kinetics of O$_2$ generation, 10 μg/mL of biocatalysts and different concentrations of $H_2O_2$ were mixed in PBS to obtain 20 mL, and then the O$_2$ concentration was measured every 5 s for 100 s. The values of $V_{max}$ and $K_m$ were calculated based on the Michaelis-Menten saturation curve. The specific calculation formula was referred to in section - Steady-state enzyme dynamic studies of POD-like activity.

## Theoretical calculation

All theoretical calculations were performed using the Density functional theory (DFT) method, as implemented in the Vienna ab initio simulation package (VASP)[117,118]. The core electrons were described using the spin-polarized projector augmented wave (PAW) method[119], and the electron exchange and correlation energy was treated within the generalized gradient approximation in the Perdew-Burke-Ernzerhof functional (GGA-PBE)[120]. The valence states of all atoms were expanded in a plane-wave basis set with a cutoff energy of 450 eV. The convergence criteria for the electronic self-consistent iteration and force were set to $10^{-5}$ eV and 0.02 eV/Å, respectively, with a Gamma-centered $1 \times 1 \times 1$ K-points. Additionally, Van der Waals interactions were included in all calculations using DFT-D3[121,122]. The HEA (111) slab was modeled by three layers of slabs. During the HEA structure simulation, the two bottom layers of the substrates were fixed, while the top layer was kept fully relaxed. A slab model was constructed with a vacuum layer of 18 Å in the z-direction to avoid interaction between neighboring images. And during the *O intermediate structure simulation, the three bottom layers of the substrates were fixed, while the rest of the layers were kept fully relaxed. A slab model was constructed with a vacuum layer of 18 Å in the z-direction to avoid interaction between neighboring images. The charge density differences were evaluated using the formula $\Delta\rho = \rho_{A+B} - \rho_A - \rho_B$, where $\rho_X$ is the electron density of X. Atomic charges were computed using the atom-in-molecule (AIM) scheme proposed by Bader[123,124].

To explore the catalytic effect, the change of Gibbs free energy (ΔG) was calculated (Eq. 5), which is defined as:

$$\Delta G = \Delta E + \Delta ZPE + \Delta H_{0 \to 298K} - T\Delta S \tag{5}$$

where ΔE is the energy change obtained from DFT calculations; ΔZPE, ΔH, and ΔS denote the difference in zero-point energy, enthalpy, and entropy due to the reaction, respectively. The enthalpy and entropy of the ideal gas molecule were obtained from standard thermodynamic tables, and some of the calculation results were analyzed using the VASPKIT package[125].

## Antibacterial experiments

*MRSA* (ATCC 43300, Gram-positive) was used as a representative pathogenic bacterium to investigate the bacterial capture and eradication abilities of the PtFeCuCoNi HEA NPs. The bacteriostatic property was studied by MIC values, MBC, the absorbance at 600 nm (OD$_{600}$), and agar plate counting. Bacteria were incubated separately with a series of final concentrations (2, 4, 6, 8, 16, 32, 64, 96, and 128 μg/mL) of ROS-generation biocatalysts (Pt and PtFeCuCoNi HEA NPs). The system, which contains $H_2O_2$ (0.2 mM) and bacteria ($10^6$ CFU mL$^{-1}$), was cultured at 37 °C for 12 h. Then, OD$_{600}$ values of the bacterial suspensions treated by the systems at different concentrations (2, 4, 6, 8, 16, 32, 64, 96, and 128 μg/mL) were recorded to calculate the MIC values. The ROS-generation biocatalysts (Pt and PtFeCuCoNi HEA NPs) systems (8 μg/mL) with $H_2O_2$ (0.2 mM) were cultured with 1 mL of

bacterial suspensions ($10^6$ CFU mL$^{-1}$) for 12 h at 37 °C. Then, the cultured suspensions were diluted $10^5$ times and taken for agar plate counting. The agar plates were cultured for another 12 h at 37 °C and counted to obtain the final colony quantities and calculate the MBC values. The antibacterial properties were estimated by comparing the OD$_{600}$ of different systems with that of the control. Bacterial protein leakage was quantified using a bicinchoninic acid (BCA) protein assay kit (Solarbio). Briefly, *MRSA* were incubated with PBS, H$_2$O$_2$, Pt + H$_2$O$_2$, PtFeCuCoNi + H$_2$O$_2$ under the indicated conditions. After incubation, supernatants were collected by centrifugation, and protein content was determined following the manufacturer's protocol. A standard curve was established using bovine serum albumin (BSA, 0–0.5 mg/mL), and the corresponding optical density at 562 nm (OD$_{562}$) was recorded. The calibration equation ($y = 1.698x + 0.166$, $R^2 = 0.992$) was used to convert OD values into absolute protein concentrations. Representative images of the gradient color reaction were also obtained. All measurements were performed in triplicate.

### Extraction and Gel Electrophoresis of EPS from *MRSA* Biofilms

Biofilm Culture and Treatment: 10 mL of *MRSA* suspension (OD$_{600}$ ≈ 0.05) was incubated at 37 °C for 24 h to form biofilms. The biofilms were then treated with PBS (control), H$_2$O$_2$, Pt NPs + H$_2$O$_2$, or PtFeCuCoNi NPs + H$_2$O$_2$ for 6 h. EPS Extraction: After treatment, the biofilms were scraped and centrifuged to remove bacterial cells. Agarose Gel Electrophoresis: 20 μL EPS was mixed with 4 μL loading buffer, heated at 95 °C for 5 min, and loaded onto a 1% agarose gel. Electrophoresis was carried out at 100 V for 60 min. Staining and Imaging: The gel was stained with SYBR Green I for 30 min in the dark, and images were captured using a fluorescence imaging system. Degradation effects were assessed by comparing band migration and fluorescence intensity.

### EPS nucleic acid electrophoresis

*MRSA* biofilms were treated with PBS (control), H$_2$O$_2$, Pt NPs + H$_2$O$_2$, or PtFeCuCoNi NPs + H$_2$O$_2$ for 6 h. EPS was extracted by centrifuging scraped biofilms at 12,000 × *g* for 20 min. The supernatant was mixed with DNA loading buffer, heated at 95 °C for 5 min, and analyzed on a 1% agarose gel. Nucleic acids were visualized using SYBR Green I. DNase I treatment confirmed nucleic acid identity.

### Observation of bacterial morphology

The bactericidal performance of these systems was visualized via SEM, TEM, and fluorescence microscopy. After treating different systems, the bacterial suspensions were fixed with a 2.5% glutaraldehyde-containing PBS solution for 12 h at 4 °C and dehydrated using a gradient of ethanol/water. Then, SEM and TEM images were obtained to assess the bacterial capture ability and morphology.

### Crystal violet staining of bacterial biofilms

The treated biofilms from each group were washed three times with PBS and then fixed with 2.5% glutaraldehyde for 20 min. Then the liquid in each well was discarded. Crystal violet test solution (0.5%) was transferred into each well for 30 min, and excess dye was removed with PBS. Furthermore, the coverslips were allowed to dry. Next, absolute ethyl alcohol was used to dissolve the crystal violet bound to the biofilm. Finally, the value of Optical Density (OD) 570 nm of the dissolved solution was examined by a microplate reader (SAF-6801, Bajiu Corporation, Shanghai, China).

### Live/Dead and LPO staining

Regarding *MRSA* biofilm culturing, the *MRSA* suspension (100 μL, $10^8$ CFU mL$^{-1}$) and a lysogeny broth medium (LB, 100 μL) were placed in ibidi 8-well slides, and then they were cultured at 37 °C. After 24 h, the medium was removed, and the unattached bacteria were gently washed away with sterile PBS three times. The resulting biofilm on the

ibidi 8-well slides was then harvested. Moreover, the killing behaviors of different samples were stained by Live/Dead BacLight Viability Kits (SYTO-9 for live cells and PI for dead cells) for observation using CLSM (St5, Leica). The LPO was detected by C11-BODIPY$^{581/591}$ (Thermo Fisher, D3861). In general, pure medium, Pt, and PtFeCuCoNi HEA NPs were co-cultured with *MRSA* for 12 h, respectively. After washing, *MRSA* was stained by C11-BODIPY$^{581/591}$ solution at a concentration of 5 mM for 20 min. Then, the CLSM (St5, Leica) was used to obtain the images. EPS were detected by Dextran Alexa Flour 647 (Thermo Fisher, D22914). In general, pure medium, H$_2$O$_2$, Pt, and PtFeCuCoNi HEA NPs were co-incubated with *MRSA* for 12 h. After washing, *MRSA* was stained with Dextran Alexa Flour 647 solution at a concentration of 10 mM for 30 min. A CLSM (St5, Leica) was then used to obtain images.

### Transcriptome sequencing of *MRSA* and data analysis

For RNA-seq analysis, *MRSA* was co-cultured with PBS, PtFeCuCoNi HEA NPs + H$_2$O$_2$ at 37 °C for 3 h. After different treatments, bacteria were collected for RNA-seq analysis by Shanghai Personalbio Co., Ltd on the Illumina platform. GO (http://www.geneontology.org) and Kyoto Encyclopedia of Genes and Genome (http://www.genome.jp/kegg/) were used to analyze the gene functions. Differential gene expression analysis was performed using the R package, and those genes conformed to |log$_2$ Fold change| > 1 (P-value < 0.05) were considered to be DEGs.

### Cell culture

HUVEC and Raw264.7 were purchased from the American Type Culture Collection (ATCC). HUVEC were cultured in the medium of Dulbecco's Modified Eagle Medium (DMEM, 6123045, Gibco) containing 10% fetal bovine serum (FBS), and 1% antibiotics mixture (10,000 U penicillin and 10 mg streptomycin) at 37 °C under 5% CO$_2$. Raw264.7 were cultured in high-glucose DMEM supplemented with 10% FBS at 37 °C in a humidified incubator with 5% CO$_2$. RSF were isolated from the skin of the Sprague-Dawley (SD) rats. Briefly, Primary dermal fibroblasts were isolated from the dorsal skin of SD rats by enzymatic digestion, in which excised dermal tissue was minced and digested with collagenase type I, followed by filtration and culture in α-MEM supplemented with 10% FBS. The collected cells were cultured in α-MEM supplemented with 10% fetal calf serum and 1% penicillin-streptomycin at 37 °C in a 5% CO$_2$ incubator. After 24 h of culture, the culture medium was replaced with fresh complete α-MEM medium to remove the nonadherent cells. The adherent cells were cultured for another 3–5 days and passaged until reaching 90% confluence.

### Cellular protection assay

Cells were pretreated with the corresponding biocatalyst (2 μg mL$^{-1}$) for 24 h and then co-incubated with 200 μM H$_2$O$_2$ for 2 h. Cells without any treatment were regarded as the Control group. Cells were treated with 200 μM H$_2$O$_2$, and those without biocatalysts were regarded as the ROS Condition group. After stimulation, the cell viability rate was detected with Cell Counting Kit-8, and the Live/Dead ratio was determined by Calcein acetoxymethyl ester/Propidium Iodide (Calcein AM/PI) Kit.

After stimulation, cell morphology was detected via cytoskeletal staining. Cells were seeded at low density and treated before fixation with 4% paraformaldehyde (#BL539A, Biosharp, China) for 15 min at room temperature, followed by permeabilization with 0.1% Triton X-100 (#T8200, Solarbio, China). Rhodamine phalloidin (#A12379, Invitrogen, China) was used for F-actin (red) staining for 30 min, and DAPI (#P0131, Beyotime, China) was used for DNA (blue) staining for 5 min. Fluorescence photographs were taken using a confocal laser endomicroscope (FV2000, Olympus, Japan). After stimulation, DCFH-DA (S0033S, Beyotime, China) was used to detect the residual ROS level, according to manufacturers' instructions. Briefly, probes were preloaded by incubating cells with the DCFH-DA working solution for

30 min. After washing with PBS three times, the intracellular ROS level was determined through a CLSM (FV3000 and Nikon, A1R MP+, Olympus, Japan). After stimulation, Calcein AM/PI staining was performed according to the instructions (C2012, Beyotime, China). Briefly, 2 μM Calcein AM solutions in PBS were used to stain live cells, and then 4.5 μM PI solutions were used to stain the dead. The cells were captured by a CLSM and counted by the Celigo Image Cytometer (Nexcelom Bioscience LCC., America).

## Cell wound scratch assay

Cells were seeded in 6-well plates and treated as before. Draw a line vertically with the 200-microliter gun tip over the ruler. The cells were washed three times with PBS, photographed under the microscope (FV2000, Olympus, Japan), and then placed at 37 °C in 5% $CO_2$. HUVEC and RSF were photographed at 12 and 24 h to observe the healing. The wound closure rates were accounted for via the ImageJ program (Media Cybernetics, Rockville, USA).

## Matrigel tube formation

The possibility of tube formation of HUVEC was tested using Matrigel (ABW, 082706, China) according to the manufacturer's instructions. Briefly, the matrix gel was thawed overnight at 4 °C, and 50 μL was added to each well of a 96-well plate. The petri dish with Matrigel was incubated at 37 °C for 30 min to allow the Matrigel to solidify. The cells were trypsinized and inoculated into Matrigel at $1 \times 10^5$ cells/cm$^2$ for 4 h. Then, observed under an automatic inverted microscope. The vascular network was analyzed by the ImageJ angiogenesis analyzer.

## Macrophage polarization

Raw264.7 cells were seeded at a density of $5 \times 10^4$/well and processed as above. Primary antibodies: anti-CCR7 (ab-196640, abcam, USA, 1:400 dilution) and anti-CD163 (ab-182422, abcam, USA, 1:400 dilution). Secondary antibodies: Fluor 647 donkey anti-rabbit (A31573, Life Tech., 1:500 dilution).

## Transcriptome sequencing and data analysis

HUVEC ($5 \times 10^6$ cells mL$^{-1}$) were cultured in 100 mm dishes and were treated with different samples for 24 h. Then, the cells were cultured overnight in a standard culture medium. After being treated with different samples, the HUVEC were lysed by TRIzol™ reagent (15596026, Invitrogen, CA, USA), and cell lysates were stored at −80 °C before sequencing. RNA concentration and purity were measured using NanoDrop 2000. The integrity of RNA samples was assessed by the DNF-471 kit, and the system was an Agilent 5300. The libraries were sequenced on an Illumina NovaSeq platform to generate 150 bp paired-end reads, according to the manufacturer's instructions. DEGs were defined by DESeq2/edgeR with thresholds of $|\log_2 FC| \geq 1$ and FDR < 0.05. The PCA, Volcano plot, Heat map, KEGG pathway enrichment, Chordal plots of KEGG, GO term enrichment, and GSEA pathway enrichment analyses were performed using the online platform Majorbio (www.majorbio.com).

## Real-time quantitative polymerase chain reaction (RT-qPCR)

HUVEC were pre-incubated with the corresponding biocatalyst (2 μg mL$^{-1}$) for 24 h, followed by co-incubation with 200 μM $H_2O_2$ for an additional 3 h. After stimulation, total mRNA was extracted from HUVEC, and the expression levels of PFKFB3, THBS1, HSPA8, CHAC1, ASNS, and PSAT1 were determined by RT-qPCR. Similarly, Raw264.7 cells were pre-incubated with the corresponding biocatalyst (1 μg mL$^{-1}$) for 24 h, and then followed by co-incubation with 500 μg/mL lipopolysaccharide (LPS) for an additional 24 h. After stimulation, total mRNA was extracted from Raw264.7 cells, and the expression levels of iNOS, IL-6, TNF-α, Arg-1, IL-10, and TGF-β were determined by RT-qPCR. The primer sequences are listed in Supplementary Table 3.

## Hemolysis test

The hemolysis test was performed on the red blood cells (RBCs) of the rat according to the reported method. The 2 mL of blood was collected from the rat, and RBCs were immediately washed with PBS and centrifuged at 5000 rpm. After washing, 1 mL of RBCs was dissolved in 25 mL of PBS. Then 300 μL of RBC was added separately to a vial containing 8, 16, 32, 64 μL of PtFeCuCoNi HEA NPs dispersed in PBS. The positive and negative controls were prepared by dispersing RBCs in 0.1% Triton X-100 and PBS, respectively. After 1 h of incubation at 37 °C, samples are centrifuged at 5000 rpm, and 200 μL of the supernatant is collected for absorbance measurement at 570 nm. The percentage of hemolysis is calculated as follows: P = (AS−AN)/(AP−AN) × 100%. Where AS is the absorbance resulting from the mixture of NPs and RBCs, AN and AP represent the absorbance of negative and positive controls, respectively.

## In vivo wound disinfection experiments

Rats were selected as models in animal experiments. The random assignment was conducted under conditions where there was minimal variation in the health status of all samples/organisms. The protocols of all the animal experiments were permitted and carried out as requested. After being anesthetized with 3% sodium pentobarbital, a hole around 1 cm in diameter in the epidermis was created. Then 200 μL of *MRSA* suspension ($1 \times 10^8$ CFU mL$^{-1}$) was dropped onto the wound and incubated for 1 day. After that, 200 μL of liquid containing PtFeCuCoNi HEA NPs (8 μg/mL) and $H_2O_2$ (0.2 mM) was added to treat the infected wound. Normal saline, $H_2O_2$ (0.2 mM), and vancomycin (80 μg/mL) were also used as contrasts. After 12 days, the treated wounds were cut off and fixed with 10% formaldehyde solution for histological H&E, Masson trichrome (Solarbio, G1015).

## Immunofluorescence staining

Skin samples from SD rats were fixed with 4% buffered paraformaldehyde overnight. After dehydration, samples were embedded in paraffin and sectioned. The paraffin sections were dewaxed in xylene and then rehydrated with a gradient of alcohols. After the slides were immersed in antigen retrieval solution for 30 min at 95 °C, they were washed with PBS twice and fixed in 4% paraformaldehyde for 10 min at room temperature, then permeabilized in PBS containing 0.1% Triton X-100 for 10 min and blocked in PBS with 1% bovine serum albumin (BSA) for 1 h. Primary antibodies were diluted in PBS with 1% BSA and incubated overnight at 4 °C. Then, the cells were incubated with secondary antibodies with the required fluorophore in the dark at room temperature for 2 h. Before imaging, cells were counterstained with 10 μg mL$^{-1}$ DAPI and mounted in the antifading mounting medium (S2110, Solarbio, China). Samples were washed with PBS three times between each step. A CLSM (FV3000 and Nikon, A1R MP+, Olympus, Japan) was used to capture images.

Primary antibodies and corresponding concentrations used in this study were α-SMA (ab5694, abcam, USA, 1:100 dilution), CD31 (ab24590, abcam, USA, 1:100 dilution), iNOS (sc-7271, Santa Cruz Biotechnology, USA, 1:200 dilution), CD206 (sc-58986, Santa Cruz Biotechnology, USA, 1:200 dilution), F4/80 (sc-25830, Santa Cruz Biotechnology, USA, 1:100 dilution), IL-1β (sc-52012, Santa Cruz Biotechnology, USA, 1:200 dilution), TNF-α (sc-52746, Santa Cruz Biotechnology, USA, 1:200 dilution), VEGF (ab46154, abcam, USA, 1:200 dilution). The secondary antibodies were Alexa Fluor 647 goat anti-mouse IgG (ab150115, Abcam, USA, 1:200 dilution), Alexa Fluor 647 donkey anti-rabbit IgG (ab150075, Abcam, USA, 1:200 dilution) and Alexa Fluor 488 goat anti-rabbit IgG (ab150077, Abcam, USA, 1:200 dilution), Alexa Fluor 555 goat anti-mouse IgG (A21422, Invitrogen, Thermo Fisher Scientific, USA, 1:200 dilution).

## In vivo toxicity assessment

To assess the systemic toxicological profile of synthetic biocatalytic constructs, comprehensive histopathological analyses were conducted on SD rat models following a 2-week post-operative period. Major organ systems, specifically the heart, liver, spleen, lung, and kidneys, were harvested and subjected to H&E staining protocols for microscopic evaluation of tissue architecture and cellular morphology.

## Statistical analysis

Image J (version 1.8.0) and Image-Pro Plus (version 6.0) were used for in vitro and in vivo imaging analysis. Data were analyzed using GraphPad Prism 10.0 software (GraphPad Prism, San Diego, California, USA). Figures were formed using Origin 2022. Sample size ($n$), probability ($p$) value, data normalization, and specific statistical tests for each experiment were clarified in the figure legends. All of the data were presented as the mean ± SD from a minimum of three independent experiments. The comparison of mean values across multiple groups was performed using one-way ANOVA followed by Tukey's post-hoc test; all tests were two-sided. Statistical significance was set at $*p < 0.05$, $**p < 0.01$, $***p < 0.001$, and ns represents a non-significant difference. A value of $p < 0.05$ was considered significant.

## Ethical statement

Animal experiments and procedures, including euthanasia, were conducted following protocols approved by the Institutional Animal Care and Use Committee at Sichuan University (Number: WCHSIRB-AT-2025-513). Additionally, the Laboratory Animal Welfare and Ethics Committee of West China Hospital of Stomatology reviewed and approved the study. All experiments involving animal use were performed in accordance with the ARRIVE guidelines.

## Reporting summary

Further information on research design is available in the Nature Portfolio Reporting Summary linked to this article.

# Data availability

All data supporting the results of this study are available within the paper and its Supplementary Information. All raw data generated for the figures in this study are provided in the source data file. Source data are available for Figs. 2c, d, h–k, 3b–n, 4d–h, 5b, c, f, i, q, t, m, o, 7b, d, f, g, i, k, m and 9d, e, g, j, k, n, o, r, s and Supplementary Figs. 2–15, 17–19, 21–25, 27, 28, 30, 38–40, 44, 45, 47, 49, 54, 55, 58 and 60 in the associated source data file. The raw sequencing data generated in this study have been deposited in the NCBI Sequence Read Archive (SRA) under the BioProject accession number PRJNA1310237. These data are publicly available and can be accessed through the NCBI SRA database. Source data are provided with this paper.

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

## Acknowledgements

This work was financially supported by the National Key R&D Program of China (2024YFE0201200 [C.C.], 2022YFA1104400 [W.D.T.]), the National Natural Science Foundation of China (82100959 [T.C.], 52373148 [C.C.], 52173133 [C.C.], 52525311 [C.C.], 82561160101 [C.C.]), the Sichuan Science and Technology Program (2024YFFK0068 [Z.L.]), and the Research and Development Program, West China Hospital of Stomatology Sichuan University (RD-02-202517 [T.C.]). The authors would like to thank Dr. Hanjiao Chen of the Analytical & Testing Center of Sichuan University for their assistance on EPR.

## Author contributions

C.H. and Y.Q.W. contributed equally to this work. C.H., Y.Q.W., S.H.G., T.W., M.A., Z.L., and H.L.D. performed the experiments and analyzed the results. C.H., Y.Q.W., Z.L., T.C., L.C., W.D.T., and J.L. assisted with the figure production and experimental design. C.H., Y.Q.W., Z.L., T.C., and C.C. wrote the manuscript. T.C. and C.C. designed the experiments, corrected the manuscript, and supervised the whole project. All authors discussed the results and commented on the manuscript.

## Competing interests

The authors declare no competing interests.
