## [Transparent Peer Review file · Nature Communications]

High-Entropy Alloy Janus Artificial Enzymes for pH-Gated Sequential Redox Therapy of Drug-Resistant Bacterial Infection

Corresponding Author: Professor Chong Cheng

Version 0:

Reviewer comments:

Reviewer #1

(Remarks to the Author)

This manuscript introduces an interesting high-entropy alloy (PtFeCuCoNi)-based Janus artificial enzymes with pH-responsive dual enzymatic activities for sequential treatment of drug-resistant bacterial infections and chronic inflammatory wounds. Under acidic conditions, the biocatalytic materials demonstrates potent POD-like activity, effectively eradicating biofilms and killing bacteria. At neutral pH, it switches to antioxidant (SOD/CAT-like) functions, alleviating oxidative stress and promoting tissue regeneration. Through comprehensive in vitro and in vivo validation, the authors show that the Janus artificial enzymes not only eliminates MRSA and biofilms at ultralow concentrations but also reprograms macrophage polarization and enhances angiogenesis, offering a promising antibiotic-free strategy for managing complex wound microenvironments. Compared with earlier works, this study shows a new and interesting biocatalytic pathway to treat infected wounds, the data are sufficient and can fully validate the concepts, which offers a promising antibiotic-free strategy for managing complex wound microenvironments and is recommended for publication after addressing the following points.

1. Since the high-entropy alloys (PtFeCuCoNi) are based on different elements, I am wondering how the precursor ratios affect HEA NP crystal structure and enzyme-like activity. This will validate the optimality and improve the reproducibility of the synthesis protocol. Has the authors tried with different ratios? Maybe more discussion is needed in this part.
2. Will these materials also show SOD- and CAT- activities under acidic conditions? It is suggested to supplement more information on these catalytic activities to further verify the precision and response thresholds of pH gating effects.
3. The description on page 9, first paragraph, does not correspond to Figure 3f and 3g.
4. The catalytic activity and stability of HEA NPs over multiple reaction cycles require further investigation.
5. The writing could be improved. For example:
The horizontal axis of Figure 7b.
The p(H₂O₂) values in Figures 7 and 9.
The large or lowercase of first letter of the references title should be unified.
6. In Supplementary Fig. 38, the authors provided the detection data of the inflammatory factor IL-1 β at the skin defect site on day 10, which supports the claim that the PtFeCuCoNi HEA NPs reduces tissue inflammation during the late stage of infection. However, data on systemic inflammation are lacking and should be supplemented with relevant experiments.
7. The antimicrobial section is comprehensive, addressing both planktonic and biofilm MRSA. However, the opening paragraph of the Discussion could be further strengthened. The statement "Infections caused by antibiotic-resistant pathogens represent an escalating clinical challenge due to antibiotic treatment failure, compounded by complex inflammation and dynamic microenvironmental alterations." would be more compelling if expanded with relevant examples of severe infections caused by antibiotic-resistant bacteria (e.g., endocarditis, intra-abdominal infections, or bacteremia). This would situate the work within the broader clinical landscape.

Reviewer #2

(Remarks to the Author)

This manuscript reports the design of bioinspired Janus artificial enzymes based on high-entropy alloys (PtFeCuCoNi) for sequential therapy of drug-resistant bacterial infections and inflammatory wounds. The study is well-motivated, drawing on

the biphasic microenvironment of chronic wounds, and demonstrates that the proposed HEA biocatalysts can achieve pH-dependent switching between oxidase/peroxidase-like ROS generation under acidic conditions and antioxidant-like scavenging at neutral pH. The combination of theoretical calculations and experimental analyses provides a coherent picture of the catalytic mechanism, and the *in vivo* data indicate antimicrobial efficacy against MRSA as well as immunomodulatory and pro-regenerative effects. The concept of exploiting multi-metal synergy in HEAs for redox modulation is novel and of potential interest for the field of nanomedicine. The manuscript is scientifically sound and is suitable for publication in Nature Communications. I therefore recommend its rapid publication after the following minor revisions are addressed:

- (1) In the Supplementary Methods, it was mentioned that "Raman spectra were examined via the HORIBA HR Evolution Raman spectrometer with a 532 nm laser source", but this characterization technique was not included in the main manuscript. Is it a mistake?
- (2) The synthesis methods for the mentioned FeCuCoNi, PtCuCoNi, PtFeCoNi, PtFeCuNi, and PtFeCuCo alloys are not provided in the manuscript, which should be supplemented.
- (3) While the study highlights the advantages of HEA materials, the current text provides limited discussion of prior biomedical research on HEAs. The authors are encouraged to expand the Introduction or Discussion with additional references and a more explicit consideration of potential toxicity concerns associated with high-entropy alloys.
- (4) In Supplementary Fig. 28, the cytocompatibility of PtFeCuCoNi HEA NPs is presented. However, as Pt is a key constituent metal, the cytotoxicity of Pt NPs alone should also be discussed, which would strengthen the argument that the alloy formulation indeed reduces toxicity relative to single-metal systems.
- (5) In Fig. 6, the transcriptional responses provide valuable mechanistic insight. However, the current text would benefit from the inclusion of supporting references for some of the highlighted pathways. Including key citations would help place these findings within the broader context of bacterial stress responses.

Reviewer #3

(Remarks to the Author)

This manuscript describes the development of high-entropy alloy (HEA) (PtFeCuCoNi) as bioinspired nanocatalysts for the treatment of infected and non-healing wounds. The work addresses an important clinical challenge by targeting both the antimicrobial and tissue-regenerative phases of wound healing. The authors provide convincing evidence that the HEA nanozymes exhibit robust catalytic adaptability, functioning as ROS-generating oxidase/peroxidase mimics in acidic environments while switching to ROS-scavenging antioxidant activity at neutral pH. The integration of transcriptomic profiling, mechanistic studies, and *in vivo* validation provides strong support for the dual regulatory role of these materials in modulating oxidative stress and inflammation. The concept of tailoring redox activity through entropy-driven alloy design is innovative and of clear significance to the biomedical nanomaterials community. It is suggested that this study is well executed and suitable for publication in Nature Communications, pending minor revisions to improve clarity and contextual discussion.

1. The mechanism for the synthesis of the PtFeCuCoNi HEA NPs via the hydrothermal method should be discussed in the main manuscript.
2. Some earlier references regarding the HEA materials for antibacterial studies should be carefully compared and discussed included in the manuscript.
3. The contents of the elements measured by XPS and ICP in the manuscript are inconsistent. Please check the data and explain the difference.
4. In Fig. 6f, an interesting mechanistic illustration is presented; however, the results differ from previous sequencing studies on active bactericidal processes, where a decrease in ATP activity was observed. The authors are encouraged to further explain the reasons for this discrepancy and provide relevant literature support.
5. Fig. 7n presents a schematic illustration of cellular effects under high-concentration H₂O₂ treatment and the protective role of HEA NPs. However, the labeling of the cell types is missing, making it unclear which three cells are being represented.
6. Fig. 8d and Fig. 8f present GO and KEGG analyses, respectively. It is noted that Fig. 8d compares PtFeCuCoNi vs H₂O₂, while Fig. 8f shows the opposite comparison, H₂O₂ vs PtFeCuCoNi. As the rationale for this discrepancy has not been explained in the manuscript, the authors are advised to provide further discussion.
7. Supplementary Figs. 31 and 32 show scratch assays of HUVECs and RSFs, respectively. However, unlike Figs. 7a and 7c, the scratch assay for RAW264.7 cells is missing. The authors are advised to either provide the corresponding experiment or explain the reason for its absence.

Reviewer #4

(Remarks to the Author)

Gao et al. designed a high-entropy alloy-based Janus artificial enzyme targeting chronic wound infections. This enzyme leverages the unique microenvironment characteristics at the infection site to catalyze antimicrobial and anti-inflammatory actions, demonstrating excellent antimicrobial efficacy both *in vitro* and *in vivo*. Additionally, the study found that it can promote the PFKFB3 pathway, inhibit the expression of inflammatory cytokines, and accelerate wound healing. The results are very interesting; however, the manuscript still has some issues to address. It is recommended that the authors make further additions and revisions.

1. Through TEM and SEM observations, the Janus artificial enzyme exhibits a nanostructure. The authors should further supplement the DLS and material stability data of the artificial enzyme, particularly regarding its bioavailability. It is also recommended that the authors include stability data of the artificial enzyme under different environmental conditions.
2. The HEA nanoenzyme demonstrated different catalytic activities under acidic and neutral pH conditions, which is very interesting and differs from other reported HEA-based artificial enzymes. However, the underlying mechanism is not

sufficiently discussed. Strengthening this part would improve the manuscript.

2. Fig. 5C is used to detect bacterial protein leakage, typically assessed by the BCA assay. However, the manuscript does not provide a detailed description of this method. Given the limited bacterial protein content, the detection results showed a deep purple color and lacked a standard curve. It is recommended that the authors provide further clarification on this issue.

3. The authors demonstrated the reduction of EPS through SEM characterization, but this claim has certain limitations. It is recommended that the authors include a biofilm EPS gel electrophoresis experiment to further validate this result.

4. The authors demonstrated lipid peroxidation in MRSA after treatment with the artificial enzyme by the increase in LPO. It is recommended that the authors include MDA detection at the bacterial level to further confirm this phenomenon.

5. The antibacterial mechanism of the artificial enzyme primarily involves the generation of ROS. However, the manuscript lacks relevant experiments at the bacterial level. It is recommended that the authors include ROS generation at the bacterial level after labeling with the DCFH-DA probe.

6. At the cellular level, the artificial enzyme exhibits anti-inflammatory effects. The authors confirmed its ROS scavenging ability using the DCFH-DA probe; however, ROS scavenging is only one step in the anti-inflammatory process. It is recommended that the authors supplement the study with the polarization trends of M1 and M2 macrophages at the cellular level and concurrently assess the release of inflammatory cytokines TNF- α , IL-6, and IL-10 at the cellular level.

7. The authors indicate that the artificial enzyme can activate the PFKFB3 pathway at the transcriptomic level. However, the authenticity of the activation results based solely on transcriptomic data remains to be further verified. It is recommended that the authors supplement the study with RT-qPCR testing of relevant factors to further confirm these results.

Version 1:

Reviewer comments:

Reviewer #1

(Remarks to the Author)

The authors have addressed my concerns.

Reviewer #2

(Remarks to the Author)

The authors have replied to all my question, made revisions according to my suggestions, and agreed to the publication.

Reviewer #3

(Remarks to the Author)

The authors have appropriately addressed the requests and concerns of the reviewers in a very detailed and thorough manner.

Publication of the manuscript can be recommended in the present form.

Reviewer #4

(Remarks to the Author)

The very detailed revision from the authors have addressed all of my concerns. I recommend the manuscript is accepted as it is.

Point-by-point response to the detailed comments by reviewers of “High-Entropy Alloy-based Janus Artificial Enzymes with pH-Gated Redox Biocatalysis for Sequential Therapy of Drug-Resistant Bacteria and Inflammatory Wounds”

REVIEWER COMMENTS

Reviewer #1 (Remarks to the Author):

“This manuscript introduces an interesting high-entropy alloy (PtFeCuCoNi)-based Janus artificial enzymes with pH-responsive dual enzymatic activities for sequential treatment of drug-resistant bacterial infections and chronic inflammatory wounds. Under acidic conditions, the biocatalytic materials demonstrates potent POD-like activity, effectively eradicating biofilms and killing bacteria. At neutral pH, it switches to antioxidant (SOD/CAT-like) functions, alleviating oxidative stress and promoting tissue regeneration. Through comprehensive in vitro and in vivo validation, the authors show that the Janus artificial enzymes not only eliminates MRSA and biofilms at ultralow concentrations but also reprograms macrophage polarization and enhances angiogenesis, offering a promising antibiotic-free strategy for managing complex wound microenvironments. Compared with earlier works, this study shows a new and interesting biocatalytic pathway to treat infected wounds, the data are sufficient and can fully validate the concepts, which offers a promising antibiotic-free strategy for managing complex wound microenvironments and is recommended for publication after addressing the following points.”

Response to the general comment:

We sincerely thank you for recognizing the novelty and translational potential of our pH-responsive, ROS-catalytic high-entropy alloy nanozymes. In response to the reviewers’ constructive comments, we have carried out additional experiments, refined data interpretation, and thoroughly revised the manuscript and Supplementary Information to address all concerns. New data and analyses have been incorporated where necessary to reinforce our conclusions. We believe that these substantial revisions have further strengthened the rigor, clarity, and overall impact of our work. We

greatly appreciate your valuable time and insightful feedback, which have been instrumental in improving the quality of our study.

(1) Since the high-entropy alloys (PtFeCuCoNi) are based on different elements, I am wondering how the precursor ratios affect HEA NP crystal structure and enzyme-like activity. This will validate the optimality and improve the reproducibility of the synthesis protocol. Has the authors tried with different ratios? Maybe more discussion is needed in this part.

Response to comment:

Thank you for your valuable comments. We acknowledge the limited data in our research on how precursor ratios affect the HEA NP crystal structure and enzyme-like activity. In response to your feedback, we have prepared HEA NPs with different precursor ratios (Pt₂FeCuCoNi, PtFe₂CuCoNi, PtFeCu₂CoNi, PtFeCuCo₂Ni, PtFeCuCoNi₂), and explored their crystal structures and enzymatic catalytic properties. It was found that the HEA synthesized with an equimolar precursor ratio exhibited the strongest XRD diffraction peaks and the optimal enzyme-mimicking performance. We appreciate your thoughtful input, which has significantly informed our work. The corresponding details can be found in the revised manuscript and revised supplementary information, as also shown below:

Page 6 in the revised manuscript: “We also investigated the effect of precursor ratio on the crystal structure of the HEA NPs (Supplementary Fig. 3). It was observed that the PtFeCuCoNi HEA NPs formed from an equimolar precursor ratio exhibited the most intense XRD diffraction peaks. This is attributed to the near-random distribution of the five constituent elements among the *fcc* lattice sites, which allows the crystal to maintain superior periodicity and thereby generate strong XRD diffraction peaks. Conversely, when the proportion of any single element was doubled, the excessive metal atoms introduced more intense and localized lattice expansion and strain, thereby disrupting the periodic arrangement of the crystal lattice and ultimately leading to a significant reduction in XRD diffraction peak intensity³³.”

Supplementary Fig. 3 X-ray diffraction (XRD) of HEA NPs with different precursor ratios.

References:

33 Liu, Y. H. et al. Toward controllable and predictable synthesis of high-entropy alloy. nanocrystals. *Sci. Adv.* 9, eadf9931 (2023).

Page 10 in the revised manuscript: “It was found that the HEA synthesized with an equimolar precursor ratio exhibited the optimal enzyme-mimicking performance (Supplementary Fig. 14). This is attributed to the optimal *d*-band center and lattice structure of the equimolar PtFeCuCoNi HEA NPs, which strikes an optimal balance between the adsorption and desorption of reactants³⁹.”

Supplementary Fig. 13 a) POD-like activity, b) Time-dependent CAT-like activity via TiSO₄-based UV-vis spectra in the presence of biocatalytic materials and H₂O₂, and c) SOD-like activity of HEA NPs with different

precursor ratios (n=3 independent experiments, data are presented as mean values \pm SD). Source data are provided as a Source Data file.

References:

39 Yao, Y. G. et al. High-entropy nanoparticles: Synthesis-structure-property relationships and data-driven discovery. *Science* 376, eabn3103 (2022).

(2) Will these materials also show SOD- and CAT- activities under acidic conditions? It is suggested to supplement more information on these catalytic activities to further verify the precision and response thresholds of pH gating effects.

Response to comment:

Thank you for your valuable and constructive comments for enhancing the quality of our manuscript. In response to your suggestions, we supplemented the CAT performance under acidic conditions. However, because SOD was tested with DMSO solvent and its pH could not be modulated, we used CAT to assess its ROS-scavenging ability under acidic conditions. Your feedback has greatly strengthened our paper, and we appreciate your thoughtful review. The corresponding details can be found in the revised manuscript and revised supplementary information, as shown below:

Page 10 in the revised manuscript: “Importantly, PtFeCuCoNi HEA NPs exhibited pH-dependent ROS-production and ROS-scavenging activities, with a predominance of ROS production in acidic environments and a predominance of ROS scavenging in neutral conditions, confirming its excellent microenvironmental adaptability (Supplementary Fig. 15).”

Supplementary Fig. 15 PtFeCuCoNi NPs show **a and b** peroxidases (POD)-mimetic and **c** catalase (CAT) activities in a pH-dependent manner ($n = 3$ independent experiments, data are presented as mean \pm SD). a.u. indicates the arbitrary units. Source data are provided as a Source Data file.

(3) The description on page 9, first paragraph, does not correspond to Figure 3f and 3g.

Response to comment:

We appreciate the reviewer's insightful comment. Your careful examination is greatly appreciated, and the updated corrections are included in the revised manuscript, as also illustrated below:

Page 10 in the revised manuscript: "Subsequently, the ¹O₂ species can be detected by the 9,10-

diphenanthraquinone (DPA) in a time-dependent manner (Fig. 3f). In addition, adopting 5,5-dimethyl-1-pyrroline N-oxide (DMPO) and 2,2,6,6-tetramethylpiperidine (TEMP) as the specific spin trap reagents, respectively, the electron paramagnetic resonance (EPR) detection also confirms that the major ROS in HEA NPs is $\cdot\text{O}_2^-$ and $^1\text{O}_2$ (Fig. 3g and Supplementary Fig. 9).”

(4) The catalytic activity and stability of HEA NPs over multiple reaction cycles require further investigation.

Response to comment:

Thank you for your good comments and helpful suggestions. We have supplemented the catalytic activity and stability of HEA NPs over multiple reaction cycles. The corresponding supporting literature has been added in the revised manuscript, as also shown below:

Page 10 in the revised manuscript: “The long-term activity and stability of PtFeCuCoNi HEA NPs as a CAT-like antioxidant have also been evaluated, revealing consistent maintenance of high activity with no observable decline after five cycles of testing (Supplementary Fig. 13).”

Supplementary Fig. 13 Cyclic stability test of H_2O_2 decomposition of PtFeCuCoNi HEA NPs.

(5) The writing could be improved. For example: The horizontal axis of Figure 7b. The $p(\text{H}_2\text{O}_2)$ values in Figures 7 and 9. The large or lowercase of first letter of the references title should be unified.

Response to comment:

Thank you for your valuable comments. In response, we have further refined the text throughout the manuscript, including standardizing the horizontal axis label in Figure 7b, clarifying the $p(\text{H}_2\text{O}_2)$ annotations in Figure 9 with explanations and relevant references, and unifying the capitalization of the first letters in reference titles to ensure consistent formatting and clear presentation.

1) The horizontal axis label in Figure 7b has been corrected and standardized.

Fig. 7 b3 Quantitative analysis of DCFH-DA fluorescence across various cell types subjected to distinct treatments.

2) The $p(\text{H}_2\text{O}_2)$ values in Figures 7 and 9 have been clarified with appropriate annotations and explanatory notes in the figure captions. The entire content following “ p ” is already presented in subscript format, and the number “2” in “ H_2O_2 ” cannot be further subscripted due to typographical constraints. This presentation is consistent with conventions used in prior publications (Nat. Commun., 2024, 15, 9592; Nat. Commun., 2025, 16, 856).

3) The large or lowercase of the first letter of the reference title has been unified. We have identified the first letter of the title of the reference as larger and the rest as lowercase.

(6) In Supplementary Fig. 38, the authors provided the detection data of the inflammatory factor IL-1 β at the skin defect site on day 10, which supports the claim that the PtFeCuCoNi HEA NPs reduce tissue inflammation during the late stage of infection. However, data on systemic inflammation are lacking and should be supplemented with relevant experiments.

Response to comment:

We sincerely thank the reviewer for this insightful comment. We fully agree that assessment of systemic inflammatory responses is critical for comprehensively evaluating the therapeutic efficacy of the material. Accordingly, we have supplemented the revised manuscript with peripheral blood analyses from rats at day 12 post-treatment. The results show that neutrophil (NEU) and white blood cells (WBC) counts in the HEA NP-treated group were comparable to those in healthy controls, indicating substantial attenuation of systemic inflammation. In addition, the lymphocyte (LYM) percentage was higher and approached normal levels, suggesting recovery of immune function and restoration of immune homeostasis. These findings further corroborate that PtFeCuCoNi HEA NPs effectively alleviate both local and systemic inflammatory responses during infection resolution (Supplementary Fig. 55).

Page 29 in the revised manuscript: “Moreover, peripheral blood analysis on day 12 revealed that neutrophil and white blood cell counts in the HEA NPs-treated rats were comparable to those of healthy controls, indicating attenuation of systemic inflammation. In addition, lymphocyte levels were higher and close to normal, suggesting recovery of immune homeostasis (Supplementary Fig. 55).”

Supplementary Fig. 55 The number of WBC (White Blood Cell), NEU (Neutrophil), and the percent of LYM (Lymphocyte) in the peripheral blood of rats on day 12. $n = 3$ independent replicates. Data are presented as means \pm SD, ns, not significant, $*p < 0.05$, $**p < 0.01$; one-way ANOVA with multiple comparisons test, all tests were two-sided. Source data are provided as a Source Data file.

(7) The antimicrobial section is comprehensive, addressing both planktonic and biofilm MRSA. However, the opening paragraph of the Discussion could be further strengthened. The statement “Infections caused by antibiotic-resistant pathogens represent an escalating clinical challenge due to antibiotic treatment failure, compounded by complex inflammation and dynamic microenvironmental alterations.” would be more compelling if expanded with relevant examples of severe infections caused by antibiotic-resistant bacteria (e.g., endocarditis, intra-abdominal infections, or bacteremia). This would situate the work within the broader clinical landscape.

Response to comment:

We followed the reviewer's advice and agree that incorporating representative examples of severe antibiotic-resistant infections would better contextualize our study within the broader clinical framework. Accordingly, we have revised the opening paragraph of the discussion to include specific examples, such as chronic wound infections, pulmonary infections, intra-abdominal infections, urinary tract infections, and infective endocarditis, which underscore the serious clinical impact and therapeutic challenges posed by multidrug-resistant pathogens. The revised text now reads as follows:

Page 32 in the revised manuscript: “Infections caused by antibiotic-resistant pathogens represent an escalating clinical challenge due to antibiotic treatment failure. They can lead to hard-to-heal wound infections¹¹⁰, urinary tract infection¹¹¹, pulmonary infections, intra-abdominal infections¹¹², and infective endocarditis¹¹³. Compounded by complex inflammation and dynamic microenvironmental alterations, these factors greatly increase the difficulty of clinical management.”

References:

110 Holubnycha, V. M. & Kholodylo, O. V. War impact on antimicrobial resistance and

- bacteriological profile of wound infections in Ukraine. *Commun. Med-London* **5**, 394 (2025).
- 111 Shang, L. M. *et al.* Nanozyme-reinforced hydrogel coatings for prevention of catheter-associated urinary tract infection. *Nano Today* **56**, 102271 (2024).
- 112 Zhu, J. Y. *et al.* Lactate promotes invasive *Klebsiella pneumoniae* liver abscess syndrome by Increasing capsular polysaccharide biosynthesis via the PTS-CRP axis. *Nat. Commun.* **16**, 6057 (2025).
- 113 Pongbangli, N., Chaiwarith, R., Phrommintikul, A. & Wongcharoen, W. Trends in infective endocarditis over two decades in a Thai tertiary care setting. *Sci. Rep.* **15**, 13746 (2025).

Reviewer #2 (Remarks to the Author):

“This manuscript reports the design of bioinspired Janus artificial enzymes based on high-entropy alloys (PtFeCuCoNi) for sequential therapy of drug-resistant bacterial infections and inflammatory wounds. The study is well-motivated, drawing on the biphasic microenvironment of chronic wounds, and demonstrates that the proposed HEA biocatalysts can achieve pH-dependent switching between oxidase/peroxidase-like ROS generation under acidic conditions and antioxidant-like scavenging at neutral pH. The combination of theoretical calculations and experimental analyses provides a coherent picture of the catalytic mechanism, and the in vivo data indicate antimicrobial efficacy against MRSA as well as immunomodulatory and pro-regenerative effects. The concept of exploiting multi-metal synergy in HEAs for redox modulation is novel and of potential interest for the field of nanomedicine. The manuscript is scientifically sound and is suitable for publication in Nature Communications. I therefore recommend its rapid publication after the following minor revisions are addressed:”

Response to the general comment:

Thank you for the outstanding summary and encouraging feedback on our manuscript. We have carefully revised the manuscript and supplementary materials, addressing all queries and concerns in detail. We greatly appreciate your valuable guidance, which has helped us refine the study, and we believe these revisions have further strengthened the paper's quality and impact.

(1) In the Supplementary Methods, it was mentioned that “Raman spectra were examined via the HORIBA HR Evolution Raman spectrometer with a 532 nm laser source”, but this characterization technique was not included in the main manuscript. Is it a mistake?

Response to comment:

We sincerely thank the reviewer for identifying the errors in our supporting information. We have corrected and deleted it, and added new instruments for structural characterization. Thanks again for your kind reminder, and the updated corrections are included in the revised supporting information, as also illustrated below:

Page 53 in the revised supporting information: “Inductively coupled plasma-Mass Spectrometry (ICP-MS) was carried out using a Agilent 7850 (Agilent Technologies, CA, USA). In-situ FTIR measurements were conducted using an infrared spectrometer (Thermo Scientific, iS50 FTIR) equipped with an in-situ spectrum cell (Shanghai Yuanfang Technology Co., Ltd., SPECCEL-III).”

(2) The synthesis methods for the mentioned FeCuCoNi, PtCuCoNi, PtFeCoNi, PtFeCuNi, and PtFeCuCo alloys are not provided in the manuscript, which should be supplemented.

Response to comment:

Thanks for your important and helpful comments on improving the quality of our manuscript. We have added to the problem, and the updated corrections are included in the manuscript, as also illustrated below:

Page 34 in the revised manuscript: “The preparation conditions of FeCuCoNi, PtCuCoNi, PtFeCoNi, PtFeCuNi, and PtFeCuCo NPs are similar to that of PtFeCuCoNi-HEA NPs except that one of the metal precursors is absent.”

(3) While the study highlights the advantages of HEA materials, the current text provides limited discussion of prior biomedical research on HEAs. The authors are encouraged to expand the Introduction or Discussion with additional references and a more explicit consideration of potential toxicity concerns associated with high-entropy alloys.

Response to comment:

Thank you for your valuable suggestion regarding the discussion of previous biomedical studies and potential toxicity concerns of high-entropy alloys (HEAs). We will address your comments in detail from the following two aspects: 1) prior biomedical research on HEAs; 2) evaluation of cytotoxicity.

1) Prior biomedical research on HEAs:

Page 32 in the revised manuscript: “Various high-entropy alloys have been explored for antibacterial applications. For example, MnFeCoNiCu nanozymes showed antibacterial activity against *S. enteritidis* and *E. coli* at 40 µg/mL¹¹⁴; the Al_{0.6}CoCrCu_{0.1}FeNiSi_{0.2} alloy, designed as a low-copper AHEA, achieved complete *E. coli* elimination by day 3¹¹⁵; Zhou et al. reported a Cu-Ag HEA with strong antibacterial effects, though the high cost of Ag may limit its broader application¹¹⁶. In summary, most previously reported HEAs exhibit single-mode antibacterial activity and lack the multifunctional or biocompatible properties required for effective biomedical use.”

References:

- 114 Feng, J. X. *et al.* Transition metal high-entropy nanozyme: multi-site orbital coupling modulated high-efficiency peroxidase mimics. *Adv. Sci.* **10**, 2303078 (2023).
- 115 Chen, S. Y. *et al.* A novel low copper content antibacterial high-entropy alloy: Role of synergistic effect of multiple elements. *J. Mater. Res. Technol.* **38**, 581-596 (2025).

116 Zhou, E. Z. *et al.* A novel Cu-bearing high-entropy alloy with significant antibacterial behavior against corrosive marine biofilms. *J. Mater. Sci. Technol.* **46**, 201-210 (2020).

2) Evaluation of cytotoxicity: To comprehensively evaluate the biosafety of PtFeCuCoNi HEA nanoparticles, we conducted both in vitro and in vivo assessments. In vitro cytocompatibility and hemolysis assays demonstrated negligible toxicity toward HUVECs, RSFs, and Raw264.7 cells at concentrations up to $4 \mu\text{g}\cdot\text{mL}^{-1}$, and no significant hemolysis even at $64 \mu\text{g}\cdot\text{mL}^{-1}$. In vivo, peripheral blood biochemical analysis, including alanine aminotransferase (ALT), aspartate aminotransferase (AST), urea (UREA), and creatinine (CREA), revealed no abnormalities compared with untreated controls (Supplementary Fig. 60). Histopathological examination of major organs (heart, liver, spleen, lung, and kidney) by H&E staining also showed normal tissue architecture at the therapeutic dose. Collectively, these findings confirm that PtFeCuCoNi HEA nanoparticles possess excellent biocompatibility in addition to potent antibacterial efficacy, highlighting their potential for safe and effective biomedical applications.

Page 29 in the revised manuscript: “Major-organ histology, hemolysis test, and serum biochemistry showed no adverse effects following HEA NPs administration (Supplementary Figs. 58, 59, 60).”

Supplementary Fig. 60 Biochemical analysis of the rat serum after treatments with different samples. $n = 3$ independent replicates. Data are presented as means \pm SD; ns, not significant; * $p < 0.05$, ** $p < 0.01$; one-way ANOVA with multiple comparisons test; all tests were two-sided. Source data are provided as a Source Data file.

(4) In Supplementary Fig. 28, the cytocompatibility of PtFeCuCoNi HEA NPs is presented. However, as Pt is a key constituent metal, the cytotoxicity of Pt NPs alone should also be discussed, which would strengthen the argument that the alloy formulation indeed reduces toxicity relative to single-metal systems.

Response to comment:

We took the advice and have supplemented the cytocompatibility evaluation of monometallic Pt NPs under identical experimental conditions. The additional results show that Pt NPs exhibited no statistically significant effect on the viability of HUVEC and RSF at concentrations up to $10 \mu\text{g}\cdot\text{mL}^{-1}$, and no notable cytotoxicity toward Raw264.7 cells at concentrations up to $4 \mu\text{g}\cdot\text{mL}^{-1}$ (Supplementary Figs. 40, 41). These data indicate that both PtFeCuCoNi HEA NPs and Pt NPs maintain excellent cytocompatibility at concentrations below $4 \mu\text{g}\cdot\text{mL}^{-1}$. The revised manuscript now reads as follows:

Page 23 in the revised manuscript: “Cytocompatibility testing showed no appreciable toxicity of PtFeCuCoNi HEA NPs or Pt NPs to human umbilical vein endothelial cells (HUVEC), rat skin fibroblasts (RSF), or murine macrophages (Raw264.7) up to $4 \mu\text{g}\cdot\text{mL}^{-1}$ (Supplementary Fig. 39-41)”

Supplementary Fig. 40 Viability of human umbilical vein endothelial cells (HUVEC), rat skin fibroblasts (RSF), and murine mononuclear macrophage cells (Raw264.7), after incubation with Pt ($n = 6$ independent biological replicates). Data are presented as means \pm SD, * $p < 0.05$, ** $p < 0.01$, *** $p < 0.001$, ns, not significant; one-way ANOVA with multiple comparisons test, all tests were two-sided. Source data are provided as a Source Data file.

Supplementary Fig. 41 Representative Live/Dead dual-stained fluorescence images of human umbilical vein endothelial cells (HUVEC), rat skin fibroblasts (RSF), and murine macrophages (Raw264.7) after incubation with Pt NPs.

(5) In Fig. 6, the transcriptional responses provide valuable mechanistic insight. However, the current text would benefit from the inclusion of supporting references for some of the highlighted pathways.

Including key citations would help place these findings within the broader context of bacterial stress responses.

Response to comment:

Thank you for the helpful suggestions. We have incorporated additional references in the revised manuscript to strengthen the interpretation of the transcriptional mechanisms and contextualize the observed pathways within the broader framework of bacterial stress responses. The corresponding citations and revised text are provided below:

Page 21 in the revised manuscript: “Furthermore, Kyoto Encyclopedia of Genes and Genomes (KEGG) analysis revealed broad perturbation of metabolic networks, with repression of sulfur metabolism⁷¹, cysteine⁷² and methionine metabolism⁷³, histidine metabolism⁷⁴, and glycerophospholipid metabolism⁷³, indicating disruption of thiol-based antioxidant defenses and membrane-lipid homeostasis (Fig. 6d). Notably, gene set enrichment analysis (GSEA) further showed positive enrichment of “oxidative phosphorylation” and “ATP-dependent activity”⁷⁵, suggesting an early compensatory surge in energy metabolism, likely reflecting transient activation of bacterial respiratory and ATP-generating pathways in response to acute ROS stress^{42,76}. In contrast, “ATP-binding cassette (ABC) transporters”⁷⁷ and “histidine metabolism”⁷⁸ were negatively enriched (Fig. 6e), indicating impaired substrate trafficking and amino-acid biosynthesis¹².”

References:

- 71 Seregina, T. A., Lobanov, K. V., Shakulov, R. S. & Mironov, A. S. Enhancement of the Bactericidal Effect of Antibiotics by Inhibition of Enzymes Involved in Production of Hydrogen Sulfide in Bacteria. *Mol. Biol.* **56**, 638-648 (2022).
- 72 Tomé, D. Amino acid metabolism and signalling pathways: potential targets in the control of infection and immunity (vol 23, pg 1, 2021). *Eur. J. Clin. Nutr.* **75**, 1319–1327 (2021).
- 74 Bem, A. E. *et al.* Bacterial histidine kinases as novel antibacterial drug targets. *ACS Chem. Biol.* **10**, 213-224 (2015).
- 75 Fisher, G. *et al.* Allosteric rescue of catalytically impaired ATP phosphoribosyltransferase variants links protein dynamics to active-site electrostatic preorganisation. *Nat. Commun.* **13**,

7607 (2022).

- 76 Bie, L. Y. *et al.* Comparative analysis of transcriptomic response of escherichia coli K-12 MG1655 to nine representative classes of antibiotics. *Microbiol. Spectr.* **11**, e00317-00323 (2023).
- 77 Rees, D. C., Johnson, E. & Lewinson, O. ABC transporters: the power to change. *Nat. Rev. Mol. Cell Bio.* **10**, 218-227 (2009).
- 78 Hoyos-Manchado, R. *et al.* RNA metabolism is the primary target of formamide. *Sci. Rep.* **7** (2017).

Reviewer #3 (Remarks to the Author):

“This manuscript describes the development of high-entropy alloy (HEA) (PtFeCuCoNi) as bioinspired nanocatalysts for the treatment of infected and non-healing wounds. The work addresses an important clinical challenge by targeting both the antimicrobial and tissue-regenerative phases of wound healing. The authors provide convincing evidence that the HEA nanozymes exhibit robust catalytic adaptability, functioning as ROS-generating oxidase/oxidase mimics in acidic environments while switching to ROS-scavenging antioxidant activity at neutral pH. The integration of transcriptomic profiling, mechanistic studies, and in vivo validation provides strong support for the dual regulatory role of these materials in modulating oxidative stress and inflammation. The concept of tailoring redox activity through entropy-driven alloy design is innovative and of clear significance to the biomedical nanomaterials community. It is suggested that this study is well executed and suitable for publication in Nature Communications, pending minor revisions to

improve clarity and contextual discussion.”

Response to the general comment:

We sincerely thank the reviewer for the positive evaluation and thoughtful feedback. We are grateful for your recognition of the clinical significance and scientific innovation of our work on bioadaptive high-entropy alloy (HEA) nanozymes. We have conducted additional systematic experiments and made extensive revisions throughout the manuscript to improve clarity, precision, and contextual discussion. All essential data have been included to support our conclusions, and each point has been carefully addressed in both the main text and the Supplementary Information. We believe that these revisions have further strengthened the scientific rigor and overall quality of the work, and we appreciate your constructive review and recommendation for publication.

(1) The mechanism for the synthesis of the PtFeCuCoNi HEA NPs via the hydrothermal method should be discussed in the main manuscript.

Response to comment:

We are grateful to the reviewers for this valuable suggestion and We are very grateful for your comments and have supplemented the synthesis mechanism of the HEA NPs prepared via the hydrothermal method in the manuscript, as also illustrated below:

Page 5 in the revised manuscript: “The HEA-based Janus artificial enzymes were synthesized via a one-step solvothermal approach using Pd(acac)₂, Fe(acac)₃, Cu (NO₃)₂·3H₂O, Co (NO₃)₃·6H₂O, Ni (NO₃)₃·6H₂O as metal precursors (Fig. 2a), *N,N*-dimethylformamide (DMF) and ethylene glycol (EG) as the solvents. A mauve, finely dispersed mixture of the raw materials was formed through ultrasonication. As the temperature increased, the mixture turned black, possibly due to the coordination and reduction of the five metals. Subsequently, the single-phase HEA formed after the diffusion and rearrangement of atoms.”

(2) Some earlier references regarding the HEA materials for antibacterial studies should be carefully compared and discussed included in the manuscript.

Response to comment:

Thank you for these constructive suggestions. As mentioned above, we have incorporated additional references on previous studies of HEA materials with antibacterial properties to provide clearer context and a comparison with our work. The corresponding citations and discussion have been added in the revised manuscript, as shown below:

Page 32 in the supporting information: “Various high-entropy alloys have been explored for antibacterial applications. For example, MnFeCoNiCu nanozymes showed antibacterial activity against *S. enteritidis* and *E. coli* at 40 µg/mL¹¹⁴; the Al_{0.6}CoCrCu_{0.1}FeNiSi_{0.2} alloy, designed as a low-copper AHEA, achieved complete *E. coli* elimination by day 3¹¹⁵; Zhou et al. reported a Cu-Ag HEA with strong antibacterial effects, though the high cost of Ag may limit its broader application¹¹⁶. In summary, most previously reported HEAs exhibit single-mode antibacterial activity and lack the multifunctional or biocompatible properties required for effective biomedical use.”

References:

- 114 Feng, J. X. *et al.* Transition Metal High-Entropy Nanozyme: Multi-Site Orbital Coupling Modulated High-Efficiency Peroxidase Mimics. *Adv. Sci.* **10**, 2303078 (2023).
- 115 Chen, S. Y. *et al.* A novel low copper content antibacterial high-entropy alloy: Role of synergistic effect of multiple elements. *J. Mater. Res. Technol.* **38**, 581-596 (2025).
- 116 Zhou, E. Z. *et al.* A novel Cu-bearing high-entropy alloy with significant antibacterial behavior against corrosive marine biofilms. *J. Mater. Sci. Technol.* **46**, 201-210 (2020).

(3) The contents of the elements measured by XPS and ICP in the manuscript are inconsistent. Please check the data and explain the difference.

Response to comment:

Thank you for your attentive review and valuable feedback. We have carefully checked the data

and earlier literature. The discrepancy arises from the inherent differences between XPS (a surface-sensitive technique) and ICP (a bulk analysis technique). The most probable cause is the surface layer composition of the nanoparticles, different from the bulk composition. We have double-checked our ICP data and are confident in its reliability. A brief explanation to this effect has been added to the manuscript.

(4) In Fig. 6f, an interesting mechanistic illustration is presented; however, the results differ from previous sequencing studies on active bactericidal processes, where a decrease in ATP activity was observed. The authors are encouraged to further explain the reasons for this discrepancy and provide relevant literature support.

Response to comment:

We sincerely appreciate the reviewer's constructive comment. We fully agree that the expression pattern of ATP synthase and oxidative phosphorylation-related genes reflects the metabolic state of bacteria under oxidative stress, and we thank you for highlighting this important mechanistic aspect.

In the study of Co–Ru atomic pair catalysts (APCR) (*Adv. Mater.* 2024, 36, 2408787), transcriptomic analysis revealed the suppression of oxidative phosphorylation and disruption of redox homeostasis, consistent with the metabolic exhaustion observed during the late bactericidal phase. In contrast, our transcriptomic profiling was performed at an early stage after HEA treatment, when a portion of bacteria remained metabolically active and were undergoing adaptive responses to HEA-catalyzed ROS stimulation, rather than irreversible oxidative damage.

At this early phase, bacteria typically initiate a transient compensatory metabolic activation to sustain energy production and counteract oxidative imbalance. This transient upregulation of ATP synthesis and oxidative phosphorylation has been reported in several studies. For instance, a comparative transcriptomic analysis of *E. coli* exposed to nine classes of antibiotics demonstrated that genes involved in oxidative phosphorylation, such as NADH: quinone oxidoreductase I,

cytochrome oxidases, and ATP synthase, were upregulated under ciprofloxacin, erythromycin, and tetracycline treatment (*mBio* 2022, 13, e03645-21), suggesting that early oxidative stress may trigger enhanced ATP production as a compensatory mechanism. Similarly, a ribosome profiling study under acidic stress showed significant upregulation of ATP-dependent copper-exporting enzymes (CopA and CusA) (*mSystems* 2021, 6, e01234-20), reflecting increased ATP-dependent activity to maintain metal ion homeostasis and survival under stress conditions.

Therefore, the differences among these studies likely arise from the temporal dynamics of bacterial energy metabolism. During the early phase of oxidative or acidic stress, bacteria exhibit metabolic activation characterized by transient upregulation of ATP-dependent pathways and oxidative phosphorylation, whereas in the later energy-depletion phase, ATP production declines and metabolic collapse ensues, leading to cell death.

In summary, our findings most likely capture the early adaptive phase of bacterial response to HEA-induced ROS, while the APCR study reflects the late metabolic exhaustion stage. This interpretation suggests that bacterial energy metabolism under oxidative stress is dynamic and time-dependent, and our results provide complementary insight into the temporal evolution of stress-responsive metabolism, representing a novel mechanistic observation in ROS-mediated antibacterial research.

Page 21 in the revised manuscript: “Notably, gene set enrichment analysis (GSEA) further showed positive enrichment of “oxidative phosphorylation” and “ATP-dependent activity”⁷⁵, suggesting an early compensatory surge in energy metabolism, likely reflecting transient activation of bacterial respiratory and ATP-generating pathways in response to acute ROS stress^{42,76}, while “ATP-binding cassette (ABC) transporters”⁷⁷ and “histidine metabolism”⁷⁸ were negatively enriched (Fig. 6e), indicating impaired substrate trafficking and amino-acid biosynthesis¹².”

References:

- 42 Gao, Y. et al. Infectious and inflammatory microenvironment self-adaptive artificial peroxisomes with synergetic Co-Ru pair centers for programmed diabetic ulcer therapy. *Adv. Mater.* 36, 2408787 (2024).

76 Bie, L. Y. *et al.* Comparative analysis of transcriptomic response of *Escherichia coli* K-12 MG1655 to nine representative classes of antibiotics. *Microbiol. Spectr.* **11**, e00317-00323 (2023).

(5) Fig. 7n presents a schematic illustration of cellular effects under high-concentration H_2O_2 treatment and the protective role of HEA NPs. However, the labeling of the cell types is missing, making it unclear which three cells are being represented.

Response to comment:

Thank you for pointing this out. We apologize for the omission. In Fig. 7n, the three representative cell types correspond to vascular endothelial cells (VEC), anti-inflammatory macrophages (M2), and rat skin fibroblasts (RSF), which are the principal cell populations involved in anti-inflammation regulation, angiogenesis, and tissue remodeling during wound healing. The corresponding labels have now been added, and the schematic illustration has been updated in the revised figure to improve clarity and interpretability.

Fig. 7(n) Schematic illustration of cell effects upon the addition of high-concentration H_2O_2 and HEA NPs' protection.

(6) Fig. 8d and Fig. 8f present GO and KEGG analyses, respectively. It is noted that Fig. 8d compares PtFeCuCoNi vs H₂O₂, while Fig. 8f shows the opposite comparison, H₂O₂ vs PtFeCuCoNi. As the rationale for this discrepancy has not been explained in the manuscript, the authors are advised to provide further discussion.

Response to comment:

Thank you for pointing this out. The inconsistency in the comparison directions of the GO and KEGG analyses arises from our intention to capture two complementary biological dimensions—the protective mechanisms activated by the HEA NPs and the molecular injuries caused by oxidative stress. Specifically, the GO enrichment analysis (PtFeCuCoNi vs. H₂O₂) was designed to highlight the functional restoration and metabolic protection afforded by the HEA nanozyme, focusing on biological processes associated with cellular homeostasis, cytoskeletal organization, and angiogenic repair. In contrast, the KEGG analysis (H₂O₂ vs PtFeCuCoNi) emphasized oxidative stress-induced molecular disruption, revealing pathways linked to apoptosis, inflammatory signaling, and energy imbalance. This bidirectional comparison framework provides a holistic “damage-protection” perspective, clarifying both the pathophysiological impact of oxidative stress and the restorative effects of HEA nanozyme.

To further strengthen this interpretation, we integrated the GSEA results (based on GO-BP gene sets) within the main text. Notably, “tube development” and “cytoskeleton organization” were significantly enriched in the PtFeCuCoNi group, indicating enhanced structural integrity and pro-regenerative activity. Conversely, “apoptotic signaling pathway” was suppressed compared with the H₂O₂ group, confirming the material’s anti-apoptotic and antioxidative functions. Enrichment of “cellular response to oxygen-containing compounds” further aligned with the ROS and cell viability data, validating the redox-protective behavior of the HEA nanozyme.

(7) Supplementary Figs. 31 and 32 show scratch assays of HUVECs and RSFs, respectively. However, unlike Figs. 7a and 7c, the scratch assay for RAW264.7 cells is missing. The authors are advised to either provide the corresponding experiment or explain the reason for its absence.

Response to comment:

We sincerely thank the reviewer for this careful and constructive comment. Scratch assays are primarily used to evaluate the migration and proliferative capacity of cells involved in tissue regeneration. In the context of wound healing, HUVECs and RSFs are the principal effector cells that drive angiogenesis and extracellular matrix remodeling. Therefore, we performed scratch assays for these cell types to assess the regenerative potential of the HEA nanoparticles (Supplementary Figs. 44, 45). In contrast, Raw264.7 macrophages primarily serve immunomodulatory rather than migratory roles during skin repair, orchestrating inflammatory resolution through M1/M2 polarization. Moreover, due to their growth characteristics, which require frequent subculture to maintain viability, they are unsuitable for conventional scratch assays, as prolonged culture after scratching often leads to proliferation arrest and apoptosis. Hence, migration assays were not performed for Raw264.7 cells.

To comprehensively assess the effect of PtFeCuCoNi HEA NPs on macrophage functionality, we instead conducted RT-qPCR analyses of M1 (iNOS) and M2 (Arg1) markers, along with cytokines (TNF- α , IL-6, IL-10, and TGF- β). The results revealed that the HEA NPs suppressed pro-inflammatory mediators while enhancing anti-inflammatory gene expression (Supplementary Figs. 47, 48). Together with the immunofluorescence data showing decreased CCR7 and increased CD163 expression (Fig. 7j–m and Supplementary Fig. 46), these findings indicate that the HEA nanoparticles effectively reprogram macrophages toward an anti-inflammatory phenotype.

Page 23 in the revised manuscript: “To evaluate macrophage polarization under lipopolysaccharide (LPS) stimulation, real-time quantitative polymerase chain reaction (RT-qPCR) was performed. Compared with Pt NPs, PtFeCuCoNi HEA NPs were found to exert a stronger inhibitory effect on the mRNA expression levels of pro-inflammatory cytokines, such as inducible nitric oxide synthase (iNOS), interleukin-6 (IL-6), and tumor necrosis factor-alpha (TNF- α), while concurrently augmenting genes associated with anti-inflammatory activity, including arginase-1 (Arg1), interleukin-10 (IL-10), and transforming growth factor-beta (TGF- β) (Supplementary Figs. 47, 48). Furthermore, immunofluorescence analysis revealed that HEA NP

pretreatment significantly reduced C-C chemokine receptor type 7 (CCR7) expression, a signature marker of M1 macrophages, relative to both LPS alone and Pt treatment (Fig. 7j, k and Supplementary Fig. 46).”

Supplementary Fig. 47 a The relative mRNA expression of iNOS, TNF- α , and IL-6 in Raw264.7 cells was measured by Real-time quantitative polymerase chain reaction (RT-qPCR) after 24 h of LPS stimulation ($n = 3$ independent biological replicates). **b** The relative mRNA expression of Arg-1, IL-10, and TGF- β in Raw264.7 cells following 24 h of LPS stimulation ($n = 3$ independent biological replicates). Data are presented as means \pm SD, * $p < 0.05$, ** $p < 0.01$, *** $p < 0.001$, **** $p < 0.0001$, ns, not significant; one-way ANOVA with multiple comparisons test, all tests were two-sided. Source data are provided as a Source Data file.

Supplementary Fig. 48 Schematic depicting the effects of PtFeCuCoNi HEA NPs on macrophage polarization. M0 indicates macrophages, M1 indicates pro-inflammatory macrophages, M2 indicates anti-inflammatory macrophages, and LPS indicates lipopolysaccharide.

Reviewer #4 (Remarks to the Author):

“Gao et al. designed a high-entropy alloy-based Janus artificial enzyme targeting chronic wound infections. This enzyme leverages the unique microenvironment characteristics at the infection site to catalyze antimicrobial and anti-inflammatory actions, demonstrating excellent antimicrobial efficacy both in vitro and in vivo. Additionally, the study found that it can promote the PFKFB3 pathway, inhibit the expression of inflammatory cytokines, and accelerate wound healing. The results are very interesting; however, the manuscript still has some issues to address. It is recommended that the authors make further additions and revisions.”

Response to comment:

We sincerely appreciate the reviewer’s recognition of our high-entropy alloy-based Janus artificial enzyme as a promising bioadaptive platform for infection control and tissue repair. In response to your insightful and constructive comments, we have conducted additional experiments and comprehensively revised the manuscript to address all the concerns raised. In particular, we have included systematic evaluations of material stability, ROS generation and lipid peroxidation at the bacterial level, EPS degradation, macrophage polarization and cytokine expression profiling, as well as RT-qPCR validation of transcriptomic results. All newly obtained data and corresponding discussions have been incorporated into the revised main text and Supplementary Information. We believe that these revisions have substantially strengthened the mechanistic depth, scientific rigor, and overall clarity of the manuscript. We are deeply grateful for your thoughtful feedback, which has greatly enhanced the quality, coherence, and impact of our work.

(1) Through TEM and SEM observations, the Janus artificial enzyme exhibits a nanostructure. The authors should further supplement the DLS and material stability data of the artificial enzyme, particularly regarding its bioavailability. It is also recommended that the authors include stability data of the artificial enzyme under different environmental conditions.

Response to comment:

Thank you for your insightful comments and helpful suggestions. In response to your feedback, we supplemented the PtFeCuCoNi HEA NPs with DLS. Additionally, the stability of the material was evaluated over a twelve-day period in four different environments: PBS (pH 4.5), PBS (pH 7.4), Dulbecco's Modified Eagle Medium (DMEM) cell culture medium, and Fetal Bovine Serum (FBS). The corresponding supporting literature has been added in the revised manuscript, as also shown below:

Page 6 in the revised manuscript: “Meanwhile, the number-averaged size of PtFeCuCoNi HEA NPs is measured to be ≈ 122.4 nm in dynamic light scattering, indicative of a stable aqueous phase with low agglomeration tendency. (Supplementary Fig. 2).”

Supplementary Fig. 2 Dynamic light scattering (DLS) of PtFeCuCoNi HEA NPs.

Page 16 in the revised manuscript: “Afterward, we rigorously evaluated the solution stability of HEA NPs. The PtFeCuCoNi HEA NPs exhibit exceptional dispersibility, with no detectable precipitation or phase separation following static incubation at room temperature for 12 days (Supplementary Fig. 29). Moreover, assessments of particle size and zeta potential of PtFeCuCoNi HEA NPs across various intervals (days 3, 6, 9, and 12) indicate no significant dimensional alterations or aggregation (Supplementary Fig. 30). Therefore, PtFeCuCoNi HEA NPs exhibit robust

stability and hold promise for enhanced therapeutic efficacy in forthcoming *in vivo* and *in vitro* studies.”

Supplementary Fig. 29 Digital photographs of PtFeCuCoNi HEA NPs in four physiologically relevant solvents: PBS (pH 4.5), PBS (pH 7.4), Dulbecco's Modified Eagle Medium (DMEM) cell culture medium, and fetal bovine serum (FBS).

Supplementary Fig. 30 Analysis of the long-term stability of PtFeCuCoNi HEA NPs in terms of zeta potential values and hydrodynamic diameter in four physiologically relevant solvents: **a** PBS (pH 4.5), **b** PBS (pH 7.4), **c** DMEM cell culture medium, and **d** FBS.

(2) The HEA nanoenzyme demonstrated different catalytic activities under acidic and neutral pH conditions, which is very interesting and differs from other reported HEA-based artificial enzymes. However, the underlying mechanism is not sufficiently discussed. Strengthening this part would improve the manuscript.

Response to comment:

We are grateful to the reviewers for this valuable suggestion and positive comment on our concept of pH-controlled bioadaptive ROS regulation. We agree that investigating the underlying reasons for the different catalytic activities observed in acidic and neutral environments is highly useful for further understanding catalytic mechanisms and validating the originality of our study. Further details will be provided in the subsequent discussion.

Density functional theory (DFT) calculations indicate that variations in hydrogen ion (H^+) concentration, particularly under acidic and neutral conditions, lead to distinct reaction pathways and products when the catalyst interacts with H_2O_2 . Initially, the Fe active site adsorbs an H_2O_2 molecule, leading to the release of H_2O and the subsequent formation of an $*O$ intermediate. In a neutral environment, H_2O_2 interacts with $*O$ to result in the creation of $*OH$ and $*OOH$, thereby exhibiting catalase-like activity. Conversely, in an acidic environment, H^+ exhibits a greater propensity to interact with $*O$ (with a Gibbs free energy barrier of -0.436 eV) compared to H_2O_2 (with a Gibbs free energy barrier of -0.384 eV), giving rise to $*OH$ intermediates. Subsequent re-adsorption of H_2O_2 leads to the generation of $*H_2O$ and $*OOH$, thereby showing POD-like activity. Notably, OOH is the precursor to produce $\bullet O_2^-$ and 1O_2 (*Adv. Mater.*, **2024**, 36, 2408787; *Nat. Commun.*, **2024**, 15, 1010; *J. Am. Chem. Soc.*, **2023**, 145, 8965-8978).

In summary, the proton source in neutral conditions is solely H₂O₂, leading the active sites to predominantly interact with H₂O₂ and display catalase-like activity, whereas under acidic conditions, the proton sources include both H⁺ and H₂O₂, and H⁺ has a higher affinity for the *O intermediates than H₂O₂, thus resulting in peroxidase-like activity. The corresponding mechanism of different catalytic activities triggered by acidic and neutral environments has been added in the revised manuscript and revised supplementary information, as also shown below:

Page 14 in the revised manuscript: “Furthermore, according to our previous studies, the structure of adsorbed O* is a key intermediate in compartmentalizing the POD-like and CAT-like pathways⁴². Therefore, to demonstrate the continuous pH-switchable catalytic activity of PtFeCuCoNi HEA NPs, we complemented the DFT studies of the free energy of the structure of PtFeCuCoNi HEA NPs adsorbing O* (Structure 1) for the adsorption of H⁺ or H₂O₂ under acidic and neutral conditions. As shown in Supplementary Fig. 20, the free energies of structure 1 for the adsorption of both H⁺ and H₂O₂ are negative, indicating that both processes can occur spontaneously. However, compared with adsorbing H₂O₂ (−0.384 eV), the free energy of adsorbing H⁺ by structure 1 is lower (−0.436 eV), indicating that structure 1 will preferentially combine with H⁺ to form *OH to complete the POD-like activity pathway in an acidic environment with the presence of H⁺. Furthermore, under neutral or slightly alkaline conditions, structure 1 is more prone to adsorb H₂O₂ to carry out the CAT-like activity pathway to realize the continuous pH-switchable catalytic activity.”

Supplementary Fig. 20 Free energy analysis of further adsorption of H^+ and H_2O_2 by PtFeCuCoNi HEA NPs O^* intermediates.

(3) Fig. 5C is used to detect bacterial protein leakage, typically assessed by the BCA assay. However, the manuscript does not provide a detailed description of this method. Given the limited bacterial protein content, the detection results showed a deep purple color and lacked a standard curve. It is recommended that the authors provide further clarification on this issue.

Thanks for this insightful comment, and we have added a detailed description of the bicinchoninic acid (BCA) assay used to quantify bacterial protein leakage in the revision (Fig. 5C). To ensure methodological transparency and data reliability, a standard curve was established using gradient concentrations of bovine serum albumin (BSA, 0-0.5 mg/mL) solutions. The corresponding optical density (OD_{562}) values were recorded and photographed, yielding a linear calibration

equation of $y = 1.698x + 0.166$ ($y = \text{OD}_{562}$, $x = \text{protein concentration, mg/mL}$) with $R^2 = 0.992$, confirming high linearity.

In addition, we have clarified that the “protein leakage (a.u.)” in the original figure represents relative OD values that reflect the extent of bacterial membrane disruption. Following the reviewer’s suggestion, all OD data have now been converted to absolute protein concentrations using the standard curve to achieve quantitative accuracy.

The observed deep purple coloration in the reaction system possibly resulted from the high bacterial load (10^8 CFU/mL, 1 mL total volume). Under these conditions, the strong ROS-mediated membrane disruption induced by the PtFeCuCoNi HEA NPs led to substantial cytoplasmic protein release, producing an intensified colorimetric response.

Page 35 in the revised manuscript: “**Antibacterial experiments.** Bacterial protein leakage was quantified using a bicinchoninic acid (BCA) protein assay kit (Solarbio). Briefly, *MRSA* were incubated with PBS, H_2O_2 , Pt + H_2O_2 , PtFeCuCoNi + H_2O_2 under the indicated conditions. After incubation, supernatants were collected by centrifugation, and protein content was determined following the manufacturer’s protocol. A standard curve was established using bovine serum albumin (BSA, 0-0.5 mg/mL), and the corresponding optical density at 562 nm (OD_{562}) was recorded. The calibration equation ($y = 1.698x + 0.166$, $R^2 = 0.992$) was used to convert OD values into absolute protein concentrations. Representative images of the gradient color reaction were also obtained. All measurements were performed in triplicate.”

Supplementary Fig. 23 a Representative image of Bicinchoninic Acid (BCA) reaction solutions with gradient bovine serum albumin (BSA) concentrations (0, 0.025, 0.05, 0.1, 0.2, 0.3, 0.4, 0.5 $\text{mg}\cdot\text{mL}^{-1}$), showing the increasing purple color intensity corresponding to measured OD_{562} values for standard curve quantification. **b** Standard curve of BSA for protein quantification using the BCA assay. Gradient concentrations of BSA (0, 0.025, 0.05, 0.1, 0.2, 0.3, 0.4, 0.5 $\text{mg}\cdot\text{mL}^{-1}$) were measured at OD_{562} , and the resulting data were fitted to obtain the standard curve ($y = 1.698x + 0.166$, $R^2 = 0.992$). **c** Protein leakage of *MRSA* after incubation with PBS, H_2O_2 , Pt + H_2O_2 , PtFeCuCoNi + H_2O_2 , quantified as absolute protein concentrations using the BSA standard curve from the BCA assay. ($n = 3$ independent replicates). Data are presented as means \pm SD; ns, not significant, *** $p < 0.001$; one-way ANOVA with multiple comparisons test, all tests were two-sided. Source data are provided as a Source Data file.

(4) The authors demonstrated the reduction of EPS through SEM characterization, but this claim has

certain limitations. It is recommended that the authors include a biofilm EPS gel electrophoresis experiment to further validate this result.

Response to comment:

We sincerely thank you for this valuable comment. As pointed out, relying solely on SEM to demonstrate EPS reduction indeed has certain limitations. The extracellular polymeric substances (EPS) of bacterial biofilms mainly consist of proteins, polysaccharides, and small amounts of extracellular DNA (eDNA), among which proteins and polysaccharides form the structural backbone that determines the stability and function of the biofilm. To systematically evaluate the destructive effects of PtFeCuCoNi nanozymes on *MRSA* biofilms, we employed multiple complementary analytical methods.

In our initial submission, we had already conducted SEM observation and Dextran Alexa Fluor 647 staining. SEM provides direct morphological information on the overall biofilm architecture and the spatial distribution of major EPS components, including proteins and polysaccharides. The results showed that biofilms treated with PtFeCuCoNi exhibited pronounced membrane roughening, localized collapse, and a substantial reduction in EPS coverage. In contrast, only mild changes were observed in the Pt-treated group. The control biofilms remained intact (Fig. 5j and Supplementary Fig. 26). Meanwhile, Dextran Alexa Fluor 647 staining, which specifically labels polysaccharides within the EPS matrix through its dextran-based structure that embeds into the polysaccharide network, allowed visualization of polysaccharide distribution under confocal microscopy. The fluorescence intensity of EPS polysaccharides was markedly reduced in the PtFeCuCoNi-treated group compared with the Control, H₂O₂, and Pt groups (Supplementary Fig. 34, 35), indicating a significant decrease in polysaccharide abundance.

Following the reviewer's suggestion, we further supplemented protein level analyses to strengthen the compositional evidence. SDS-PAGE analysis revealed that EPS proteins in the PtFeCuCoNi-treated samples were almost completely degraded, with only low-molecular-weight fragments remaining. In contrast, the Pt group showed partial degradation, while the control group

retained intact protein bands. In addition, we performed agarose gel electrophoresis to examine total DNA extracted from biofilm samples, which includes both extracellular DNA and a small fraction of intracellular DNA. The results showed that DNA bands in the PtFeCuCoNi-treated group were almost completely absent, with only faint signals near the loading wells; partial degradation was observed in the Pt group, whereas clear intact bands remained in the control group. These findings indicate that PtFeCuCoNi treatment severely disrupts the structural integrity of the biofilm, leading to oxidative degradation of both extracellular and intracellular components.

Collectively, the evidence from SEM morphology, Dextran Alexa Fluor 647 polysaccharide staining, and protein/DNA electrophoresis provides comprehensive, multi-level validation of the pronounced disruption of biofilm structure by PtFeCuCoNi. These results support the idea that ROS-mediated oxidative degradation of multiple biofilm components, including polysaccharides, proteins, and DNA, leads to the effective collapse of biofilm architecture and loss of integrity. This multi-method verification not only strengthens the logical completeness of our results but also fully addresses and supports the reviewer's valuable suggestion.

Page 16 in the revised manuscript: "Scanning electron microscopy (SEM) also revealed minor surface damage with abundant extracellular polymeric substances (EPS) in Pt-treated bacteria^{57,58}, whereas HEA-exposed group displayed marked membrane roughening, localized collapse, and visibly reduced EPS coverage (Fig. 5j and Supplementary Fig. 26). **Consistently, Dextran Alexa Fluor 647 staining showed markedly decreased fluorescence intensity of polysaccharides within the EPS after HEA treatment, confirming substantial matrix degradation (Supplementary Figs. 34, 35). Furthermore, sodium dodecyl sulfate polyacrylamide gel electrophoresis (SDS-PAGE) and agarose gel electrophoresis of EPS proteins and total DNA (including extracellular DNA) demonstrated extensive degradation following PtFeCuCoNi treatment, whereas Pt treatment induced only partial degradation, and controls remained largely intact (Supplementary Figs. 27, 28), corroborating the SEM observations.** TEM images revealed highly fragmented membranes and clear internalization of HEA NPs following incubation with H₂O₂ (Fig. 5k), indicative of a ROS-potentiated, membrane-disruptive bactericidal process."

Supplementary Fig. 27 Sodium dodecyl sulfate–polyacrylamide gel electrophoresis (SDS–PAGE) analysis of the proteins in the extracellular polymeric substances (EPS) of *MRSA* biofilm after incubating with PBS, Pt + H₂O₂, and PtFeCuCoNi + H₂O₂ ($n = 3$ independent replicates). Representative images are shown. Data are presented as means \pm SD, * $p < 0.05$, ** $p < 0.01$; one-way ANOVA with multiple comparisons test, all tests were two-sided. Source data are provided as a Source Data file.

Supplementary Fig. 28 Agarose gel electrophoresis analysis of DNA extracted from *MRSA* biofilm

after incubation with PBS, Pt + H₂O₂, and PtFeCuCoNi + H₂O₂ (n = 3 independent replicates). Representative gel images are shown. Data are presented as means ± SD, **p* < 0.05, ****p* < 0.001; one-way ANOVA with multiple comparisons test, all tests were two-sided. Source data are provided as a Source Data file.

(5) The authors demonstrated lipid peroxidation in *MRSA* after treatment with the artificial enzyme by the increase in LPO. It is recommended that the authors include MDA detection at the bacterial level to further confirm this phenomenon.

Response to comment:

We sincerely thank you for this insightful and constructive suggestion. We fully agree that detecting malondialdehyde (MDA), a well-recognized marker of lipid peroxidation, would provide more direct evidence of oxidative membrane damage in bacteria. Accordingly, we have performed an additional MDA assay to evaluate the lipid peroxidation levels in *MRSA* after different treatments (PBS, H₂O₂, Pt NPs, and PtFeCuCoNi HEA NPs).

As shown in Supplementary Fig. 38, the PtFeCuCoNi HEA NPs group exhibited a markedly elevated MDA content (≈0.045 nmol/10⁴ cells), which was significantly higher than the PBS, H₂O₂, Pt NPs, and PtFeCuCoNi HEA NPs groups. These findings clearly indicate that the HEA NPs induced stronger oxidative lipid damage in *MRSA*, corroborating our previous LPO results and further confirming the enhanced ROS-driven antibacterial mechanism of the material.

Page 17 in the revised manuscript: “Consistently, CLSM-based LPO analysis showed a strong green fluorescence shift in HEA NPs-exposed biofilms^{64,65}, particularly within deeper layers, suggesting extensive phospholipid oxidation and ROS-induced damage throughout the biofilm matrix (Fig. 5s-u and Supplementary Figs. 26, 27)⁶⁶⁻⁶⁸. In line with this, MDA quantification further confirmed enhanced lipid peroxidation in *MRSA* treated with PtFeCuCoNi HEA NPs compared with Pt NPs and controls (Supplementary Fig. 38).”

Supplementary Fig. 38 Malondialdehyde (MDA) levels in *MRSA* after different treatments. Quantitative analysis of MDA production in *MRSA* following treatment with PBS, H₂O₂, Pt NPs, or PtFeCuCoNi HEA NPs ($n = 3$ independent biological replicates). Data are presented as means \pm SD, *** $p < 0.001$, ns, not significant; one-way ANOVA with multiple comparisons test, all tests were two-sided. Source data are provided as a Source Data file.

(6) The antibacterial mechanism of the artificial enzyme primarily involves the generation of ROS. However, the manuscript lacks relevant experiments at the bacterial level. It is recommended that the authors include ROS generation at the bacterial level after labeling with the DCFH-DA probe.

Response to comment:

We sincerely appreciate your valuable suggestion to improve the quality of our manuscript. We understand your concern that the antibacterial mechanism of artificial enzymes mainly relies on the generation of reactive oxygen species (ROS), while the current version lacks direct experimental evidence at the bacterial level. To evaluate ROS generation at the bacterial level, we applied 2',7'-dichlorodihydrofluorescein diacetate (DCFH-DA) as a fluorescent probe to assess intracellular ROS production in *MRSA* under different treatment conditions. As shown in Supplementary Fig. 28, both the PBS and H₂O₂ groups exhibited weak green fluorescence, whereas the PtFeCuCoNi HEA NPs-treated group displayed markedly enhanced green fluorescence intensity, indicating substantial intracellular ROS accumulation. The Pt NP-treated bacteria also showed increased fluorescence,

though significantly weaker than that of the PtFeCuCoNi group. These findings demonstrate that under acidic conditions (pH = 4.5), PtFeCuCoNi HEA nanozymes effectively catalyze H₂O₂ decomposition to generate ROS, thereby elevating bacterial intracellular ROS levels and inducing oxidative damage. Our data provides direct evidence supporting the ROS-mediated antibacterial mechanism of the artificial enzyme. The corresponding details can be found in the revised manuscript and revised supplementary information, as shown below:

Page 16 in the revised manuscript: “In acidic media containing H₂O₂, the NPs predominantly produced •O₂⁻ and ¹O₂, whereas at near-neutral pH they exhibited no detectable POD-like activity (Fig. 5a). To verify ROS generation in bacteria directly, intracellular ROS levels were assessed using 2',7'-dichlorodihydrofluorescein diacetate (DCFH-DA) staining. Bacteria treated with PtFeCuCoNi HEA NPs exhibited strong green fluorescence, indicative of pronounced intracellular ROS accumulation and presenting direct evidence for the ROS-mediated antibacterial mechanism of the artificial enzyme (Supplementary Fig. 21).”

Supplementary Fig. 21 Top: Two-dimensional (2D) images from confocal laser scanning microscope (CLSM) images of 2',7'-dichlorodihydrofluorescein diacetate (DCFH-DA) fluorescence on the planktonic *MRSA* after incubating with PBS, H₂O₂, Pt + H₂O₂, PtFeCuCoNi + H₂O₂. The scale

bar represents 50 μm . Experiments were repeated independently three times with similar results. Bottom: Quantitative analysis of DCFH-DA fluorescence after different treatments ($n = 3$ independent replicates). Data are presented as means \pm SD; * $p < 0.05$, ** $p < 0.01$; ns, not significant; one-way ANOVA with multiple comparisons test; all tests were two-sided. Source data are provided as a Source Data file.

(7) At the cellular level, the artificial enzyme exhibits anti-inflammatory effects. The authors confirmed its ROS scavenging ability using the DCFH-DA probe; however, ROS scavenging is only one step in the anti-inflammatory process. It is recommended that the authors supplement the study with the polarization trends of M1 and M2 macrophages at the cellular level and concurrently assess the release of inflammatory cytokines TNF- α , IL-6, and IL-10 at the cellular level.

Response to comment:

Thanks so much for this insightful comment. As rightly noted, ROS scavenging constitutes only one aspect of the anti-inflammatory process, and a more comprehensive understanding requires evaluating macrophage polarization and inflammatory cytokine regulation at the cellular level. Accordingly, we performed additional experiments to assess macrophage polarization trends and inflammatory cytokine expression following treatment with PtFeCuCoNi HEA NPs (Supplementary Fig. 47). The results showed a clear phenotypic transition from the pro-inflammatory M1 type to the reparative M2 phenotype, as evidenced by significant downregulation of the M1 marker iNOS and upregulation of the M2 marker Arg1. In addition, secretion of pro-inflammatory cytokines (TNF- α , IL-6) was markedly decreased, whereas secretion of anti-inflammatory cytokines (IL-10, TGF- β) was significantly increased. These results were complemented by a schematic illustration summarizing the immunomodulatory effects of PtFeCuCoNi HEA NPs on macrophage polarization and cytokine balance (Supplementary Fig. 48).

Page 23 in the revised manuscript: “In parallel, CD163 expression was strongly upregulated (Fig. 7l, m), suggesting a phenotypic shift from a pro-inflammatory M1 phenotype toward a reparative M2 state. Furthermore, RT-qPCR analysis revealed that, compared to Pt NPs, PtFeCuCoNi HEA NPs exerted a stronger inhibitory effect on the mRNA expression of pro-inflammatory cytokines, such as

such as inducible nitric oxide synthase (iNOS), interleukin-6 (IL-6), and tumor necrosis factor-alpha (TNF- α), while concurrently enhancing genes associated with anti-inflammatory activity, including arginase-1 (Arg1), interleukin-10 (IL-10), and transforming growth factor-beta (TGF- β) (Supplementary Figs. 47, 48). Together, these results demonstrate that, at neutral pH, PtFeCuCoNi HEA NPs extinguish intracellular ROS, preserve cytoskeletal architecture, sustain angiogenic function, and reprogram macrophages toward tissue repair, thus maintaining redox homeostasis and protecting cells from oxidative injury (Fig. 7n).”

Supplementary Fig. 47 a By Real-time quantitative polymerase chain reaction (RT-qPCR), the relative expression of iNOS, TNF- α , and IL-6 mRNA in Raw264.7 cells was measured after 24 h of LPS stimulation ($n = 3$ independent biological replicates). **b** RT-qPCR measurements of mRNA expression (Arg-1, IL-10, and TGF- β) in Raw264.7 cells following 24 h of LPS stimulation ($n = 3$ independent biological replicates). Data are presented as means \pm SD, * $p < 0.05$, ** $p < 0.01$, *** $p <$

0.001, **** $p < 0.0001$, ns, not significant; one-way ANOVA with multiple comparisons test, all tests were two-sided. Source data are provided as a Source Data file.

Supplementary Fig. 48 Schematic depicting the effects of PtFeCuCoNi HEA NPs on macrophage polarization. M0 indicates macrophages, M1 indicates pro-inflammatory macrophages, M2 indicates anti-inflammatory macrophages, and LPS indicates lipopolysaccharide.

(8) The authors indicate that the artificial enzyme can activate the PFKFB3 pathway at the transcriptomic level. However, the authenticity of the activation results based solely on transcriptomic data remains to be further verified. It is recommended that the authors supplement the study with RT-qPCR testing of relevant factors to further confirm these results.

Response to comment:

We appreciate the reviewer's insightful comment highlighting the need for experimental

validation of the PFKFB3 pathway activation beyond transcriptomic evidence. To strengthen the reliability of our findings, we performed RT-qPCR analysis to confirm the expression patterns of key genes associated with glycolytic metabolism and oxidative stress regulation. Six representative genes were selected for validation, including upregulated (*PFKFB3*, *THBS1*, *HSPA8*) and downregulated (*CHAC1*, *ASNA*, *PSAT1*) targets. The RT-qPCR data showed that treatment with PtFeCuCoNi HEA NPs significantly increased *PFKFB3*, *THBS1*, and *HSPA8* expression, consistent with enhanced glycolytic and reparative activity, while *CHAC1*, *ASNA*, and *PSAT1* were significantly suppressed, indicating alleviation of oxidative stress-related responses. These results are in strong agreement with our RNA-seq findings and provide direct evidence that PtFeCuCoNi HEA nanozymes transcriptionally activate PFKFB3-mediated glycolysis while alleviating redox imbalance (Supplementary Fig. 49).

Page 26 in the revised manuscript: “Differential expression analysis identified significant upregulation of protective genes, including *PFKFB3*⁷⁸ (glycolysis) and *THBS1*⁷⁹ (extracellular matrix remodeling), together with downregulation of stress-associated genes such as *SLC7A11*⁶⁴, *PSAT1*⁸⁰, *CHAC1*, *MTHFD2*, and *SESN2* (Fig. 8b), suggesting a transcriptional switch from injury signaling to programs supporting metabolic recovery and structural repair. To validate these transcriptomic trends, we further performed RT-qPCR analysis of representative genes involved in glycolytic activation and oxidative stress response. Consistent with the RNA-seq results, PtFeCuCoNi HEA NPs treatment markedly upregulated *PFKFB3*, *THBS1*, and *HSPA8*, while significantly downregulating *CHAC1*, *ASNA*, and *PSAT1* compared with both H₂O₂ and Pt NPs groups (Supplementary Figs. 49). These results confirm that HEA NPs transcriptionally activate glycolysis- and repair-related programs while suppressing stress-associated pathways, thereby supporting the restoration of redox homeostasis and cellular recovery. Hierarchical clustering corroborated a shift toward a homeostatic expression profile after HEA NPs treatment (Fig. 8c).”

Supplementary Fig. 49 **a** Volcano plots of DEGs (gray: not significantly different genes; red: upregulated genes; blue: downregulated genes). **b** RT-qPCR data, the relative expression of PFKFB3, THBS1, HSPA8, CHAC1, ASNS, and PSAT1 mRNA in HUVEC cells was measured after incubating with PBS (Control), H₂O₂, Pt + H₂O₂, PtFeCuCoNi + H₂O₂. (*n* = 3 independent biological replicates). Data are presented as means ± SD; ns, not significant; **p* < 0.05, ***p* < 0.01, ****p* < 0.001, *****p* < 0.0001; one-way ANOVA with multiple comparisons test; all tests were two-sided. Source data are provided as a Source Data file.